# Decoding the Symmetry of Evolution Landscapes in Hopfield Neural Networks via Rectified Update Rule In-Depth Analysis

## Abstract

The Hopfield Neural Network (HNN) comprises $N$ binary neurons, yielding a state space of size $2^N$. Traditional update rules leave a neuron's next state unspecified when its input summation is zero, leading to symmetry-breaking artifacts and spurious cycles. To remedy this, we introduce the **Rectified Update Rule**, which retains each neuron's prior state in such tie scenarios, thereby restoring symmetry and ensuring stable convergence. Building upon this, we reformulate the **Hamming-Distance-Aware Rectified (HDAR) Update Rule**, considering the Hamming distance when memorizing two messages. This rule preserves full symmetry among the two memories and their negations and yields a complete taxonomy of dynamic regimes: convergence, self-cycle, hetero-cycle, and symmetric cycle. Importantly, we encapsulate these dynamics in **two central theorems**one characterizing single-message behavior and another for dual-messages regimeswith full proofs in Appendix. From these theorems, we derive corollaries that precisely quantify the counts and conditions of convergent versus cyclic states as functions of the network size and the Hamming distance. Extensive simulations, spanning exhaustive enumeration and Monte Carlo sampling, confirm all theoretical calculations.

## 1 Introduction

In recent years, deep neural networks (DNNs) have become the dominant paradigm across a wide spectrum of applications, including image classification (Krizhevsky et al., 2012), speech recognition (Vaswani et al., 2017), and natural language processing (Devlin et al., 2019). Despite their remarkable performance, the complexity of their feed-forward architectures renders them opaque "black-box" systems whose internal decision-making processes remain largely inscrutable. This lack of interpretability has emerged as a major barrier to deployment in high-stakes domains such as healthcare (Tjoa & Guan, 2020; Mienye et al., 2024), finance (Arrieta et al., 2020; Chen et al., 2023), and industry (Ahmed et al., 2022; Adadi & Berrada, 2018), where understanding model rationale is essential. For example, pinpointing the mechanistic basis by which a large language model (LLM) generates harmful or unsafe content remains extremely challenging, complicating efforts in model debugging, safety, and alignment.

These challenges have renewed interest in inherently more interpretable architectures. The Hopfield Neural Network (HNN), a classical model of associative memory, offers a principled alternative grounded in an explicit energy landscape that funnels input patterns toward stored memories acting as attractors. A recent breakthrough by Ramsauer et al. (2021) demonstrated that the Transformers attention mechanism is mathematically equivalent to the continuous-state update rule of modern Hopfield networks (MHNs), positioning HNN theory as a rigorous analytical lens for understanding the dynamics of contemporary DNNs.

The classical binary-state HNN was introduced by Hopfield in 1982 (Hopfield, 1982), integrating Hebbian learning (Hebb, 1949) with symmetric weight matrices to encode discrete memory messages. Hopfield showed that most initial states converge to fixed-point attractors, while others may fall into limit cycles. Subsequent work by McEliece et al. (1987) extended this analysis under synchronous updates, revealing one-step, two-step, and multi-step convergence behaviors, as well as

persistent oscillatory trajectorieslater confirmed empirically (Bao et al., 2022; Bao & Zhao, 2025). They conjectured that the basin of attraction might resemble a hypersphere of radius $N/2$, though no formal proof has been established.

A further unresolved challenge concerns the persistent discrepancy between the symmetric energy landscape implied by the symmetric weight matrix and the asymmetric evolution frequently observed in classical HNN dynamics. This asymmetry stems from a long-standing ambiguity in the update rule when a neurons input summation (local field) equals zero. Historically, this zero-summation case has been treated heuristicallymost commonly by arbitrarily assigning the next state to $+1$ or $-1$. Such ad hoc resolutions break the symmetry inherent in the energy formulation, introducing dynamic inconsistencies and obscuring the theoretical basis for both convergence guarantees and the emergence of cyclic trajectories in classical HNNs.

This work aims to investigate the update rule of HNNs and elucidate how the Hamming distance between an input pattern and stored memory messages governs the network's dynamic evolution. We focus on synchronous updates, where all neurons update simultaneously, and derive exact relationships between Hamming distance and network evolution. The analysis considers $N$-dimensional binary vectors with entries $\pm 1$. Our main contributions are as follows:

- We resolve a key ambiguity in the classical Hopfield update rule, where a neuron's next state is undefined when its input summation equals zero. To address this, we propose the **Rectified Update Rule**, which retains the neuron's previous state in such tie cases. This refinement preserves both symmetry and convergence properties, and offers insights potentially applicable to activation function design in DNNs.

- When memorizing two memory messages, we propose a **Hamming-Distance-Aware Rectified (HDAR) Update Rule**, which incorporates explicit dependence on Hamming distances. Under this rule, the network dynamics exhibit perfect symmetry and can be fully categorized into four regimes: convergence, self-cycle, hetero-cycle, and symmetric-cycle.

- We formalize these insights in **two principal theorems**: one characterizing all convergence and cyclic regimes for a single memory message, and another for the dual-messages. Proofs are provided in Appendix due to the space limitation. Building on these theorems, we derive corollaries that precisely enumerate convergent versus cyclic states as functions of the network size and the Hamming distance. Extensive simulations validate every theoretical prediction, underscoring the robustness and predictive accuracy of our analytical framework.

## 2 RECTIFIED UPDATE RULE

HNNs exemplify recurrent neural networks and are renowned globally as associative memory systems. The connection matrix $W$ serves as the network's "core", encoding all stored memory messages, $\{\xi^1, \xi^2, ..., \xi^M\}$ and fully determining the network's dynamics. Following the original formulation in Hopfield (1982), these messages are embedded using the Hebbian learning rule (Hebb, 1949), with the diagonal elements of $W$ explicitly set to zero to prevent self-connections.

$$W = \sum_{m=1}^{M} \xi^m (\xi^m)^\top - MI, \tag{1}$$

where $I$ is the identity matrix of $N \times N$, $N$ is the dimension of the memory message, $M$ is the number of the memory message, and the element in the $i$-th row and $j$-th column of the weight matrix $W$ is denoted as $w_{i,j}$. Besides, in any given message $\xi^m$, the state of each neuron is represented by a binary state variable $\xi \in -1, 1$, where $i = 1, ..., n$. The original update rule of HNNs is as follows,

$$v_i(t+1) = \begin{cases} +1, & \text{if } \sum_{j=1}^{N} w_{i,j} v_j(t) > 0, \\ -1, & \text{if } \sum_{j=1}^{N} w_{i,j} v_j(t) < 0. \end{cases} \tag{2}$$

Each neuron computes its input summation $X_i(t) = \sum_{j=1}^{N} w_{i,j} v_j(t)$, and then updates its state by comparing $X_i(t)$ to a threshold (typically 0). However, the original rule leaves the case $X_i(t) = 0$

undefined. Some implementations introduce an ad hoc sign (sgn) function that defaults to a fixed state ($+1$ or $-1$) when encountering this scenario, but such heuristics will disrupt the network's symmetry and destabilize its dynamics, as noted in Abu-Mostafa & St-Jacques (1985); Sompolinsky & Kanter (1986); Kanter & Sompolinsky (1987); McEliece et al. (1987); Komlós & Paturi (1988); Mazza (1997); Storkey (1997); Storkey & Valabregue (1999); Wang (2005); Agliari et al. (2013); Krotov & Hopfield (2016); Demircigil et al. (2017); Folli et al. (2018); Hwang et al. (2019).

$$\text{sign}\,[X_i(t)] := \left\{ \begin{array}{ll} v_i(t+1) = +1, & \text{if}\, X_i(t) \geq 0, \\ v_i(t+1) = -1, & \text{if}\, X_i(t) < 0. \end{array} \right. \quad \text{or} \quad \left\{ \begin{array}{ll} v_i(t+1) = +1, & \text{if}\, X_i(t) > 0, \\ v_i(t+1) = -1, & \text{if}\, X_i(t) \leq 0. \end{array} \right. \quad (3)$$

To address this issue, we propose a **Rectified Update Rule** as Equation 4, which retains a neuron's previous state when $X_i(t) = 0$, thereby preserving both symmetry and convergence.

$$v_i(t+1) = \left\{ \begin{array}{ll} +1, & \text{if}\, X_i(t) > 0, \\ v_i(t), & \text{if}\, X_i(t) = 0, \\ -1, & \text{if}\, X_i(t) < 0. \end{array} \right. \quad (4)$$

The remainder of this paper is centered on this formulation. Rigorous theoretical analysis, supported by extensive simulations, confirms its effectiveness and structural symmetry.

## 3 THEORETICAL RESULTS OF HNNs WHEN MEMORIZING ONE MESSAGE

Specifically, we designate one randomly chosen pattern as the memory message $\xi$. We first formalize the notions of convergence and cycle after $k$ steps, then present an theorem in case of memorizing single message. For any initial pattern $U(0)$, let $U(t)$ denote its state after $t$ update steps.

**Definition 1** (Fixed Point Convergence). *A network state sequence $V(0), V(1), V(2), \ldots$ converges to a fixed point $V^*$ if there exists a minimum time step $t \geq 0$ such that: $V(t+1) = V(t) = V^*$. The pattern stabilizes at $V^*$ after $t$ steps.*

**Definition 2** (Limit Cycle). *A network state sequence $V(0), V(1), V(2), \ldots$ settles into a limit cycle $\mathcal{C}$ if there exists a minimum integer transient time $k \geq 0$ and a minimum integer period $P \geq 2$ such that: $V(k+P) = V(k)$. The cycle is the set $\mathcal{C} = \{V(k), V(k+1), \ldots, V(k+P-1)\}$. The pattern enters the cycle after $k$ steps.*

**Theorem 1** (Dynamics with a Single Stored Memory). *Consider a HNN with $N$ neurons storing a single random binary memory message $\xi$, and let the weight matrix be $W = \xi\xi^\top - I$, where $I$ is the $N \times N$ identity matrix. Let $V$ be a perturbed version of $\xi$ at Hamming distance $r$. Under the Rectified Update Rule 4, the network exhibits the following dynamics:*

*Case 1: $N$ **even**.*

*(1) If $r < N/2$, the state $V$ converges to the memory message $\xi$ in a single update.*

*(2) If $r = N/2$, the state $V$ immediately enters a symmetric 2-cycle between $V$ and $-V$.*

*(3) If $r > N/2$, the state $V$ converges to the negation $\bar{\xi}$ ($\bar{\xi} = -\xi$) in a single update.*

*Case 2: $N$ **odd**.*

*(4) If $r < N/2$, the state $V$ converges to $\xi$ in a single update.*

*(5) If $r > N/2$, the state $V$ converges to $\bar{\xi}$ in a single update.*

*Proof.* The proof is provided in Appendix A.1 due to space constraints. $\square$

We visualize the above theorem as shown in Figure 1 to facilitate readers' understanding. From this result, we immediately deduce the following corollary:

**Corollary 1** (Quantity analysis). *When memorizing one memory message $\xi$, the convergence behavior of HNNs under Rectified Update Rule is influenced by the parity of the network size $N$:*

- *When $N$ is even, patterns at Hamming distance $N/2$ from the memory message become trapped in a symmetric-cycle, totaling $\binom{N}{N/2}$. The remaining $2^N - \binom{N}{N/2}$ patterns converge to either $\xi$ or its symmetric anti-message $\bar{\xi}$, with each attracting half of these patterns.*

- *When $N$ is odd, all $2^N$ patterns converge to either $\xi$ or $\bar{\xi}$, each counting $2^{N-1}$.*

*Proof.* See proof in Appendix Corollary A.1. □

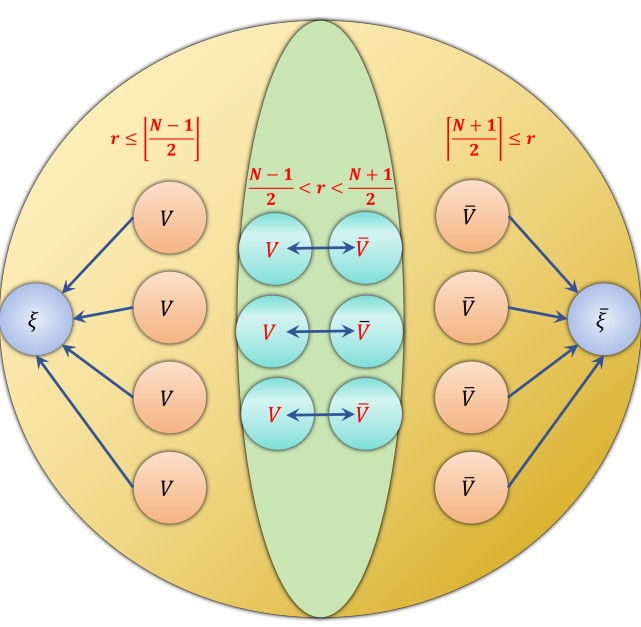

Figure 1: The evolution phenomena when memorizing one memory message. Notations: stable configurations (blue nodes), nodes converging to stable configurations (orange nodes), nodes belong to symmetric-cycle (cyan nodes).

## 4 THEORETICAL RESULTS OF HNNS WHEN MEMORIZING TWO MESSAGES

### 4.1 PREPARATION

For any two memory messages $\xi^1$ and $\xi^2$ over $N$ neurons, we decompose $\xi^1$ into two disjoint parts $A_1$, where $\xi^1$ and $\xi^2$ are identical; and $A_2$, where $\xi^1$ and $\xi^2$ are opposite. Formally denoted as:

$$\xi^1 = \begin{bmatrix} A_1 \\ A_2 \end{bmatrix}, \quad \xi^2 = \begin{bmatrix} A_1 \\ -A_2 \end{bmatrix}. \tag{5}$$

Given an arbitrary noise pattern $V$, we further refine $A_1$ and $A_2$ based on their alignment with $\xi^1$. Specifically, within $A_1$, $A_{11}$ be the neurons where $V$ matches $\xi^1$; and $A_{12}$, be the neurons where $V$ is flipped relative to $\xi^1$. An identical decomposition is applied to $A_2$, yielding $A_{21}$ and $A_{22}$. As a result, $V$ can be decomposed into four distinct parts: $V_{11}$, $V_{12}$, $V_{21}$ and $V_{22}$, expressed as:

$$V = \begin{bmatrix} V_{11} \\ V_{12} \\ V_{21} \\ V_{22} \end{bmatrix} = \begin{bmatrix} A_{11} \\ -A_{12} \\ A_{21} \\ -A_{22} \end{bmatrix}, \quad \xi^1 = \begin{bmatrix} A_{11} \\ A_{12} \\ A_{21} \\ A_{22} \end{bmatrix}, \quad \xi^2 = \begin{bmatrix} A_{11} \\ A_{12} \\ -A_{21} \\ -A_{22} \end{bmatrix}. \tag{6}$$

Let $r_1$ denote the Hamming distance between the noise pattern $V$ and the memory message $\xi^1$, $r_2$ the distance between $V$ and $\xi^2$, and $r_{12}$ the distance between $\xi^1$ and $\xi^2$. It is easily obtained that

$$\left\{ \begin{array}{c} r_1 = |A_{12}| + |A_{22}|, \\ r_2 = |A_{12}| + |A_{21}|, \\ r_{12} = |A_{21}| + |A_{22}|. \end{array} \right. \tag{7}$$

$|A_{11}|, |A_{12}|, |A_{21}|, |A_{22}|$ respectively represent the cardinalities of the corresponding subsets introduced earlier.

### 4.2 REFORMULATION OF THE RECTIFIED UPDATE RULE CONSIDERING THE HAMMING DISTANCE

When memorizing two messages $\xi^1$ and $\xi^2$, the weight matrix is $W = \xi^1(\xi^1)^\top + \xi^2(\xi^2)^\top - 2I$ in accordance with Equation 1. For any state $V$, the input to neuron $i$ is $X_i = \sum_{j=1}^{N} w_{i,j}v_j$, deduce it:

$$X_i = \xi_i^1(\xi^1)^\top V + \xi_i^2(\xi^2)^\top V - 2v_i = \xi_i^1 \left( N - 2r_1 \right) + \xi_i^2 \left( N - 2r_2 \right) - 2v_i. \tag{8}$$

By partitioning the neuron indices into those on which $\xi^1$ and $\xi^2$ agree, and those where they differ, we decompose the summation into contributions from aligned and anti-aligned components.

$$X_i = \left\{ \begin{array}{ll} 2\xi_i^1 \left( N - r_1 - r_2 - \xi_i^1 v_i \right), & \text{if } i \text{ in } A_1, \\ 2\xi_i^1 \left( r_2 - r_1 - \xi_i^1 v_i \right), & \text{if } i \text{ in } A_2. \end{array} \right. \tag{9}$$

Focusing specifically on the interaction between $\xi_i^1$ and $v_i$, we derive an update rule that explicitly integrates Hamming distance information into the neuron's transition dynamics. We refer to this refined mechanism as the **Hamming-Distance-Aware Rectified (HDAR) Update Rule**, which unifies our proposed rectified update strategy with a principled approach to managing pattern overlap considering the Hamming distance.

**Definition 3** (HDAR update rule). *When memorizing two messages $\xi^1$ and $\xi^2$, let $r_1$ denotes the Hamming distance between $V$ and $\xi^1$, $r_2$ the distance between $V$ and $\xi^2$. The HDAR update rule is:*

$$V_{11}(t + 1) = \left\{ \begin{array}{ll} V_{11}(t), & \text{if } r_1 + r_2 \leq N - 1, \\ -V_{11}(t), & \text{if } r_1 + r_2 > N - 1. \end{array} \right. \tag{10}$$

$$V_{12}(t + 1) = \left\{ \begin{array}{ll} -V_{12}(t), & \text{if } r_1 + r_2 < N + 1, \\ V_{12}(t), & \text{if } r_1 + r_2 \geq N + 1. \end{array} \right. \tag{11}$$

$$V_{21}(t + 1) = \left\{ \begin{array}{ll} V_{21}(t), & \text{if } r_2 - r_1 \geq 1, \\ -V_{21}(t), & \text{if } r_2 - r_1 < 1. \end{array} \right. \tag{12}$$

$$V_{22}(t + 1) = \left\{ \begin{array}{ll} -V_{22}(t), & \text{if } r_2 - r_1 > -1, \\ V_{22}(t), & \text{if } r_2 - r_1 \leq -1. \end{array} \right. \tag{13}$$

### 4.3 FOUR PHENOMENA OF EVOLUTION

Under the HDAR update rule 3, the network's dynamic behavior becomes both transparent and well-delineated: only **fixed-point convergence** and **cyclic trajectories** persist. When memorizing two messages, the network admits precisely four stable configurationsnamely, the two original messages and their respective negations. Cyclic behaviors, in turn, can be classified into three distinct categories: **self-cycle, hetero-cycle, and symmetric-cycle**, whose definitions are provided below.

- Self-cycle: The noise pattern $O$ enters a cycle between itself $O$ and another pattern $Q$, i.e., $O \leftrightarrow Q$.
- Hetero-cycle: The noise pattern $V$ updates in one step to a state $O$ that would lead to a self-cycle, then continues cycling according to the self-cycle behavior, i.e., $V \rightarrow O \leftrightarrow Q$.
- Symmetric-cycle: The noise pattern $P$ oscillates between itself $P$ and its anti-pattern $-P$, where they are symmetric, i.e., $P \leftrightarrow -P$.

Based on the cycle classifications above, we can obtain **three conditions for cycle identification**:

**Theorem 2** (Cycle in case of two memory messages). *When memorizing two messages $\xi_1$ and $\xi_2$, $r_1$ denotes the distance between $\xi_1$ and the noise pattern $V$, $r_2$ distance between $\xi_2$ and $V$. There is a $N$ neurons HNN, with the HDAR update rule. We can obtain the following three cycle conditions:*

1. *When $r_1 + r_2 \neq N, r_1 = r_2$ (**Condition I**):*

    - *When $r_1 + r_2 \leq N - 1$, If the noise pattern $V$ contains $V_{12}$ components, it will enter into hetero-cycle. Otherwise, it will exhibit self-cycle.*
    - *When $r_1 + r_2 \geq N + 1$, If the noise pattern $V$ contains $V_{11}$ components, it will enter into hetero-cycle. Otherwise, it will exhibit self-cycle.*

2. *When $r_1 + r_2 = N, r_1 \neq r_2$ (**Condition II**):*

    - *When $r_2 - r_1 \leq -1$, if the noise pattern $V$ contains $V_{21}$ components, it will enter into hetero-cycle. Otherwise, it will exhibit self-cycle.*
    - *When $r_2 - r_1 \geq 1$, if the noise pattern $V$ contains $V_{22}$ components, it will enter into hetero-cycle. Otherwise, it will exhibit self-cycle.*

3. *When $r_1 + r_2 = N, r_1 = r_2$ (**Condition III**):*

    - *At this time, the noise pattern will enter into a symmetric-cycle.*

*Proof.* The proof is provided in Appendix Theorem A.2 due to space constraints. ☐

Moreover, we observe that noise pattern evolution exhibits distinct behaviors depending on the parity of $N$ and the Hamming distance $r_{12}$ between the two memory messages $\xi^1$ and $\xi^2$.

**Corollary 2** (Evolution results with respect to $N$ and $r_{12}$). *Cycle phenomena are governed by whether $N$ and $r_{12}$ are even or odd, leading to predictable cycle under HDAR update rule, as follows:*

1. *$N$ **even**, $r_{12}$ **even**: All cycle conditions are satisfied, which incurs that $V$ can evolve into any of the three cycleselfcycle, heterocycle, or symmetric-cycle. The quantity $C_{\text{ee}}$ ($C_{\text{ee}}$ refers to the number of noise patterns in which the HNN eventually evolves into a cyclical state when $N$ is even and $r_{12}$ is even, $C_{\text{eo}}, C_{\text{oo}}, C_{\text{oo}}$ is defined similarly) is:*

$$
C_{\text{ee}} = 2 \left[ \binom{N - r_{12}}{\frac{N - r_{12}}{2}} \sum_{i = \frac{N - r_{12}}{2}}^{\frac{N-2}{2}} \binom{r_{12}}{i - \frac{N - r_{12}}{2}} + \binom{r_{12}}{\frac{r_{12}}{2}} \sum_{i = \frac{r_{12}}{2}}^{\frac{N-2}{2}} \binom{N - r_{12}}{i - \frac{r_{12}}{2}} \right] + \binom{r_{12}}{\frac{r_{12}}{2}} \binom{N - r_{12}}{\frac{N - r_{12}}{2}}.
$$

(14)

2. *$N$ **even**, $r_{12}$ **odd**: Any cycle conditions cannot be satisfied. Thus the quantity $C_{\text{eo}}$ is 0.*

3. *$N$ **odd**, $r_{12}$ **even**: The cycle condition I is satisfied, which incurs self-cycle or hetero-cycle. The quantity $C_{\text{oe}}$ is:*

$$
C_{\text{oe}} = 2 \binom{r_{12}}{\frac{r_{12}}{2}} \sum_{i = \frac{r_{12}}{2}}^{\frac{N-1}{2}} \binom{N - r_{12}}{i - \frac{r_{12}}{2}}.
$$

(15)

4. *$N$ **odd**, $r_{12}$ **odd**: The cycle condition II is satisfied, which incurs self-cycle or hetero-cycle. The quantity $C_{\text{oo}}$ is:*

$$
C_{\text{oo}} = 2 \binom{N - r_{12}}{\frac{N - r_{12}}{2}} \sum_{i = \frac{N - r_{12}}{2}}^{\frac{N-1}{2}} \binom{r_{12}}{i - \frac{N - r_{12}}{2}}.
$$

(16)

*Proof.* See proof in Appendix Corollary A.2. ☐

### 4.4 Convergence and Complete Convergence

We observe that the network's behavior transitions from fixedpoint convergence to cyclic trajectories as the Hamming distance to the memory message increases. To capture this transition, we define two critical domains and then present some corollaries that precisely quantifies their boundaries.

**Definition 4** (Convergence domain and complete convergence domain). *When memorizing two messages $\xi^1$ and $\xi^2$, each message and its anti-message defines a maximal Hamming distance $r_{\max}$, beyond which no pattern converges to that message under HNN dynamics. This delineates the boundary of the **convergence domain (CD)**. We further define the **complete convergence domain (CCD)** as the set of all patterns within a critical Hamming distance $r_{\mathrm{ccd}}$, where convergence to the corresponding memory message is guaranteed. For $r \leq r_{\mathrm{ccd}}$, the network consistently settles into the correct attractor. In contrast, for $r_{\mathrm{ccd}} < r \leq r_{\max}$, convergence behavior bifurcatessome patterns converge to a memory message, while others fall into cyclic trajectories.*

Although prior works informally discussed this idea, it lacked a precise mathematical formulation. Notably, when $r > r_{\max}$, the evolution trajectories follow three scenarios: self-cycle, hetero-cycle, or symmetric-cycle. In this work, we derive exact expressions for CD and CDD of memory messages.

**Corollary 3** (Bound of CCD). *If $r_{\mathrm{ccd}}$ satisfies*

$$2r_{\mathrm{ccd}} + 1 \leq r_{12} \leq N - (2r_{\mathrm{ccd}} + 1) \quad \text{and} \quad N \geq 4r_{\mathrm{ccd}} + 2, \tag{17}$$

*then any pattern $V$ with $r \leq r_{\mathrm{ccd}}$ will converge to its nearest memory message in finite update steps.*

*Proof.* See proof in Appendix Corollary A.3. ☐

**Corollary 4** (Relation between two bounds). *The minimum of $r_{\mathrm{ccd}}$ is $0$, and $r_{\max} > 0$. When $N$ is even and $r_{12}$ is even: $r_{\max} = \frac{N}{2} - 2$. For all other cases: $r_{\max} = \left\lfloor \frac{N}{2} \right\rfloor - 1$. The relation between these two bounds satisfies*

$$r_{\mathrm{ccd}} < r_{\max}. \tag{18}$$

*Proof.* See proof in Appendix Corollary A.4. ☐

To aid understanding, we visualize the full state space of $2^N$ patterns as a circle in Figure 2, with two memory messages and their negations acting as four "anchor" points. Each anchor defines a CD (shaded cyan) and a CCD (shaded pink). By definition, all patterns within a CCD deterministically converge to the nearest message. Patterns in the CD but outside the CCD display mixed behavior: some converge (pink $V$), while others enter self-cycles (gray $V$) or hetero-cycle (red $V$). Patterns outside any CD may still exhibit self-cycle or hetero-cycle, both within and beyond the CD. The central circular region corresponds to symmetric-cycle (green $V$), which emerge only when both $N$ and the $r_{12}$ are even. Notably, self-cycle caused by $r_1 = r_2$ occur strictly within the CD, whereas those induced by $r_1 + r_2 = N$ can occur inside or outside.

## 5 Experiments

The experimental environment configuration is as follows: i7-10750H CPU 2.60GHz, RAM 16G, NVIDIA GTX 1650, Python3.7. In the experiment, for a noise pattern update denoted as $V(t)$ after $t$ iterations, given the known cycle period of 2, we only track $V(t)$ and $V(t-1)$. Convergence is determined if $V(t) = V(t+1)$, indicating the noise pattern stabilizes at $V(t)$. A cycle is detected if $V(t-1) = V(t+1)$, signifying the noise is trapped in a cycle: $V(t-1) \to V(t) \to V(t-1) \to V(t)$.

### 5.1 Experiments when Memorizing One Message

When memorizing a single memory message, Theorem 1 establishes the existence of two stable configurations: the original message $\xi$ and its symmetric anti-message $\bar{\xi}$. In the case where $N$ is even, an intuitive observation arises: any pattern at a Hamming distance of $N/2$ from $\xi$ is also exactly $N/2$ away from $\bar{\xi}$, rendering it equidistant from both attractors. Consequently, such a pattern

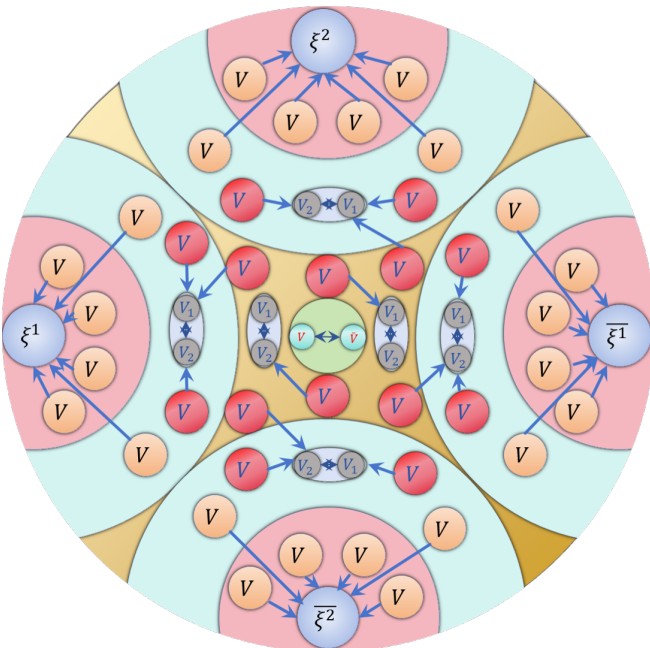

Figure 2: The evolution phenomena when memorizing two messages. Notations: stable configurations (blue $V$), nodes converging to stable configurations (orange $V$), nodes belong to self-cycle (gray $V$), nodes belong to hetero-cycle (red $V$), nodes belong to symmetric-cycle (green $V$); Complete convergence domains (shaded pink), convergence domains (shaded cyan).

lacks a preferential convergence direction and may fail to settle into either attractor. To verify this insight, we conduct the following experiments to confirm the theoretical conclusion.

A random $N$-bit binary vector is generated as the memory message to define the HNN and construct its weight matrix. All $2^N$ binary vectors-interpreted as noise patterns-are then individually updated using **Rectified Update Rule**, and their final states (convergent or cyclic) are recorded. The purpose of the experiment is to verify whether the experimental value of the cycling ratio obtained through experiments is equal to the theoretical value calculated using Collary 1. The calculation method for the cycling ratio is the number of noise patterns among all $2^N$ noise patterns in a HNN that fall into a cyclic state divided by $2^N$.

For small networks ($N \leq 20$), exhaustive enumeration is feasible, allowing for complete analysis of all patterns. By iterating over the full state space, we compute the exact number and proportion of patterns trapped in cycles. The results align precisely with theoretical calculations, validating our analytical framework. At the same time, we verified that the conditions required for a noise pattern to enter a cyclic state are precisely that it is at the same distance $N/2$ from $\xi^1$ and $\xi^2$, detailed examples can be found in **Case 1** of Appendix B.4.1.

For larger networks ($N > 20$), exhaustive analysis becomes infeasible, so we adopt Monte Carlo sampling. Here, each of the 10,000 sampled patterns is generated by independently setting each bit to $\pm 1$ with equal probability. A randomly generated memory message defines the weight matrix, and our experiments adopt the **Rectified Update Rule**. For each fixed N, the trial was repeated 50 times, recording the cycling ratio each time (the number of sampling patterns out of 10,000 that entered a cycling state, divided by 10,000). The reported results are the average values of these trials. To ensure statistical validity, both the memory message and the sample set are freshly randomized in each trial. Theoretical values are computed according to Corollary 1, retaining four decimal places. The observed cycling ratio consistently fall within the expected error margins when compared across both regimes ($N \leq 20$ and $N > 20$), confirming the accuracy of our derived formula. Detailed results are provided in Table 1.

Table 1: Comparison between theoretical and experimental values when memorizing one message.

| $N$(ODD) | THEORY | EXPERIMENT | $N$(EVEN) | THEORY | EXPERIMENT |
|---|---|---|---|---|---|
| 3 | 1 | 1 | 4 | 0.3750 | 0.3750 |
| 7 | 1 | 1 | 8 | 0.2734 | 0.2734 |
| 15 | 1 | 1 | 16 | 0.1964 | 0.1964 |
| 19 | 1 | 1 | 20 | 0.1762 | 0.1762 |
| $\vdots$ | $\vdots$ | $\vdots$ | $\vdots$ | $\vdots$ | $\vdots$ |
| 99 | 1 | 1 | 100 | 0.0796 | 0.0795 |
| 199 | 1 | 1 | 200 | 0.0563 | 0.0568 |
| 499 | 1 | 1 | 500 | 0.0357 | 0.0358 |
| 999 | 1 | 1 | 1000 | 0.0252 | 0.0250 |

### 5.2 EXPERIMENTS WHEN MEMORIZING TWO MESSAGES

Theoretical values are derived from Corollary 2, while experimental values are obtained through the following procedure. The primary objectives of the experiments are twofold: (i) to verify, for various $N$ and Hamming distances $r_{12}$ between the two stored messages, whether the theoretical and experimental cycle ratio are consistent; and (ii) to examine whether the noise-pattern landscape (including convergence behaviors and cyclic attractors) of the HNN can be accurately and comprehensively explained by the **HDAR Update Rule** and its associated theoretical framework. Depending on the network size $N$, the following two experimental schemes are adopted:

For small networks ($N \leq 20$), exhaustive enumeration is performed. First, $r_{12}$ is fixed as an odd or even number. Then, for each $N$, two $N$-bit messages separated by the Hamming distance $r_{12}$ are randomly generated, and the weight matrix is constructed. All $2^N$ noise patterns are updated using the **Rectified Update Rule**, and the experimental cycle ratio is recorded. Empirical results show that the evolution outcomes are entirely consistent with theoretical calculations. Simultaneously, a complete theoretical derivation of all $2^N$ noise patterns is performed using the **HDAR Update Rule**. It is then verified whether the theoretical derivation result for each noise pattern matches the resulting experimental result obtained using the **Rectified Update Rule**, including whether it converges to the same stable point or falls into the same type of cycle (self-cycle, hetero-cycle, or symmetric-cycle). The verification confirms that the theoretical derivations and experimental results are completely consistent, demonstrating the accuracy of the **HDAR Update Rule**.

For large networks ($N \geq 20$), Monte Carlo sampling is applied. We randomly generate 10,000 noise vectors per trial, with each bit independently set to $\pm 1$ with equal probability. Similarly, $r_{12}$ is first fixed as an odd or even number. Then, 50 independent trials are conducted. In each trial, a new pair of memory messages (with fixed $r_{12}$) and a fresh set of noise samples are generated. The noise samples are updated using the **Rectified Update Rule**, the cycle ratio for each trial is recorded (number of patterns entering cycles among 10,000 sampled patterns divided by 10,000), and the reported result is the average of these trials. Comparing this average experimental value with the theoretical value reveals that the cycle ratio consistently remains within the theoretical error margin, validating the accuracy of the convergence analysis.

Results for selected cases where $r_{12} = 3$ (odd) and $r_{12} = 4$ (even) are presented in Table 2 and Table 3, respectively, confirming that our theoretical results under the **HDAR Update Rule** are both precise and robust. Due to space limitations, additional statistical analyses and the line chart form of Table 2 and Table 3 are provided in the appendix B.3 and the analytical verification of the HNN's noise pattern landscape under the HDAR rule are provided in the appendix B.4.

## 6 CONCLUSION

First, our comprehensive literature review revealed no prior treatment of the zero-sum case in the Hopfield update rule. To address this gap, we propose a **Rectified Update Rule**, which preserves both symmetry and convergence stability. We believe this modification has potential implications beyond Hopfield networks, particularly for designing activation functions in deep neural architec-

Table 2: Comparison between theoretical and experimental values under odd $r_{12} = 3$.

| $N$(**ODD**) | $r_{12}$ | **THEORY** | **EXPERIMENT** | $N$(**EVEN**) | $r_{12}$ | **THEORY** | **EXPERIMENT** |
|---|---|---|---|---|---|---|---|
| 9 | 3 | 0.3125 | 0.3125 | 10 | 3 | 0 | 0 |
| 99 | 3 | 0.0812 | 0.0814 | 100 | 3 | 0 | 0 |
| 199 | 3 | 0.0569 | 0.0572 | 200 | 3 | 0 | 0 |
| 499 | 3 | 0.0358 | 0.0354 | 500 | 3 | 0 | 0 |
| 999 | 3 | 0.0253 | 0.0252 | 1000 | 3 | 0 | 0 |

Table 3: Comparison between theoretical and experimental values under even $r_{12} = 4$.

| $N$(**ODD**) | $r_{12}$ | **THEORY** | **EXPERIMENT** | $N$(**EVEN**) | $r_{12}$ | **THEORY** | **EXPERIMENT** |
|---|---|---|---|---|---|---|---|
| 9 | 4 | 0.375 | 0.3750 | 10 | 4 | 0.5703 | 0.5703 |
| 99 | 4 | 0.375 | 0.3751 | 100 | 4 | 0.4258 | 0.4247 |
| 199 | 4 | 0.375 | 0.3753 | 200 | 4 | 0.4106 | 0.4095 |
| 499 | 4 | 0.375 | 0.3750 | 500 | 4 | 0.3974 | 0.3971 |
| 999 | 4 | 0.375 | 0.3752 | 1000 | 4 | 0.3908 | 0.3913 |

tures. Second, when storing a single memory message, the Hopfield network exhibits two distinct behaviors: convergence to the message (stable state) or oscillation between two states (symmetric-cycle). This behavior is determined by the Hamming distance between the initial pattern and the memory message. We derive a closed-form expression for the number of stable states as a function of network size and validate it through extensive simulations. Third, for networks storing two memory messages, we identify **three distinct cyclic behaviors**self-cycle, hetero-cycle, and symmetric-cycleeach arising from specific alignments between noise patterns and memory messages. Finally, we introduce a **HDAR Update Rule** for the two-messages case, which not only explains these dynamic regimes but also reveals a previously unrecognized symmetry in the network's behavior. All theoretical calculations under the HDAR update rule are rigorously validated through simulation experiments.

Due to space constraints, we have relocated several intriguing phenomena and the full proof of two theorems and four corollaries to the appendix; readers seeking a deeper understanding are strongly encouraged to consult that material. Nevertheless, our study has two primary limitations. First, our analysis is confined to networks storing one or two memory messages. Second, all results assume synchronous updates; the asynchronous update regime remains unexplored and may exhibit qualitatively different convergence behavior. In future work, we will extend our theoretical framework to encompass multiple memory messages and investigate how asynchronous updating influences convergence dynamics, thereby broadening the applicability and robustness of our findings.

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

# Appendix

We did use Large Language Models (LLMs) during the writing process of the paper, but only for polishing the article, such as making sentences smoother and the application of vocabulary more precise, etc.

This appendix comprises Sections A, Section B, and Section C. Section A presents detailed proofs of the theorems and corollaries introduced in the main text. Section B provides additional experiments related to the proposed Rectified and HDAR update rules, demonstrating strong agreement between theoretical predictions and empirical results, while also revealing several interesting phenomena explainable by our theory. Section C reviews related works on HNNs.

APPENDIX CONTENTS

# A  PROOFS OF THE MAIN TEXT

## A.1  PROOF OF THEOREM 1

Assume a HNN comprising $N$ neurons with a single memory message $\xi$. Let $V$ denote a noise input at Hamming distance $r$ from $\xi$. The synaptic weight matrix is defined as

$$W = \begin{bmatrix} 0 & \xi_1\xi_2 & \cdots & \xi_1\xi_n \\ \xi_2\xi_1 & 0 & \cdots & \xi_2\xi_n \\ \vdots & \vdots & & \vdots \\ \xi_n\xi_1 & \xi_n\xi_2 & \cdots & 0 \end{bmatrix}. \tag{19}$$

where all diagonal entries are zero. We initialize the network state as $U(0) = V$ and evolve according to the Rectified Update Rule 4, explicitly preserving each neuron's previous state when its input sum is zero. This treatment is crucial for maintaining network symmetry and robustness and can be extended to activation functions in other deep neural architectures. For clarity, we restate the theorem below.

**Theorem A.1** (Evolution phenomena in case of one memory message)**.** *Consider a HNN with $N$ neurons storing a single random message $\xi$. The synaptic weight matrix is defined as $W = \xi\xi^\top - I$, where $I$ denotes the $N \times N$ identity matrix. Let $V$ be a perturbed version of $\xi$ at Hamming distance $r$. Under the rectified update rule 4, the network's evolution behaviors are characterized as follows:*

*Case 1: $N$ even.*

*(1) If $r < N/2$, then $V$ converges to $\xi$ after one update step.*

*(2) If $r = N/2$, then $V$ enters a symmetric-cycle between $V$ and $-V$ after zero update step.*

*(3) If $r > N/2$, then $V$ converges to $\bar{\xi}$ ($\bar{\xi} = -\xi$) after one update step.*

*Case 2: $N$ odd.*

*(4) If $r < N/2$, then $V$ converges to $\xi$ after one update step.*

*(5) If $r > N/2$, then $V$ converges to $\bar{\xi}$ after one update step.*

*Proof.* First, let's see the ordinary situation $r = 0$ and $r = N$.

- **Case 0.1:** $r = 0$. At this point, $V$ is the message $\xi$. Regardless of whether $N$ is even or odd, $\xi$ will converge to itself after $0$ steps.

- **Case 0.2:** $r = N$. At this point, $V$ is the message $\bar{\xi}$. Regardless of whether $N$ is even or odd, $\bar{\xi}$ will converge to itself after $0$ steps.

From these two ordinary situations, we observe that $\xi$ and $\bar{\xi}$ are stable configurations. Then we study the other patterns' evolution behaviors. Because the Hamming distance between $V$ and $\xi$ is $r$, without loss of generality, assume that the initial state $V$ satisfies $V_1 = -\xi^1, \ldots, V_r = -\xi^r$, and $V_i = \xi_i$, for $i = r + 1, \ldots, N$. When $r \leq \lfloor N/2 \rfloor$, we analyze the $i$-th component of $WV$ under the initialization $U(0) = V$. For clarity, consider the case $i = 1$ (i.e., $i \leq r$), and observe how the rectified update rule applies.

$$\begin{aligned} (WV)_1 =& 0 + \xi_1\xi_2(-\xi_2) + \cdots + \xi_1\xi_r(-\xi_r) + \\ & \xi_1\xi_{r+1}\xi_{r+1} + \cdots + \xi_1\xi_n\xi_n \\ =& \xi_1\left(0 - \xi_2^2 - \cdots - \xi_r^2 + \xi_{r+1}^2 + \cdots + \xi_n^2\right) \\ =& [N - 1 - 2(r-1)]\xi_1 = (N - 2r + 1)\xi_1. \end{aligned} \tag{20}$$

When $i > r$, consider $i = r + 1$ for clarity.

$$\begin{aligned} (WV)_{r+1} =& \xi_{r+1}\xi_1(-\xi_1) + \xi_{r+1}\xi_2(-\xi_2) + \cdots + \\ & \xi_{r+1}\xi_r(-\xi_r) + 0 + \cdots + \xi_{r+1}\xi_n\xi_n \\ =& \xi_{r+1}\left(-\xi_1^2 - \xi_2^2 - \cdots - \xi_r^2 + 0 + \cdots + \xi_n^2\right) \\ =& (N - 1 - 2r)\xi_{r+1}. \end{aligned} \tag{21}$$

Because $\lfloor N/2 \rfloor$ depends on whether $V$ is even or odd, we treat these cases separately:

- **$N$ even**: From Equation 20, when $1 \le i \le \frac{N}{2}$, $[W V]_i = \xi_i$. From Equation 21, when $\frac{N}{2} + 1 \le i \le N$, $[W V]_i = -\xi_i$. Hence, the noise pattern $V$ at Hamming distance $N/2$ from the memory message evolves as

$$
V(0) = \begin{pmatrix} -\xi_1 \\ \vdots \\ -\xi_{\frac{N}{2}} \\ \xi_{\frac{N}{2}+1} \\ \vdots \\ \xi_N \end{pmatrix} \rightarrow V(1) = \begin{pmatrix} \xi_1 \\ \vdots \\ \xi_{\frac{N}{2}} \\ -\xi_{\frac{N}{2}+1} \\ \vdots \\ -\xi_N \end{pmatrix} \rightarrow V(2) = \begin{pmatrix} -\xi_1 \\ \vdots \\ -\xi_{\frac{N}{2}} \\ \xi_{\frac{N}{2}+1} \\ \vdots \\ \xi_N \end{pmatrix}. \quad (22)
$$

- **$N$ odd**: For $1 \le i \le \lfloor N/2 \rfloor$, $[W V]_i = \xi_i$, and for $\lfloor N/2 \rfloor + 2 \le i \le N$, $[W V]_i = -\xi_i$, The central neuron $i = \lfloor N/2 \rfloor + 1$ satisfies $[W V]_i = 0$, so under the Rectified Update Rule it retains its previous value, preventing cyclic behavior.

In both scenarios, the parity of $N$ determines whether the pattern enters a twocycle (even $N$) or converges (odd $N$).

The Equation 22 shows that when a noisy pattern lies at Hamming distance $N/2$ from the memory message, it becomes trapped in a twocycle between itself and its negation. Intuitively, this cyclic behavior can be written as $V \rightarrow -V \rightarrow V \rightarrow -V \rightarrow \cdots$.

When $N$ is odd, regardless of the initial configuration $U(0) = V$, one update suffices to resolve the network's state. Any pattern with a Hamming distance less than $\lfloor \frac{N}{2} \rfloor$ from the original memory message $\xi$ converges to the original memory message $\xi$ itself, while patterns with a distance more than $\lfloor \frac{N}{2} \rfloor$ converge to its negation $\bar{\xi}$. $\qquad\square$

## A.2 PROOF OF COROLLARY 1

**Corollary A.1** (Quantity analysis). *When memorizing one message $\xi$, the convergence behavior of HNNs under Rectified update rule is influenced by the parity of the network size $N$:*

- *When $N$ is even, patterns at Hamming distance $\frac{N}{2}$ from the memory message become trapped in a symmetric-cycle, totaling $\binom{N}{N/2}$. The remaining $2^N - \binom{N}{N/2}$ patterns converge to either $\xi$ or its symmetric anti-message $\bar{\xi}$, with each attracting half of these patterns.*

- *When $N$ is odd, all $2^N$ patterns converge to either $\xi$ or $\bar{\xi}$, each attracting exactly $2^{N-1}$ patterns, and no symmetric cycle occur.*

*Proof.* According to Theorem 1, when $N$ is odd, the entire state space of $2^N$ patterns splits evenly: exactly $2^{N-1}$ patterns converge to the memory message $\xi$, and the remaining $2^{N-1}$ patterns converge to its symmetric anti-message $\bar{\xi}$.

When $N$ is even, all patterns at Hamming distance $N/2$ from $\xi$ become trapped in symmetric-cycle, totaling $\binom{N}{N/2}$ such patterns. The remaining $2^N - \binom{N}{N/2}$ patterns exhibit perfect symmetry: half converge to $\xi$ and half to $\bar{\xi}$. $\qquad\square$

## A.3 PROOF OF THEOREM 2

To prove Theorem 2, we will enumerate all possible evolutionary outcomes when memorizing two messages, and then organize the three cyclic conditions in Theorem 2 by combining the same conditions. For the convenience of reading, we will rephrase Theorem 2 below.

**Theorem A.2** (Cycle in case of two memory messages). *When memorizing two messages $\xi^1$ and $\xi^2$, $r_1$ denotes the distance between $\xi^1$ and the noise pattern $V$, $r_2$ distance between $\xi^2$ and $V$. There is a $N$ neurons HNN, with the HDAR update rule. We can obtain the following three cycle conditions:*

1. *When $r_1 + r_2 \neq N, r_1 = r_2$ (**Condition I**):*

   - *When $r_1 + r_2 \leq N - 1$, If the noise pattern $V$ contains $V_{12}$ components, it will enter hetero-cycle. Otherwise, it will exhibit self-cycle.*
   - *When $r_1 + r_2 \geq N + 1$, If the noise pattern $V$ contains $V_{11}$ components, it will enter hetero-cycle. Otherwise, it will exhibit self-cycle.*

2. *When $r_1 + r_2 = N, r_1 \neq r_2$ (**Condition II**):*

   - *When $r_2 - r_1 \leq -1$, if the noise pattern $V$ contains $V_{21}$ components, it will enter hetero-cycle. Otherwise, it will exhibit self-cycle.*
   - *When $r_2 - r_1 \geq 1$, if the noise pattern $V$ contains $V_{22}$ components, it will enter hetero-cycle. Otherwise, it will exhibit self-cycle.*

3. *When $r_1 + r_2 = N, r_1 = r_2$ (**Condition III**):*

   - *At this time, the noise pattern will enter a symmetric-cycle.*

*Proof.* First, let us clarify the notations:

- $V(t)$ represents the noise pattern after $t$ updates, where $V(0)$ is the original noise pattern. Similarly, $V_{11}(t)$ denotes the $V_{11}$ component after $t$ updates.

- $r(t)$ represents the Hamming distance after $t$ updates, with $r$ being the initial Hamming distance.

- $|A_{11}|, |A_{12}|, |A_{21}|, |A_{22}|$ represent the number of neurons in each part $A_{11}, A_{12}, A_{21}, A_{22}$ respectively.

Please note the following conventions for the ensuing proof:

- **Update Rules:** Every update employs our HDAR update rules (Equation 10 11 12 13 in the main text).

- **Repartition After Each Update:** After each application of the update rule, the noise pattern must be repartitioned according to Equation 6. As a result, some subsets $V_{11}, V_{12}, V_{21}, V_{22}$ may "disappear"i.e., certain indices no longer satisfy that subset's original definition and are absorbed into other subsets.

- **ThreeStep Iteration at Each Time Step $t$:**

   1. **Partition:** Divide the current state $V(t)$ into subsets via Equation 6.
   2. **Compute Distances:** Calculate the current Hamming distances $r_1(t)$ and $r_2(t)$.
   3. **Update:** Apply the HDAR update rules to each subset.

- **Terminology:** It should be noted that "unchanged" indicates that the subset retains its previous value during that update; "flipped" indicates that every bit in the subset is inverted.

CASE 1: WHEN THE NOISE PATTERN $V$ CONTAINS ONLY TWO OF THE FOUR COMPONENTS $V_{11}, V_{12}, V_{21}, V_{22}$

Since:

- $V_{11}$ and $V_{12}$ represent the identical parts between $\xi^1$ and $\xi^2$.

- $V_{21}$ and $V_{22}$ represent the opposite parts between $\xi^1$ and $\xi^2$.

- With $1 \leq r_{12} \leq N - 1$.

We have the constraints:

$$|V_{11}| + |V_{12}| \geq 1,$$
$$|V_{21}| + |V_{22}| \geq 1. \tag{23}$$

Therefore, the two components must consist of:

- One part from $\{V_{11}, V_{12}\}$ (identical parts).

- And one part from $\{V_{21}, V_{22}\}$ (opposite parts).

**1.1 When the noise pattern $V$ only contains $V_{11}$ and $V_{21}$:**

$$V = \begin{bmatrix} V_{11} \\ V_{21} \end{bmatrix} = \begin{bmatrix} A_{11} \\ A_{21} \end{bmatrix} \rightarrow \begin{bmatrix} A_{11} \\ A_{21} \end{bmatrix}. \tag{24}$$

In this case, $V = \xi^1$, $r_1 = 0$, and $r_2 = |A_{21}|$. Then:

- $r_2 + r_1 = |A_{21}| < N$, i.e., $r_2 + r_1 \leq N - 1$. According to Equation 10: $V_{11}(1) = A_{11}$ (remains unchanged).

- $r_2 - r_1 = |A_{21}| > 0$, i.e., $r_2 - r_1 \geq 1$. According to Equation 12: $V_{21}(1) = A_{21}$ (remains unchanged).

Therefore, $V(1) = V(0)$, which means the noise pattern $V$ converges to $\xi^1$ after zero update step.

**1.2 When the noise pattern $V$ only contains $V_{11}$ and $V_{22}$:**

$$V = \begin{bmatrix} V_{11} \\ V_{22} \end{bmatrix} = \begin{bmatrix} A_{11} \\ -A_{22} \end{bmatrix} \rightarrow \begin{bmatrix} A_{11} \\ -A_{22} \end{bmatrix}. \tag{25}$$

In this case, $V = \xi^2$, $r_1 = |A_{22}|$, and $r_2 = 0$. Then:

- $r_2 + r_1 = |A_{22}| < N$, i.e., $r_2 + r_1 \leq N - 1$. According to Equation 10: $V_{11}(1) = A_{11}$ (remains unchanged).

- $r_2 - r_1 = -|A_{22}| < 0$, i.e., $r_2 - r_1 \leq -1$. According to Equation 13: $V_{22}(1) = -A_{22}$ (remains unchanged).

Therefore, $V(1) = V(0)$, which means the noise pattern $V$ converges to $\xi^2$ after zero update step.

**1.3 When the noise pattern $V$ only contains $V_{12}$ and $V_{21}$:**

$$V = \begin{bmatrix} V_{12} \\ V_{21} \end{bmatrix} = \begin{bmatrix} -A_{12} \\ A_{21} \end{bmatrix} \rightarrow \begin{bmatrix} -A_{12} \\ A_{21} \end{bmatrix}. \tag{26}$$

In this case, $V = \bar{\xi}^2$, $r_1 = |A_{12}|$, and $r_2 = |A_{12}| + |A_{21}|$. Then:

- $r_2 + r_1 = |A_{12}| + |A_{21}| + |A_{12}| = N + |A_{12}| \geq N + 1$. According to Equation 11: $V_{12}(1) = -A_{12}$ (remains unchanged).

- $r_2 - r_1 = |A_{21}| > 0$, i.e., $r_2 - r_1 \geq 1$. According to Equation 12: $V_{21}(1) = A_{21}$ (remains unchanged).

Therefore, $V(1) = V(0)$, which means the noise pattern $V$ converges to $\bar{\xi}^2$ after zero update step.

**1.4 When the noise pattern $V$ only contains $V_{12}$ and $V_{22}$:**

$$V = \begin{bmatrix} V_{12} \\ V_{22} \end{bmatrix} = \begin{bmatrix} -A_{12} \\ -A_{22} \end{bmatrix} \rightarrow \begin{bmatrix} -A_{12} \\ -A_{22} \end{bmatrix}. \tag{27}$$

In this case, $V = \bar{\xi}^1$, $r_1 = |A_{12}| + |A_{22}|$, and $r_2 = |A_{12}|$. Then:

- $r_2 + r_1 = |A_{12}| + |A_{12}| + |A_{22}| = |A_{12}| + N \geq N + 1$. According to Equation 11: $V_{12}(1) = -A_{12}$ (remains unchanged).

- $r_2 - r_1 = -|A_{22}| < 0$, i.e., $r_2 - r_1 \leq -1$. According to Equation 13: $V_{22}(1) = -A_{22}$ (remains unchanged).

Therefore, $V(1) = V(0)$, which means the noise pattern $V$ converges to $\bar{\xi}^1$ after zero update step.

CASE 2: WHEN THE NOISE PATTERN $V$ CONTAINS ONLY THREE OF THE FOUR COMPONENTS $V_{11}, V_{12}, V_{21}, V_{22}$

Since any three components can be selected, there are a total of $\binom{4}{3} = 4$ possible combinations.

**2.1 When** $V = \begin{bmatrix} V_{12} \\ V_{21} \\ V_{22} \end{bmatrix} = \begin{bmatrix} -A_{12} \\ A_{21} \\ -A_{22} \end{bmatrix}, \xi^1 = \begin{bmatrix} A_{12} \\ A_{21} \\ A_{22} \end{bmatrix}, \xi^2 = \begin{bmatrix} A_{12} \\ -A_{21} \\ -A_{22} \end{bmatrix}.$

In this case, $r_1 = |A_{12}| + |A_{22}|$, and $r_2 = |A_{12}| + |A_{21}|$. Then:

- $r_2 + r_1 = |A_{12}| + |A_{21}| + |A_{12}| + |A_{22}| = |A_{12}| + N \geq N + 1$. According to Equation 11: $V_{12}(1) = -A_{12}$ (remains unchanged)

The components of $V_{21}$ and $V_{22}$ need to be categorized and discussed separately:

**2.1.1 If** $r_2 + r_1 \geq N + 1, -1 < r_2 - r_1 < 1$, **i.e.,** $r_1 = r_2$, $|A_{21}| = |A_{22}|$:

$$V = \begin{bmatrix} V_{12} \\ V_{21} \\ V_{22} \end{bmatrix} = \begin{bmatrix} -A_{12} \\ A_{21} \\ -A_{22} \end{bmatrix} = \begin{bmatrix} -A_{12} \\ -A_{21} \\ A_{22} \end{bmatrix} = \begin{bmatrix} -A_{12} \\ A_{21} \\ -A_{22} \end{bmatrix}. \tag{28}$$

- $r_2 - r_1 = |A_{21}| - |A_{22}| = 0$, i.e., $-1 < r_2 - r_1 < 1$. According to Equation 12 and 13: $V_{21}(1) = -A_{21}$ (flipped); $V_{22}(1) = A_{22}$ (flipped).

Therefore, $V(1) = \begin{bmatrix} -A_{12} \\ -A_{21} \\ A_{22} \end{bmatrix}$, where $r_1(1) = |A_{12}| + |A_{21}|$ and $r_2(1) = |A_{12}| + |A_{22}|$. Then:

- $r_2(1) + r_1(1) = |A_{12}| + |A_{22}| + |A_{12}| + |A_{21}| = |A_{12}| + N \geq N + 1$.. According to Equation 11: $V_{12}(2) = -A_{12}$ (remains unchanged).

- $r_2(1) - r_1(1) = |A_{21}| - |A_{22}| = 0$, i.e., $-1 < r_2(1) - r_1(1) < 1$.. According to Equation 12 and 13: $V_{21}(2) = A_{21}$ (flipped); $V_{22}(2) = -A_{22}$ (flipped).

Therefore, $V(2) = \begin{bmatrix} -A_{12} \\ A_{21} \\ -A_{22} \end{bmatrix} = V(0)$. The noise pattern $V$ enters a self-cycle after zero update step.

**2.1.2 If** $r_2 + r_1 \geq N + 1, r_2 - r_1 \geq 1$, **i.e.** $|A_{21}| > |A_{22}|$:

$$V = \begin{bmatrix} V_{12} \\ V_{21} \\ V_{22} \end{bmatrix} = \begin{bmatrix} -A_{12} \\ A_{21} \\ -A_{22} \end{bmatrix} = \begin{bmatrix} -A_{12} \\ A_{21} \\ A_{22} \end{bmatrix} = \begin{bmatrix} -A_{12} \\ A_{21} \\ A_{22} \end{bmatrix}. \tag{29}$$

- $r_2 - r_1 = |A_{21}| - |A_{22}| > 0$, i.e., $r_2 - r_1 \geq 1$. According to Equation 12 and 13: $V_{21}(1) = A_{21}$ (remains unchanged); $V_{22}(1) = A_{22}$ (flipped).

In conclusion, $V(1) = \begin{bmatrix} -A_{12} \\ A_{21} \\ A_{22} \end{bmatrix}$. At this point, due to the flipping of the $V_{22}$ component, the situation becomes identical to Case 1.3. After one more update, the pattern remains unchanged. Therefore, $V(2) = V(1)$, and the noise pattern $V$ will converge to $\bar{\xi}^2$ after zero update step.

**2.1.3 If $r_2 + r_1 \geq N + 1, r_2 - r_1 \leq -1$, i.e. $|A_{21}| < |A_{22}|$:**

$$V = \begin{bmatrix} V_{12} \\ V_{21} \\ V_{22} \end{bmatrix} = \begin{bmatrix} -A_{12} \\ A_{21} \\ -A_{22} \end{bmatrix} = \begin{bmatrix} -A_{12} \\ -A_{21} \\ -A_{22} \end{bmatrix} = \begin{bmatrix} -A_{12} \\ -A_{21} \\ -A_{22} \end{bmatrix}. \tag{30}$$

- $r_2 - r_1 = |A_{21}| - |A_{22}| < 0$, i.e., $r_2 - r_1 \leq -1$. According to Equation 12 and 13: $V_{21}(1) = -A_{21}$ (flipped); $V_{22}(1) = -A_{22}$ (remains unchanged).

In conclusion, $V(1) = \begin{bmatrix} -A_{12} \\ -A_{21} \\ -A_{22} \end{bmatrix}$. At this point, due to the flipping of the $V_{21}$ component, the situation becomes identical to Case 1.4. After one more update, the pattern remains unchanged. Therefore, $V(2) = V(1)$, and the noise pattern $V$ will converge to $\bar{\xi}^1$ after zero update step.

**2.2 When $V = \begin{bmatrix} V_{11} \\ V_{21} \\ V_{22} \end{bmatrix} = \begin{bmatrix} A_{11} \\ A_{21} \\ -A_{22} \end{bmatrix}, \xi^1 = \begin{bmatrix} A_{11} \\ A_{21} \\ A_{22} \end{bmatrix}, \xi^2 = \begin{bmatrix} A_{11} \\ -A_{21} \\ -A_{22} \end{bmatrix}.$**

In this case, $r_1 = |A_{22}|$, and $r_2 = |A_{21}|$. Then:

- $r_2 + r_1 = |A_{21}| + |A_{22}| = N - |A_{11}| \leq N - 1$. According to Equation 10: $V_{11}(1) = A_{11}$ (remains unchanged).

The components of $V_{21}$ and $V_{22}$ need to be categorized and discussed separately:

**2.2.1 If $r_2 + r_1 \leq N - 1, -1 < r_2 - r_1 < 1$, i.e., $r_1 = r_2, |A_{21}| = |A_{22}|$:**

$$V = \begin{bmatrix} V_{11} \\ V_{21} \\ V_{22} \end{bmatrix} = \begin{bmatrix} A_{11} \\ A_{21} \\ -A_{22} \end{bmatrix} = \begin{bmatrix} A_{11} \\ -A_{21} \\ A_{22} \end{bmatrix} = \begin{bmatrix} A_{11} \\ A_{21} \\ -A_{22} \end{bmatrix}. \tag{31}$$

- $r_2 - r_1 = |A_{21}| - |A_{22}| = 0$, i.e., $-1 < r_2 - r_1 < 1$. According to Equation 12:and 13: $V_{21}(1) = -A_{21}$ (flipped); $V_{22}(1) = A_{22}$ (flipped).

In conclusion, $V(1) = \begin{bmatrix} A_{11} \\ -A_{21} \\ A_{22} \end{bmatrix}$, where $r_1(1) = |A_{21}|$ and $r_2(1) = |A_{22}|$. Then:

- $r_2 + r_1 = |A_{22}| + |A_{21}| = N - |A_{11}| \leq N - 1$. According to Equation 10: $V_{11}(2) = A_{11}$ (remains unchanged).

- $r_2(1) - r_1(1) = |A_{21}| - |A_{22}| = 0$, i.e., $-1 < r_2(1) - r_1(1) < 1$.. According to Equation 12 and 13: $V_{21}(2) = A_{21}$ (flipped); $V_{22}(2) = -A_{22}$ (flipped).

Therefore, $V(2) = \begin{bmatrix} A_{11} \\ A_{21} \\ -A_{22} \end{bmatrix} = V(0)$. The noise pattern $V$ enters a self-cycle after zero update step.

**2.2.2 If $r_2 + r_1 \leq N - 1, r_2 - r_1 \geq 1$, i.e. $|A_{21}| > |A_{22}|$:**

$$V = \begin{bmatrix} V_{11} \\ V_{21} \\ V_{22} \end{bmatrix} = \begin{bmatrix} A_{11} \\ A_{21} \\ -A_{22} \end{bmatrix} = \begin{bmatrix} A_{11} \\ A_{21} \\ A_{22} \end{bmatrix} = \begin{bmatrix} A_{11} \\ A_{21} \\ A_{22} \end{bmatrix}. \tag{32}$$

- $r_2 - r_1 = |A_{21}| - |A_{22}| > 0$, i.e., $r_2 - r_1 \geq 1$. According to Equation 12:and 13: $V_{21}(1) = A_{21}$ (remains unchanged); $V_{22}(1) = A_{22}$ (flipped).

In conclusion, $V(1) = \begin{bmatrix} A_{11} \\ A_{21} \\ A_{22} \end{bmatrix}$. At this point, due to the flipping of the $V_{22}$ component, the situation becomes identical to Case 1.1. After one more update, the pattern remains unchanged. Therefore, $V(2) = V(1)$, and the noise pattern $V$ will converge to $\xi^1$ after zero update step.

**2.2.3 If $r_2 + r_1 \leq N - 1$, $r_2 - r_1 \leq -1$, i.e. $|A_{21}| < |A_{22}|$:**

$$V = \begin{bmatrix} V_{11} \\ V_{21} \\ V_{22} \end{bmatrix} = \begin{bmatrix} A_{11} \\ A_{21} \\ -A_{22} \end{bmatrix} = \begin{bmatrix} A_{11} \\ -A_{21} \\ -A_{22} \end{bmatrix} = \begin{bmatrix} A_{11} \\ -A_{21} \\ -A_{22} \end{bmatrix}. \tag{33}$$

- $r_2 - r_1 = |A_{21}| - |A_{22}| < 0$, i.e., $r_2 - r_1 \leq -1$. According to Equation 12:and 13: $V_{21}(1) = -A_{21}$ (flipped); $V_{22}(1) = -A_{22}$ (remains unchanged).

In conclusion, $V(1) = \begin{bmatrix} A_{11} \\ -A_{21} \\ -A_{22} \end{bmatrix}$. At this point, due to the flipping of the $V_{21}$ component, the situation becomes identical to Case 1.2. After one more update, the pattern remains unchanged. Therefore, $V(2) = V(1)$, and the noise pattern $V$ will converge to $\xi^2$ after zero update step.

**2.3 When $V = \begin{bmatrix} V_{11} \\ V_{12} \\ V_{22} \end{bmatrix} = \begin{bmatrix} A_{11} \\ -A_{12} \\ -A_{22} \end{bmatrix}, \xi^1 = \begin{bmatrix} A_{11} \\ A_{12} \\ A_{22} \end{bmatrix}, \xi^2 = \begin{bmatrix} A_{11} \\ A_{12} \\ -A_{22} \end{bmatrix}$.**

In this case, $r_1 = |A_{12}| + |A_{22}|$, and $r_2 = |A_{12}|$. Then:

- $r_2 - r_1 = -|A_{22}| \leq -1$. According to Equation 13: $V_{22}(1) = -A_{22}$ (remains unchanged).

The components of $V_{11}$ and $V_{12}$ need to be categorized and discussed separately:

**2.3.1 If $N - 1 < r_2 + r_1 < N + 1, r_2 - r_1 \leq -1$, i.e., $r_1 + r_2 = N$, $|A_{11}| = |A_{12}|$:**

$$V = \begin{bmatrix} V_{11} \\ V_{12} \\ V_{22} \end{bmatrix} = \begin{bmatrix} A_{11} \\ -A_{12} \\ -A_{22} \end{bmatrix} = \begin{bmatrix} -A_{11} \\ A_{12} \\ -A_{22} \end{bmatrix} = \begin{bmatrix} A_{11} \\ -A_{12} \\ -A_{22} \end{bmatrix}. \tag{34}$$

- $r_2 + r_1 = |A_{12}| + |A_{22}| + |A_{12}| = |A_{11}| + |A_{22}| + |A_{12}| = N$, i.e., $N - 1 < r_2 + r_1 < N + 1$. According to Equation 10 and 11: $V_{11}(1) = -A_{11}$ (flipped); $V_{12}(1) = A_{12}$ (flipped).

In conclusion, $V(1) = \begin{bmatrix} -A_{11} \\ A_{12} \\ -A_{22} \end{bmatrix}$, where $r_1(1) = |A_{11}| + |A_{22}|$ and $r_2(1) = |A_{11}|$. Then:

- $r_2(1) - r_1(1) = -|A_{22}| \leq -1$. According to Equation 13: $V_{22}(2) = -A_{22}$ (remains unchanged).

- $r_2(1) + r_1(1) = |A_{11}| + |A_{11}| + |A_{22}| = |A_{11}| + |A_{12}| + |A_{22}| = N$, i.e., $N - 1 < r_2(1) + r_1(1) < N + 1$.. According to Equation 10 and 11: $V_{11}(2) = A_{11}$ (flipped); $V_{12}(2) = -A_{12}$ (flipped).

Therefore, $V(2) = \begin{bmatrix} A_{11} \\ -A_{12} \\ -A_{22} \end{bmatrix} = V(0)$. The noise pattern $V$ enters a self-cycle after zero update step.

**2.3.2 If $r_2 + r_1 \leq N - 1$, $r_2 - r_1 \leq -1$, i.e. $|A_{11}| > |A_{12}|$:**

$$V = \begin{bmatrix} V_{11} \\ V_{12} \\ V_{22} \end{bmatrix} = \begin{bmatrix} A_{11} \\ -A_{12} \\ -A_{22} \end{bmatrix} = \begin{bmatrix} A_{11} \\ A_{12} \\ -A_{22} \end{bmatrix} = \begin{bmatrix} A_{11} \\ A_{12} \\ -A_{22} \end{bmatrix}. \tag{35}$$

- $r_2 + r_1 = |A_{12}| + |A_{22}| + |A_{12}| < |A_{11}| + |A_{22}| + |A_{12}| = N$, i.e., $r_2 + r_1 \leq N - 1$. According to Equation 10:and 11: $V_{11}(1) = A_{11}$ (remains unchanged); $V_{12}(1) = A_{12}$ (flipped).

In conclusion, $V(1) = \begin{bmatrix} A_{11} \\ A_{12} \\ -A_{22} \end{bmatrix}$. At this point, due to the flipping of the $V_{12}$ component, the situation becomes identical to Case 1.2. After one more update, the pattern remains unchanged. Therefore, $V(2) = V(1)$, and the noise pattern $V$ will converge to $\xi^2$ after zero update step.

**2.3.3 If $r_2 + r_1 \geq N + 1, r_2 - r_1 \leq -1$, i.e. $|A_{11}| < |A_{12}|$:**

$$V = \begin{bmatrix} V_{11} \\ V_{12} \\ V_{22} \end{bmatrix} = \begin{bmatrix} A_{11} \\ -A_{12} \\ -A_{22} \end{bmatrix} = \begin{bmatrix} -A_{11} \\ -A_{12} \\ -A_{22} \end{bmatrix} = \begin{bmatrix} -A_{11} \\ -A_{12} \\ -A_{22} \end{bmatrix}. \tag{36}$$

.

- $r_2 + r_1 = |A_{12}| + |A_{22}| + |A_{12}| > |A_{11}| + |A_{22}| + |A_{12}| = N$, i.e., $r_2 + r_1 \geq N + 1$. According to Equation 10:and 11: $V_{11}(1) = -A_{11}$ (flipped); $V_{12}(1) = -A_{12}$ (remains unchanged).

In conclusion, $V(1) = \begin{bmatrix} -A_{11} \\ -A_{12} \\ -A_{22} \end{bmatrix}$. At this point, due to the flipping of the $V_{11}$ component, the situation becomes identical to Case 1.2. After one more update, the pattern remains unchanged. Therefore, $V(2) = V(1)$, and the noise pattern $V$ will converge to $\bar{\xi}^1$ after zero update step.

**2.4 When $V = \begin{bmatrix} V_{11} \\ V_{12} \\ V_{21} \end{bmatrix} = \begin{bmatrix} A_{11} \\ -A_{12} \\ A_{21} \end{bmatrix}, \xi^1 = \begin{bmatrix} A_{11} \\ A_{12} \\ A_{21} \end{bmatrix}, \xi^2 = \begin{bmatrix} A_{11} \\ A_{12} \\ -A_{21} \end{bmatrix}.$**

In this case, $r_1 = |A_{12}|$, and $r_2 = |A_{12}| + |A_{21}|$. Then:

- $r_2 - r_1 = |A_{21}| \geq 1$. According to Equation 12: $V_{21}(1) = A_{21}$ (remains unchanged).

The components of $V_{11}$ and $V_{12}$ need to be categorized and discussed separately:

**2.4.1 If $N - 1 \leq r_2 + r_1 \leq N + 1, r_2 - r_1 \geq 1$, i.e., $r_1 + r_2 = N, |A_{11}| = |A_{12}|$:**

$$V = \begin{bmatrix} V_{11} \\ V_{12} \\ V_{21} \end{bmatrix} = \begin{bmatrix} A_{11} \\ -A_{12} \\ A_{21} \end{bmatrix} = \begin{bmatrix} -A_{11} \\ A_{12} \\ A_{21} \end{bmatrix} = \begin{bmatrix} A_{11} \\ -A_{12} \\ A_{21} \end{bmatrix}. \tag{37}$$

- $r_2 + r_1 = |A_{12}| + |A_{21}| + |A_{12}| = |A_{11}| + |A_{21}| + |A_{12}| = N$, i.e., $N - 1 < r_2 + r_1 < N + 1$. According to Equation 10:and 11: $V_{11}(1) = -A_{11}$ (flipped); $V_{12}(1) = A_{12}$ (flipped).

In conclusion, $V(1) = \begin{bmatrix} -A_{11} \\ A_{12} \\ A_{21} \end{bmatrix}$, where $r_1(1) = |A_{11}|$ and $r_2(1) = |A_{11}| + |A_{21}|$. Then:

- $r_2(1) - r_1(1) = |A_{21}| \geq 1$. According to Equation 12: $V_{21}(2) = A_{21}$ (remains unchanged).

- $r_2(1) + r_1(1) = |A_{11}| + |A_{11}| + |A_{21}| = |A_{11}| + |A_{12}| + |A_{21}| = N$, i.e., $N - 1 < r_2(1) + r_1(1) < N + 1$.. According to Equation 10 and 11: $V_{11}(2) = A_{11}$ (flipped); $V_{12}(2) = -A_{12}$ (flipped).

Therefore, $V(2) = \begin{bmatrix} A_{11} \\ -A_{12} \\ A_{21} \end{bmatrix} = V(0)$. The noise pattern $V$ enters a self-cycle after zero update step.

**2.4.2 If $r_2 + r_1 \leq N - 1, r_2 - r_1 \geq 1$, i.e. $|A_{11}| > |A_{12}|$:**

$$V = \begin{bmatrix} V_{11} \\ V_{12} \\ V_{21} \end{bmatrix} = \begin{bmatrix} A_{11} \\ -A_{12} \\ A_{21} \end{bmatrix} = \begin{bmatrix} A_{11} \\ A_{12} \\ A_{21} \end{bmatrix} = \begin{bmatrix} A_{11} \\ A_{12} \\ A_{21} \end{bmatrix}. \tag{38}$$

- $r_2 + r_1 = |A_{12}| + |A_{21}| + |A_{12}| < |A_{11}| + |A_{21}| + |A_{12}| = N$, i.e., $r_2 + r_1 \leq N - 1$. According to Equation 10:and 11: $V_{11}(1) = A_{11}$ (remains unchanged); $V_{12}(1) = A_{12}$ (flipped).

In conclusion, $V(1) = \begin{bmatrix} A_{11} \\ A_{12} \\ A_{21} \end{bmatrix}$. At this point, due to the flipping of the $V_{12}$ component, the situation becomes identical to Case 1.1. After one more update, the pattern remains unchanged. Therefore, $V(2) = V(1)$, and the noise pattern $V$ will converge to $\xi^1$ after zero update step.

**2.4.3 If $r_2 + r_1 \geq N + 1, r_2 - r_1 \geq 1$, i.e. $|A_{11}| < |A_{12}|$:**

$$V = \begin{bmatrix} V_{11} \\ V_{12} \\ V_{21} \end{bmatrix} = \begin{bmatrix} A_{11} \\ -A_{12} \\ A_{21} \end{bmatrix} = \begin{bmatrix} -A_{11} \\ -A_{12} \\ A_{21} \end{bmatrix} = \begin{bmatrix} -A_{11} \\ -A_{12} \\ A_{21} \end{bmatrix}. \tag{39}$$

- $r_2 + r_1 = |A_{12}| + |A_{21}| + |A_{12}| > |A_{11}| + |A_{21}| + |A_{12}| = N$, i.e., $r_2 + r_1 \geq N + 1$. According to Equation 10:and 11: $V_{11}(1) = -A_{11}$ (flipped); $V_{12}(1) = -A_{12}$ (remains unchanged).

In conclusion, $V(1) = \begin{bmatrix} -A_{11} \\ -A_{12} \\ A_{21} \end{bmatrix}$. At this point, due to the flipping of the $V_{11}$ component, the situation becomes identical to Case 1.3. After one more update, the pattern remains unchanged. Therefore, $V(2) = V(1)$, and the noise pattern $V$ will converge to $\bar{\xi}^2$ after zero update step.

CASE 3: ALL FOUR SEGMENTS $V_{11}, V_{12}, V_{21}, V_{22}$ OF THE NOISE PATTERN $V$ APPEAR

$$V = \begin{bmatrix} V_{11} \\ V_{12} \\ V_{21} \\ V_{22} \end{bmatrix} = \begin{bmatrix} A_{11} \\ -A_{12} \\ A_{21} \\ -A_{22} \end{bmatrix}, \quad \xi^1 = \begin{bmatrix} A_{11} \\ A_{12} \\ A_{21} \\ A_{22} \end{bmatrix}, \quad \xi^2 = \begin{bmatrix} A_{11} \\ A_{12} \\ -A_{21} \\ -A_{22} \end{bmatrix}. \tag{40}$$

We notice that $r_1 = |A_{12}| + |A_{22}|$, $r_2 = |A_{12}| + |A_{21}|$ and $r_{12} = |A_{21}| + |A_{22}|$. As mentioned earlier:

- $V_{11}$ represents the case where noise $V$ is the same as $\xi^1$, but $\xi^1$ is the same as $\xi^2$.

- $V_{12}$ represents the case where noise $V$ is different from $\xi^1$, but $\xi^1$ is the same as $\xi^2$.

- $V_{21}$ represents the case where noise $V$ is the same as $\xi^1$, but $\xi^1$ is different from $\xi^2$.

- $V_{22}$ represents the case where noise $V$ is different from $\xi^1$, but $\xi^1$ is different from $\xi^2$.

When all four components of the noise $V$ exist, each component has two possibilities: remaining unchanged or flipping. According to the HDAR update rule, there should be $2^4 = 16$ possible combinations. However, seven of these cases cannot exist due to contradictory inequality conditions. For example, when $V_{11}$, $V_{12}$, $V_{21}$, $V_{22}$ all remain unchanged, we have

$$1 \leq r_2 - r_1 \leq -1, \; N + 1 \leq r_2 + r_1 \leq N - 1. \tag{41}$$

Clearly, this system of inequalities is inconsistent, therefore such cases cannot exist. Here are seven impossible cases:

- $V_{11}(1) = A_{11}$, $V_{12}(1) = -A_{12}$, $V_{21}(1) = A_{21}$, $V_{22}(1) = -A_{22}$ (all unchanged).

- $V_{11}(1) = A_{11}$, $V_{12}(1) = -A_{12}$, $V_{21}(1) = A_{21}$ (unchanged). $V_{22}(1) = A_{22}$ (flipped).

- $V_{11}(1) = A_{11}$, $V_{12}(1) = -A_{12}$, $V_{22}(1) = -A_{22}$ (unchanged). $V_{21}(1) = -A_{21}$ (flipped).

- $V_{11}(1) = A_{11}$, $V_{12}(1) = -A_{12}$ (unchanged). $V_{21}(1) = -A_{21}$, $V_{22}(1) = A_{22}$ (flipped).

- $V_{11}(1) = A_{11}$, $V_{21}(1) = A_{21}$, $V_{22}(1) = -A_{22}$ (unchanged). $V_{12}(1) = A_{12}$ (flipped).

- $V_{12}(1) = -A_{12}$, $V_{21}(1) = A_{21}$, $V_{22}(1) = -A_{22}$ (unchanged). $V_{11}(1) = -A_{11}$ (flipped).

- $V_{21}(1) = A_{21}$, $V_{22}(1) = -A_{22}$ (unchanged). $V_{11}(1) = -A_{11}$, $V_{12}(1) = A_{12}$ (flipped).

This leaves us with the remaining nine cases.

**3.1 If $r_2 + r_1 \leq N - 1, r_2 - r_1 \geq 1$:**

$$V = \begin{bmatrix} V_{11} \\ V_{12} \\ V_{21} \\ V_{22} \end{bmatrix} = \begin{bmatrix} A_{11} \\ -A_{12} \\ A_{21} \\ -A_{22} \end{bmatrix} \rightarrow \begin{bmatrix} A_{11} \\ A_{12} \\ A_{21} \\ A_{22} \end{bmatrix} \rightarrow \begin{bmatrix} A_{11} \\ A_{12} \\ A_{21} \\ A_{22} \end{bmatrix}. \tag{42}$$

At this point:

- According to Equation 10: $V_{11}(1) = A_{11}$ (remains unchanged).

- According to Equation 12: $V_{12}(1) = A_{12}$ (flipped).

- According to Equation 13: $V_{21}(1) = A_{21}$ (remains unchanged).

- According to Equation 11: $V_{22}(1) = A_{22}$ (flipped).

In conclusion, $V(1) = \begin{bmatrix} A_{11} \\ A_{12} \\ A_{21} \\ A_{22} \end{bmatrix}$, where $r_1(1) = 0$ and $r_2(1) = |A_{21}| + |A_{22}|$.

- $r_2(1) + r_1(1) = |A_{21}| + |A_{22}| < N$, i.e., $r_2(1) + r_1(1) \leq N - 1$. According to Equation 10: $V_{11}(2) = A_{11}$, $V_{12}(2) = A_{12}$ (remain unchanged).

- $r_2(1) - r_1(1) = |A_{21}| + |A_{22}| > 0$, i.e., $r_2(1) - r_1(1) \geq 1$. According to Equation 12: $V_{21}(2) = A_{21}$, $V_{22}(2) = -A_{22}$ (remain unchanged).

Therefore, $V(2) = \begin{bmatrix} A_{11} \\ A_{12} \\ A_{21} \\ A_{22} \end{bmatrix} = V(1) = \xi^1$. The noise pattern $V$ will converge to $\xi^1$ after zero update step.

In fact, for $V(1)$: The flipping of $V_{12}$ and $V_{22}$ leads to the disappearance of these two parts. $A_{12}$ is incorporated into the $V_{11}$ part, and $A_{22}$ is incorporated into the $V_{21}$ part, this situation has effectively become the case described in Case 1.1. After one more update, the pattern remains unchanged.

**3.2 If $r_2 + r_1 \leq N - 1, r_2 - r_1 \leq -1$:**

$$V = \begin{bmatrix} V_{11} \\ V_{12} \\ V_{21} \\ V_{22} \end{bmatrix} = \begin{bmatrix} A_{11} \\ -A_{12} \\ A_{21} \\ -A_{22} \end{bmatrix} \rightarrow \begin{bmatrix} A_{11} \\ A_{12} \\ -A_{21} \\ -A_{22} \end{bmatrix} \rightarrow \begin{bmatrix} A_{11} \\ A_{12} \\ -A_{21} \\ -A_{22} \end{bmatrix}. \tag{43}$$

At this point:

- According to Equation 10:$V_{11}(1) = A_{11}$ (remains unchanged).

- According to Equation 12:$V_{12}(1) = A_{12}$ (flipped).

- According to Equation 13:$V_{21}(1) = -A_{21}$ (flipped).

- According to Equation 11:$V_{22}(1) = -A_{22}$ (remains unchanged).

In conclusion, $V(1) = \begin{bmatrix} A_{11} \\ A_{12} \\ -A_{21} \\ -A_{22} \end{bmatrix}$, where $r_1(1) = |A_{21}| + |A_{22}|$ and $r_2(1) = 0$.

- $r_2(1) + r_1(1) = |A_{21}| + |A_{22}| < N$, i.e., $r_2(1) + r_1(1) \leq N - 1$. According to Equation 10: $V_{11}(2) = A_{11}$, $V_{12}(2) = A_{12}$ (remain unchanged).

- $r_2(1) - r_1(1) = -(|A_{21}| + |A_{22}|) < 0$, i.e., $r_2(1) - r_1(1) \leq -1$.According to Equation 13: $V_{21}(2) = -A_{21}$, $V_{22}(2) = -A_{22}$ (remain unchanged).

Therefore, $V(2) = \begin{bmatrix} A_{11} \\ A_{12} \\ -A_{21} \\ -A_{22} \end{bmatrix} = V(1) = \xi^2$. The noise pattern $V$ will converge to $\xi^2$ after zero update step.

In fact, for $V(1)$: The flipping of $V_{12}$ and $V_{21}$ leads to the disappearance of these two parts. $A_{12}$ is incorporated into the $V_{11}$ part, and $A_{21}$ is incorporated into the $V_{22}$ part, this situation has effectively become the case described in Case 1.2. After one more update, the pattern remains unchanged.

**3.3 If $r_2 + r_1 \geq N + 1, r_2 - r_1 \leq -1$:**

$$V = \begin{bmatrix} V_{11} \\ V_{12} \\ V_{21} \\ V_{22} \end{bmatrix} = \begin{bmatrix} A_{11} \\ -A_{12} \\ A_{21} \\ -A_{22} \end{bmatrix} \rightarrow \begin{bmatrix} -A_{11} \\ -A_{12} \\ -A_{21} \\ -A_{22} \end{bmatrix} \rightarrow \begin{bmatrix} -A_{11} \\ -A_{12} \\ -A_{21} \\ -A_{22} \end{bmatrix}. \tag{44}$$

At this point:

- According to Equation 10:$V_{11}(1) = -A_{11}$ (flipped).

- According to Equation 12:$V_{12}(1) = -A_{12}$ (remains unchanged).

- According to Equation 13:$V_{21}(1) = -A_{21}$ (flipped).

- According to Equation 11:$V_{22}(1) = -A_{22}$ (remains unchanged).

In conclusion, $V(1) = \begin{bmatrix} -A_{11} \\ -A_{12} \\ -A_{21} \\ -A_{22} \end{bmatrix}$, where $r_1(1) = |A_{11}| + |A_{12}| + |A_{21}| + |A_{22}|$ and $r_2(1) = |A_{11}| + |A_{12}|$.

- $r_2(1)+r_1(1) = |A_{11}|+|A_{12}|+|A_{11}|+|A_{12}|+|A_{21}|+|A_{22}| = |A_{11}|+|A_{12}|+N \geq N+1$. According to Equation 11: $V_{11}(2) = -A_{11}$, $V_{12}(2) = -A_{12}$ (remain unchanged).

- $r_2(1)-r_1(1) = |A_{11}|+|A_{12}|-(|A_{11}|+|A_{12}|+|A_{21}|+|A_{22}|) = -(|A_{21}|+|A_{22}|) \leq -2 < -1$. According to Equation 13: $V_{21}(2) = -A_{21}$, $V_{22}(2) = -A_{22}$ (remain unchanged).

Therefore, $V(2) = \begin{bmatrix} -A_{11} \\ -A_{12} \\ -A_{21} \\ -A_{22} \end{bmatrix} = V(1) = \bar{\xi}^1$. The noise pattern $V$ will converge to $\bar{\xi}^1$ after zero update step.

In fact, for $V(1)$: The flipping of $V_{11}$ and $V_{21}$ leads to the disappearance of these two parts. $A_{11}$ is incorporated into the $V_{12}$ part, and $A_{21}$ is incorporated into the $V_{22}$ part, this situation has effectively become the case described in Case 1.4. After one more update, the pattern remains unchanged.

**3.4 If $r_2 + r_1 \geq N + 1, r_2 - r_1 \geq 1$:**

$$V = \begin{bmatrix} V_{11} \\ V_{12} \\ V_{21} \\ V_{22} \end{bmatrix} = \begin{bmatrix} A_{11} \\ -A_{12} \\ A_{21} \\ -A_{22} \end{bmatrix} \rightarrow \begin{bmatrix} -A_{11} \\ -A_{12} \\ A_{21} \\ A_{22} \end{bmatrix} \rightarrow \begin{bmatrix} -A_{11} \\ -A_{12} \\ A_{21} \\ A_{22} \end{bmatrix}. \tag{45}$$

At this point:

- According to Equation 10: $V_{11}(1) = -A_{11}$ (flipped).

- According to Equation 12: $V_{12}(1) = -A_{12}$ (remains unchanged).

- According to Equation 13: $V_{21}(1) = A_{21}$ (remains unchanged).

- According to Equation 11: $V_{22}(1) = A_{22}$ (flipped).

In conclusion, $V(1) = \begin{bmatrix} -A_{11} \\ -A_{12} \\ A_{21} \\ A_{22} \end{bmatrix}$, where $r_1(1) = |A_{11}|+|A_{12}|$ and $r_2(1) = |A_{11}|+|A_{12}|+|A_{21}|+|A_{22}|$.

- $r_2(1)+r_1(1) = |A_{11}|+|A_{12}|+|A_{21}|+|A_{22}|+|A_{11}|+|A_{12}| = N+|A_{11}|+|A_{12}| \geq N+1$. According to Equation 11: $V_{11}(2) = -A_{11}$, $V_{12}(2) = -A_{12}$ (remain unchanged).

- $r_2(1)-r_1(1) = |A_{11}|+|A_{12}|+|A_{21}|+|A_{22}|-(|A_{11}|+|A_{12}|) = |A_{21}|+|A_{22}| \geq 2 > 1$. According to Equation 12: $V_{21}(2) = A_{21}$, $V_{22}(2) = A_{22}$ (remain unchanged).

Therefore, $V(2) = \begin{bmatrix} -A_{11} \\ -A_{12} \\ A_{21} \\ A_{22} \end{bmatrix} = V(1) = \bar{\xi}^2$. The noise pattern $V$ will converge to $\bar{\xi}^2$ after zero update step.

In fact, for $V(1)$: The flipping of $V_{11}$ and $V_{22}$ leads to the disappearance of these two parts. $A_{11}$ is incorporated into the $V_{12}$ part, and $A_{22}$ is incorporated into the $V_{21}$ part, this situation has effectively become the case described in Case 1.3. After one more update, the pattern remains unchanged.

**3.5 If $N - 1 \leq r_2 + r_1 \leq N + 1, -1 < r_2 - r_1 < 1$, i.e. $r_1 = r_2 = N/2$:**

$$V = \begin{bmatrix} V_{11} \\ V_{12} \\ V_{21} \\ V_{22} \end{bmatrix} = \begin{bmatrix} A_{11} \\ -A_{12} \\ A_{21} \\ -A_{22} \end{bmatrix} \rightarrow \begin{bmatrix} -A_{11} \\ A_{12} \\ -A_{21} \\ A_{22} \end{bmatrix} \rightarrow \begin{bmatrix} A_{11} \\ -A_{12} \\ A_{21} \\ -A_{22} \end{bmatrix}. \tag{46}$$

At this point:

- According to Equation 10:$V_{11}(1) = -A_{11}$ (flipped).

- According to Equation 12:$V_{12}(1) = A_{12}$ (flipped).

- According to Equation 13:$V_{21}(1) = -A_{21}$ (flipped).

- According to Equation 11:$V_{22}(1) = A_{22}$ (flipped).

Thus, $V(1) = -V(0)$, where $r_1(1) = |A_{11}| + |A_{21}|$, $r_2(1) = |A_{11}| + |A_{22}|$. Then:

- $r_2(1) + r_1(1) = N$.

- $r_2(1) - r_1(1) = 0$.

At this point:

- According to Equation 10:$V_{11}(2) = A_{11}$ (flipped).

- According to Equation 12:$V_{12}(2) = -A_{12}$ (flipped).

- According to Equation 13:$V_{21}(2) = A_{21}$ (flipped).

- According to Equation 11:$V_{22}(2) = -A_{22}$ (flipped).

Therefore, $V(2) = -V(1) = V(0)$, and the noise pattern enters a symmetric-cycle between its original state and its negated state after zero update step.

**3.6 If $r_2 + r_1 \leq N - 1$, $-1 < r_2 - r_1 < 1$, i.e. $r_1 = r_2, |A_{21}| = |A_{22}|$:**

$$V = \begin{bmatrix} V_{11} \\ V_{12} \\ V_{21} \\ V_{22} \end{bmatrix} = \begin{bmatrix} A_{11} \\ -A_{12} \\ A_{21} \\ -A_{22} \end{bmatrix} \rightarrow \begin{bmatrix} A_{11} \\ A_{12} \\ -A_{21} \\ A_{22} \end{bmatrix} \rightarrow \begin{bmatrix} A_{11} \\ A_{12} \\ A_{21} \\ -A_{22} \end{bmatrix} \rightarrow \begin{bmatrix} A_{11} \\ A_{12} \\ -A_{21} \\ A_{22} \end{bmatrix}. \tag{47}$$

At this point:

- According to Equation 10:$V_{11}(1) = A_{11}$ (remains unchanged).

- According to Equation 12:$V_{12}(1) = A_{12}$ (flipped).

- According to Equation 13:$V_{21}(1) = -A_{21}$ (flipped).

- According to Equation 11:$V_{22}(1) = A_{22}$ (flipped).

Noted that $V(1) = \begin{bmatrix} A_{11} \\ A_{12} \\ -A_{21} \\ A_{22} \end{bmatrix}$, where $r_1(1) = |A_{21}|$ and $r_2(1) = |A_{22}|$.

- $r_2(1) + r_1(1) = |A_{22}| + |A_{21}| = N - (|A_{11}| + |A_{12}|) \leq N - 2 < N - 1$. According to Equation 10: $V_{11}(2) = A_{11}$, $V_{12}(2) = A_{12}$ (remain unchanged).

- $r_2(1) - r_1(1) = |A_{22}| - |A_{21}| = 0$, i.e., $-1 < r_2(1) - r_1(1) < 1$. According to Equation 12 and 13: $V_{21}(2) = A_{21}$, $V_{22}(2) = -A_{22}$ (flipped).

In conclusion, $V(2) = \begin{bmatrix} A_{11} \\ A_{12} \\ A_{21} \\ -A_{22} \end{bmatrix}$, where $r_1(2) = |A_{22}|$ and $r_2(2) = |A_{21}|$.

- $r_2(2) + r_1(2) = |A_{21}| + |A_{22}| = N - (|A_{11}| + |A_{12}|) \leq N - 2 < N - 1$. According to Equation 10: $V_{11}(3) = A_{11}$, $V_{12}(3) = A_{12}$ (remain unchanged).

- $r_2(2) - r_1(2) = |A_{21}| - |A_{22}| = 0$, i.e., $-1 < r_2(2) - r_1(2) < 1$. According to Equation 12 and 13: $V_{21}(3) = -A_{21}$, $V_{22}(3) = A_{22}$ (flipped).

Therefore, $V(3) = \begin{bmatrix} A_{11} \\ A_{12} \\ -A_{21} \\ A_{22} \end{bmatrix} = V(1)$. The noise pattern $V$ enters a hetero-cycle after one update step.

In fact, for $V(1)$: The flipping of $V_{12}$ leads to the disappearance of this part itself. $A_{12}$ is incorporated into the $V_{11}$ part, given that $|A_{21}| = |A_{22}|$, this situation has effectively become the case described in Case 2.2.1.

**3.7 If $r_2 + r_1 \geq N + 1, -1 < r_2 - r_1 < 1$, i.e. $r_1 = r_2, |A_{21}| = |A_{22}|$:**

$$ V = \begin{bmatrix} V_{11} \\ V_{12} \\ V_{21} \\ V_{22} \end{bmatrix} = \begin{bmatrix} A_{11} \\ -A_{12} \\ A_{21} \\ -A_{22} \end{bmatrix} \rightarrow \begin{bmatrix} -A_{11} \\ -A_{12} \\ -A_{21} \\ A_{22} \end{bmatrix} \rightarrow \begin{bmatrix} -A_{11} \\ -A_{12} \\ A_{21} \\ -A_{22} \end{bmatrix} \rightarrow \begin{bmatrix} -A_{11} \\ -A_{12} \\ -A_{21} \\ A_{22} \end{bmatrix}. \tag{48} $$

At this point:

- According to Equation 10: $V_{11}(1) = -A_{11}$ (flipped).

- According to Equation 12: $V_{12}(1) = -A_{12}$ (remains unchanged).

- According to Equation 13: $V_{21}(1) = -A_{21}$ (flipped).

- According to Equation 11: $V_{22}(1) = A_{22}$ (flipped).

Noted that, $V(1) = \begin{bmatrix} -A_{11} \\ -A_{12} \\ -A_{21} \\ A_{22} \end{bmatrix}$. where $r_1(1) = |A_{11}| + |A_{12}| + |A_{21}|$ and $r_2(1) = |A_{11}| + |A_{12}| + |A_{22}|$.

- $r_2(1) + r_1(1) = |A_{11}| + |A_{12}| + |A_{22}| + |A_{11}| + |A_{12}| + |A_{21}| = N + |A_{11}| + |A_{12}| \geq N + 2 > N + 1$. According to Equation 11: $V_{11}(2) = -A_{11}$, $V_{12}(2) = -A_{12}$ (remain unchanged).

- $r_2(1) - r_1(1) = |A_{22}| - |A_{21}| = 0$, i.e., $-1 < r_2(1) - r_1(1) < 1$. According to Equation 12 and 13: $V_{21}(2) = A_{21}$, $V_{22}(2) = -A_{22}$ (flipped).

In conclusion, $V(2) = \begin{bmatrix} -A_{11} \\ -A_{12} \\ A_{21} \\ -A_{22} \end{bmatrix}$, where $r_1(2) = |A_{11}| + |A_{12}| + |A_{22}|$ and $r_2(2) = |A_{11}| + |A_{12}| + |A_{21}|$.

- $r_2(2) + r_1(2) = |A_{11}| + |A_{12}| + |A_{21}| + |A_{11}| + |A_{12}| + |A_{22}| = N + |A_{11}| + |A_{12}| \geq N + 2 > N + 1$. According to Equation 11: $V_{11}(3) = -A_{11}$, $V_{12}(3) = -A_{12}$ (remain unchanged).

- $r_2(2) - r_1(2) = |A_{21}| - |A_{22}| = 0$, i.e., $-1 < r_2(2) - r_1(2) < 1$. According to Equation 12 and 13: $V_{21}(3) = -A_{21}$, $V_{22}(3) = A_{22}$ (flipped).

Therefore, $V(3) = \begin{bmatrix} -A_{11} \\ -A_{12} \\ -A_{21} \\ A_{22} \end{bmatrix} = V(1)$. The noise pattern $V$ enters a hetero-cycle after one update step.

**3.8 If $N - 1 < r_2 + r_1 < N + 1, r_2 - r_1 \geq 1$, i.e. $r_1 + r_2 = N, |A_{11}| = |A_{12}|$:**

$$V = \begin{bmatrix} V_{11} \\ V_{12} \\ V_{21} \\ V_{22} \end{bmatrix} = \begin{bmatrix} A_{11} \\ -A_{12} \\ A_{21} \\ -A_{22} \end{bmatrix} \rightarrow \begin{bmatrix} -A_{11} \\ A_{12} \\ A_{21} \\ A_{22} \end{bmatrix} \rightarrow \begin{bmatrix} A_{11} \\ -A_{12} \\ A_{21} \\ A_{22} \end{bmatrix} \rightarrow \begin{bmatrix} -A_{11} \\ A_{12} \\ A_{21} \\ A_{22} \end{bmatrix}. \tag{49}$$

At this point:

- According to Equation 10:$V_{11}(1) = -A_{11}$ (flipped).

- According to Equation 12:$V_{12}(1) = A_{12}$ (flipped).

- According to Equation 13:$V_{21}(1) = A_{21}$ (remains unchanged).

- According to Equation 11:$V_{22}(1) = A_{22}$ (flipped).

Noted that $V(1) = \begin{bmatrix} -A_{11} \\ A_{12} \\ A_{21} \\ A_{22} \end{bmatrix}$, where $r_1(1) = |A_{11}|$ and $r_2(1) = |A_{11}| + |A_{21}| + |A_{22}|$.

- $r_2(1) + r_1(1) = |A_{11}| + |A_{21}| + |A_{22}| + |A_{11}| = |A_{11}| + |A_{21}| + |A_{22}| + |A_{12}| = N$, i.e., $N - 1 < r_2(1) + r_1(1) < N + 1$. According to Equation 10 and 11: $V_{11}(2) = A_{11}$, $V_{12}(2) = -A_{12}$ (flipped).

- $r_2(1) - r_1(1) = |A_{21}| + |A_{22}| \geq 2 \geq 1$. According to Equation 12: $V_{21}(2) = A_{21}$, $V_{22}(2) = A_{22}$ (remain unchanged).

In conclusion, $V(2) = \begin{bmatrix} A_{11} \\ -A_{12} \\ A_{21} \\ A_{22} \end{bmatrix}$, where $r_1(2) = |A_{12}|$ and $r_2(2) = |A_{12}| + |A_{21}| + |A_{22}|$.

- $r_2(2) + r_1(2) = |A_{12}| + |A_{21}| + |A_{22}| + |A_{12}| = |A_{12}| + |A_{21}| + |A_{22}| + |A_{11}| = N$, i.e., $N - 1 < r_2(2) + r_1(2) < N + 1$. According to Equation 10 and 11: $V_{11}(3) = -A_{11}$, $V_{12}(3) = A_{12}$ (flipped).

- $r_2(2) - r_1(2) = |A_{21}| + |A_{22}| \geq 2 \geq 1$. According to Equation 12: $V_{21}(3) = A_{21}$, $V_{22}(3) = A_{22}$ (remain unchanged).

Therefore, $V(3) = \begin{bmatrix} A_{11} \\ -A_{12} \\ A_{21} \\ A_{22} \end{bmatrix} = V(1)$. The noise pattern $V$ enters a hetero-cycle after one update step.

In fact, for $V(1)$: The flipping of $V_{22}$ leads to the disappearance of this part itself. $A_{22}$ is incorporated into the $V_{21}$ part, given that $|A_{11}| = |A_{12}|$, this situation has effectively become the case described in Case 2.4.1.

**3.9 If $N - 1 < r_2 + r_1 < N + 1, r_2 - r_1 \leq -1$, i.e. $r_1 + r_2 = N, |A_{11}| = |A_{12}|$:**

$$V = \begin{bmatrix} V_{11} \\ V_{12} \\ V_{21} \\ V_{22} \end{bmatrix} = \begin{bmatrix} A_{11} \\ -A_{12} \\ A_{21} \\ -A_{22} \end{bmatrix} \rightarrow \begin{bmatrix} -A_{11} \\ A_{12} \\ -A_{21} \\ -A_{22} \end{bmatrix} \rightarrow \begin{bmatrix} A_{11} \\ -A_{12} \\ -A_{21} \\ -A_{22} \end{bmatrix} \rightarrow \begin{bmatrix} -A_{11} \\ A_{12} \\ -A_{21} \\ -A_{22} \end{bmatrix}. \tag{50}$$

At this point:

- According to Equation 10:$V_{11}(1) = -A_{11}$ (flipped).

- According to Equation 12:$V_{12}(1) = A_{12}$ (flipped).

- According to Equation 13:$V_{21}(1) = -A_{21}$ (flipped).

- According to Equation 11:$V_{22}(1) = -A_{22}$ (remains unchanged).

Noted that $V(1) = \begin{bmatrix} -A_{11} \\ A_{12} \\ -A_{21} \\ -A_{22} \end{bmatrix}$, where $r_1(1) = |A_{11}| + |A_{21}| + |A_{22}|$ and $r_2(1) = |A_{11}|$.

- $r_2(1) + r_1(1) = |A_{11}| + |A_{11}| + |A_{21}| + |A_{22}| = |A_{11}| + |A_{12}| + |A_{21}| + |A_{22}| = N$, i.e., $N - 1 < r_2(1) + r_1(1) < N + 1$. According to Equation 10 and 11: $V_{11}(2) = A_{11}$, $V_{12}(2) = -A_{12}$ (flipped).

- $r_2(1) - r_1(1) = -(|A_{21}| + |A_{22}|) \leq -2 \leq -1$. According to Equation 13: $V_{21}(2) = -A_{21}$, $V_{22}(2) = -A_{22}$ (remain unchanged).

In conclusion, $V(2) = \begin{bmatrix} A_{11} \\ -A_{12} \\ -A_{21} \\ -A_{22} \end{bmatrix}$, where $r_1(2) = |A_{12}| + |A_{21}| + |A_{22}|$ and $r_2(2) = |A_{12}|$.

- $r_2(2) + r_1(2) = |A_{12}| + |A_{12}| + |A_{21}| + |A_{22}| = |A_{11}| + |A_{12}| + |A_{21}| + |A_{22}| = N$, i.e., $N - 1 < r_2(2) + r_1(2) < N + 1$. According to Equation 10 and 11: $V_{11}(3) = -A_{11}$, $V_{12}(3) = A_{12}$ (flipped).

- $r_2(2) - r_1(2) = -(|A_{21}| + |A_{22}|) \leq -2 \leq -1$. According to Equation 13: $V_{21}(3) = -A_{21}$, $V_{22}(3) = -A_{22}$ (remain unchanged).

Therefore, $V(3) = \begin{bmatrix} -A_{11} \\ A_{12} \\ -A_{21} \\ -A_{22} \end{bmatrix} = V(1)$. The noise pattern enters a hetero-cycle after one update step.

In fact, for $V(1)$: The flipping of $V_{21}$ leads to the disappearance of this part itself. $A_{21}$ is incorporated into the $V_{22}$ part, given that $|A_{11}| = |A_{12}|$, this situation has effectively become the case described in Case 2.3.1.

From these results, we have identified three fundamental conditions required for the noise pattern to enter a cyclical regime. The complete analytical results are documented in Theorem 2. $\square$

A.4 PROOF OF COROLLARY 2

**Corollary A.2** (Evolution results with respect to $N$ and $r_{12}$). *These phenomena are governed by whether $N$ and $r_{12}$ are even or odd, leading to predictable cycle under HDAR update rule, as follows:*

1. *$N$ **even**, $r_{12}$ **even**: All cycle conditions are satisfied, which incurs that $V$ can evolve into any of the three cycleselfcycle, heterocycle, or symmetric-cycle. The quantity $C_{ee}$ is:*

$$
C_{ee} = 2 \left[ \binom{N - r_{12}}{\frac{N - r_{12}}{2}} \sum_{i = \frac{N - r_{12}}{2}}^{\frac{N-2}{2}} \binom{r_{12}}{i - \frac{N - r_{12}}{2}} + \binom{r_{12}}{\frac{r_{12}}{2}} \sum_{i = \frac{r_{12}}{2}}^{\frac{N-2}{2}} \binom{N - r_{12}}{i - \frac{r_{12}}{2}} \right] + \binom{r_{12}}{\frac{r_{12}}{2}} \binom{N - r_{12}}{\frac{N - r_{12}}{2}}.
$$

(51)

2. *$N$ **even**, $r_{12}$ **odd**: Any cycle conditions cannot be satisfied. Thus the quantity $C_{eo}$ is $0$.*

3. $N$ **odd**, $r_{12}$ **even**: *The cycle condition I is satisfied, which incurs self-cycle or hetero-cycle. The quantity $C_{\text{oe}}$ is:*

$$C_{\text{oe}} = 2\binom{r_{12}}{\frac{r_{12}}{2}} \sum_{i=\frac{r_{12}}{2}}^{\frac{N-1}{2}} \binom{N-r_{12}}{i-\frac{r_{12}}{2}}. \tag{52}$$

4. $N$ **odd**, $r_{12}$ **odd**: *The cycle condition II is satisfied, which incurs self-cycle or hetero-cycle. The quantity $C_{\text{oo}}$ is:*

$$C_{\text{oo}} = 2\binom{N-r_{12}}{\frac{N-r_{12}}{2}} \sum_{i=\frac{N-r_{12}}{2}}^{\frac{N-1}{2}} \binom{r_{12}}{i-\frac{N-r_{12}}{2}}. \tag{53}$$

*Proof.* **1. $N$ even, $r_{12}$ even.**

Explanation: The derivation approach for the first and second types of cycling is the same as described above, with the only difference being that the upper limit for $i$ can only go up to $\frac{N-2}{2}$. This is because when $N$ is even and $r_{12}$ is even, symmetric cycling exists, thus requiring the upper limit to be reduced by one. For the third type of cycling ( symmetric-cycle), the conditions $r_1 = r_2 = \frac{N}{2}$ must be satisfied. To achieve $r_1 = r_2$, we need $|V_{21}| = |V_{22}| = r_{12}/2$. This means that among the $r_{12}$ terms where $\xi^1$ and $\xi^2$ differ, exactly $r_{12}/2$ terms must be selected. Similarly, among the $N - r_{12}$ terms where $\xi^1$ and $\xi^2$ are identical, exactly $(N - r_{12})/2$ terms should be selected.

**2. $N$ even, $r_{12}$ odd.**

Explanation: It is obviously that $C_{\text{eo}} = 0$.

**3. $N$ odd, $r_{12}$ even.**

Explanation: We know that $r_1 = |V_{12}| + |V_{22}|$ and $r_2 = |V_{12}| + |V_{21}|$, while $|V_{21}| + |V_{22}| = r_{12}$. To enter the first type of circling, the condition $r_1 = r_2$ must be satisfied, which implies $|V_{21}| = |V_{22}| = r_{12}/2$. Therefore, if we use $i$ to represent the Hamming distance from the noise to $\xi^1$, namely $r_1$, it is clear that $i$ should start from $r_{12}/2$ and end at $\frac{N-1}{2}$. Regardless of the value of $i$, among the $r_{12}$ terms where $\xi^1$ and $\xi^2$ differ, $r_{12}/2$ terms should be selected. Among the $N - r_{12}$ terms where $\xi^1$ and $\xi^2$ are identical, $i - r_{12}/2$ terms should be selected. Due to the symmetry of the network, the final result multiplied by 2 gives the number of noise configurations that enter the circling state.

**4. $N$ odd, $r_{12}$ odd.**

Explanation: We know that $r_1 = |V_{12}| + |V_{22}|$ and $r_2 = |V_{12}| + |V_{21}|$, while $|V_{21}| + |V_{22}| = r_{12}$. To enter the second type of circling, the condition $r_1 + r_2 = N$ must be satisfied, which implies $|V_{11}| = |V_{12}| = (N - r_{12})/2$. Therefore, if we use $i$ to represent the Hamming distance from the noise to $\xi^1$, namely $r_1$, it is clear that $i$ should start from $(N - r_{12})/2$ and end at $(N - 1)/2$. Regardless of the value of $i$, among the $N - r_{12}$ terms where $\xi^1$ and $\xi^2$ are identical, $(N - r_{12})/2$ terms should be selected. Among the $r_{12}$ terms where $\xi^1$ and $\xi^2$ differ, $i - (N - r_{12})/2$ terms should be selected. Due to the symmetry of the network, the final result multiplied by 2 gives the number of noise configurations that enter the circling state. $\qquad\square$

A.5 PROOF OF COROLLARY 3

**Corollary A.3** (Bound of complete convergence domain). *If $r_{\text{ccd}}$ satisfies*

$$2r_{\text{ccd}} + 1 \leq r_{12} \leq N - (2r_{\text{ccd}} + 1) \quad \text{and} \quad N \geq 4r_{\text{ccd}} + 2, \tag{54}$$

*then any pattern $V$ with $r \leq r_{\text{ccd}}$ will converge to its nearest memory message in finite update steps.*

*Proof.* From the preparation work in Section 4.1 and Definition 3, we know that for a noise pattern to converge to a specific memory message, the components opposite to the memory message should be flipped while the identical components remain unchanged. Taking $\xi_1$ as an example, this requires

keeping $v_{11}$ and $v_{21}$ unchanged while flipping $v_{12}$ and $v_{22}$. Based on this condition, we obtain four inequality groups:

$$r_2 - r_1 \geq 1, \quad r_2 + r_1 \leq N - 1, \quad r_2 - r_1 > -1, \quad r_2 + r_1 < N + 1. \tag{55}$$

Taking the intersection yields:

$$r_2 - r_1 \geq 1, \quad r_2 + r_1 \leq N - 1. \tag{56}$$

Since

$$r_1 = |V_{12}| + |V_{22}|, \quad r_2 = |V_{12}| + |V_{21}|, \quad r_{12} = |V_{21}| + |V_{22}|. \tag{57}$$

The inequalities can then be transformed as follows:

$$\begin{aligned} r_2 - r_1 \geq 1 &\rightarrow |V_{21}| - |V_{22}| \geq 1 \\ &\rightarrow |V_{21}| + |V_{22}| \geq 1 + 2|V_{22}| \\ &\rightarrow r_{12} \geq 1 + 2|V_{22}|. \end{aligned} \tag{58}$$

$$\begin{aligned} r_2 + r_1 \leq N - 1 &\rightarrow |V_{21}| + |V_{22}| + 2|V_{12}| \leq N - 1 \\ &\rightarrow r_{12} \leq N - 1 - 2|V_{12}|. \end{aligned} \tag{59}$$

Taking the intersection gives:

$$1 + 2|V_{22}| \leq r_{12} \leq N - 1 - 2|V_{12}|. \tag{60}$$

To ensure all noise patterns with Hamming distance $r_1$ from $\xi_1$ converge to $\xi_1$, we consider the minimal range case:

- When all opposite components are in $|V_{22}|$, $r_{12}$ attains its maximal lower bound $1 + 2r_1$.

- When all opposite components are in $|V_{12}|$, $r_{12}$ attains its minimal upper bound $N - 1 - 2r_1$.

Thus we obtain the tightest convergence bound $r_1$, of which is $r_{\mathrm{ccd}}$:

$$1 + 2r_{\mathrm{ccd}} \leq r_{12} \leq N - 1 - 2r_{\mathrm{ccd}}. \tag{61}$$

Furthermore, this requires the system size constraint:

$$N \geq 4r_{\mathrm{ccd}} + 1. \tag{62}$$

From this, we can conclude that if $r_{\mathrm{ccd}}$ satisfies $1 + 2r_{\mathrm{ccd}} \leq r_{12} \leq N - 1 - 2r_{\mathrm{ccd}}, N \geq 4r_{\mathrm{ccd}} + 2$, then all noise patterns with a Hamming distance $r \leq r_{\mathrm{ccd}}$ from the target message will converge to it. Here, $r$ corresponds to the critical threshold $r_{\mathrm{ccd}}$ mentioned in the preceding Definition 4. $\qquad\square$

### A.6 PROOF OF COROLLARY 4

**Corollary A.4** (Relation between two bounds). *The minimum of $r_{\mathrm{ccd}}$ is 0, and $r_{\max} > 0$. When $N$ is even and $r_{12}$ is even: $r_{\max} = \frac{N}{2} - 2$. For all other cases: $r_{\max} = \lfloor \frac{N}{2} \rfloor - 1$. The relation between these two bounds satisfies*

$$r_{\mathrm{ccd}} < r_{\max}. \tag{63}$$

*Proof.* Let's prove to obtain the bounds of $r$ firstly. According to Equation 60, we can deduce that for the noise pattern $V$ to converge to $\xi^1$, it must at least satisfy:

$$\begin{aligned} 1 + 2|V_{22}| \leq N - 1 - 2|V_{12}| &\rightarrow 2|V_{12}| + 2|V_{22}| + 2 \leq N \\ &\rightarrow 2r_1 + 2 \leq N \\ &\rightarrow r_1 \leq \frac{N}{2} - 1. \end{aligned} \tag{64}$$

Only when $r_1 \leq \frac{N}{2} - 1$ is satisfied can the noise pattern $V$ potentially converge to $\xi^1$. Considering the case where $N$ is odd, we have: $r_1 \leq \lfloor \frac{N}{2} \rfloor - 1$. However, there is an exception: when $N$ is even and $r_{12}$ is even, the noise pattern $V$ can only converge to $\xi^1$ if $r_1 \leq \frac{N}{2} - 2$. The proof is as follows:

From the above Equations 58 and 59,

$$|V_{21}| - |V_{22}| \geq 1, \quad |V_{21}| + |V_{22}| + 2|V_{12}| \leq N - 1. \tag{65}$$

Assuming $|V_{21}| - |V_{22}| = 1$, then $|V_{21}| = |V_{22}| + 1$, and $r_{12} = |V_{21}| + |V_{22}| = 2|V_{22}| + 1$. In this case, $r_{12}$ must be odd, which contradicts the assumption. Therefore,

$$|V_{21}| - |V_{22}| > 1 \rightarrow |V_{21}| - |V_{22}| \geq 2$$
$$\rightarrow r_{12} \geq 2 + 2|V_{22}|. \tag{66}$$

Similarly, assuming $|V_{21}| + |V_{22}| + 2|V_{12}| = N - 1$, then $r_{12} = |V_{21}| + |V_{22}| = N - 1 - 2|V_{12}|$. Here, $r_{12}$ is also odd, which contradicts the assumption. Thus,

$$|V_{21}| + |V_{22}| + 2|V_{12}| < N - 1 \rightarrow |V_{21}| + |V_{22}| + 2|V_{12}| \leq N - 2$$
$$\rightarrow r_{12} \leq N - 2 - 2|V_{12}|. \tag{67}$$

In conclusion,

$$2 + 2|V_{22}| \leq r_{12} \leq N - 2 - 2|V_{12}|. \tag{68}$$

Using a similar proof method, we can conclude that in this case, the condition $r_1 \leq \frac{N}{2} - 2$ must be satisfied. Therefore,

- When $N$ is even and $r_{12}$ is even, the noise pattern can only converge to the message if the Hamming distance $r$ satisfies $r \leq \frac{N}{2} - 2$.

- In all other cases, the condition $r \leq \lfloor \frac{N}{2} \rfloor - 1$ must be satisfied.

Here, $r$ corresponds to the $r_{\max}$ mentioned in our previous Definition 4.That is:

When $N$ is even and $r_{12}$ is even:

$$r_{\max} = \frac{N}{2} - 2. \tag{69}$$

For all other case:

$$r_{\max} = \left\lfloor \frac{N}{2} \right\rfloor - 1. \tag{70}$$

According to the definition of CD and CCD, the memory message $\xi$ converge to itself. Thus the minimum of $r_{ccd}$ is 0, and $r_{\max} > 0$. To prove $r_{ccd} < r_{\max}$, we enumerate all the situation of $r_{ccd}$ as follows:

**(1) When $r_{ccd} = 0$:**

Since the convergence domain $r_{\max}$ is clearly greater than 0, we have $r_{ccd} < r_{\max}$.

**(2) When $r_{ccd} = 1$:**

According to Corollary A.3, the number of neurons $N$ must satisfy

$$N \geq 6, \quad 3 \leq r_{12} \leq N - 3. \tag{71}$$

For the case $N = 6$, we should have $r_{12} = 3$. Then according to Equation 70:

$$r_{\max} = \left\lfloor \frac{N}{2} \right\rfloor - 1 = 2 > r_{ccd} = 1. \tag{72}$$

For cases when $N > 6$:

- If $N$ is even and $r_{12}$ is even, according to Equation 69:

$$r_{\max} = \frac{N}{2} - 2 > 1 > r_{ccd} = 1. \tag{73}$$

- For other combinations of parity for $N$ and $r_{12}$, according to Equation 70:

$$r_{\max} = \left\lfloor \frac{N}{2} \right\rfloor - 1 > 2 > r_{\mathrm{ccd}} = 1. \tag{74}$$

Therefore, when the complete convergence domain $r_{\mathrm{ccd}}$ is 1, we have $r_{\mathrm{ccd}} < r_{\max}$.

**(3) When $r_{\mathrm{ccd}} \geq 2$:**

According to Equation A.5, the number of neurons $N$ must satisfy:

$$N \geq 4r_{\mathrm{ccd}} + 2, \quad 1 + 2r_{\mathrm{ccd}} \leq r_{12} \leq N - 1 - 2r_{\mathrm{ccd}}. \tag{75}$$

For cases when $N$ is even and $r_{12}$ is even, according to Equation 69:

$$r_{\max} = \frac{N}{2} - 2 \geq 2r_{\mathrm{ccd}} - 1 \geq r_{\mathrm{ccd}} + 1 > r_{\mathrm{ccd}}. \tag{76}$$

For other cases:

$$r_{\max} = \left\lfloor \frac{N}{2} \right\rfloor - 1 \geq 2r_{\mathrm{ccd}} > r_{\mathrm{ccd}}. \tag{77}$$

Therefore, when the complete convergence domain $r_{\mathrm{ccd}}$ is any number greater than or equal to 2, we have $r_{\mathrm{ccd}} < r_{\max}$.

In conclusion, for any complete convergence domain $r_{\mathrm{ccd}}$, the inequality $r_{\mathrm{ccd}} < r_{\max}$ always holds true. $\qquad\square$

## B    MORE EXPERIMENTS

### B.1    MEMORIZING TWO MESSAGES WITH FIXED $r_{12}$ RATIO

We fix the proportional relationship between the Hamming distance of $\xi^1$ and $\xi^2$ and the number of neurons as $\frac{r_{12}}{N} = m$, where $m$ is a constant.

For small-scale networks ($N \leq 10$), we perform exhaustive enumeration. Two $N$-bit patterns with Hamming distance $r_{12}$ are randomly generated, and the corresponding weight matrix is constructed. All $2^N$ possible noise patterns are evolved under the HDAR update rule, and the proportion of patterns entering cyclic trajectories is recorded. In all trials, the empirical cycle ratios match the theoretical values exactly.

For large-scale networks ($N \geq 99$), we adopt Monte Carlo sampling due to the infeasibility of exhaustive search. For each of 20 independent trials, a new pair of messages (with fixed $r_{12}$) is generated, along with 10,000 randomly sampled noise vectors (each bit independently drawn from $\pm 1$ with equal probability). Each noise sample is iterated under the HDAR update rule, and its asymptotic behavior is recorded. Across all trials, the observed cycle ratios consistently fall within theoretical error bounds, supporting the validity of our convergence analysis.

To ensure statistical robustness, we repeat each experiment 50 times under fixed $N$ and $r_{12}$. In each repetition:

* Memory messages are newly generated.
* A fresh set of noise samples is generated.

Final results are averaged over the 50 repetitions. Across all settings, empirical deviations from theoretical predictions are negligible, confirming the accuracy of our derived proportion formula. Detailed results for the case $r_{12}/N = 1/3$ are presented in Table 4.

Table 4: Comparison between theoretical and experimental values under the same $r_{12}$ ratio.

| $N$(ODD) | $r_{12}$ | THEORY | EXPERIMENT | $N$(EVEN) | $r_{12}$ | THEORY | EXPERIMENT |
|---|---|---|---|---|---|---|---|
| 99 | 33 | 0.0978 | 0.0977 | 100 | 33 | 0 | 0 |
| 199 | 66 | 0.0978 | 0.0976 | 200 | 66 | 0.1600 | 0.1597 |
| 499 | 166 | 0.0618 | 0.0620 | 500 | 166 | 0.1028 | 0.1023 |
| 999 | 333 | 0.0309 | 0.0310 | 1000 | 333 | 0 | 0 |

### B.2    MEMORIZING TWO MESSAGES WITH FIXED $r_{12} = 30$

We randomly select two $N$-bit binary vectors as memory messages, ensuring a fixed Hamming distance $r_{12}$ between them (with $N > r_{12}$). Due to the infeasibility of exhaustive noise enumeration, we adopt Monte Carlo sampling, using the same noise generation procedure as in the previous experiment.

For each fixed $N$, we construct a weight matrix from the selected memory messages and evolve randomly generated noise samples under the HDAR update rule, recording whether each pattern converges or enters a cycle. This process is repeated 50 times, with both the memory messages and noise samples regenerated in each trial. The average cycle ratio is computed over all repetitions and compared against theoretical predictions.

To investigate how network size influences dynamics, we fix $r_{12} = 30$ and vary $N$, conducting experiments across multiple odd values of $N$. As shown in Table 5, the empirical results reveal a striking phenomenon: the cycle ratio remains constant across all odd values of $N$, aligning precisely with theoretical expectations.

Based on experimental observations, we conclude that for any fixed even value of $r_{12}$, the cycle ratio remains invariant across all odd values of $N$, and is precisely given by $\frac{\binom{r_{12}}{r_{12}/2}}{2^{r_{12}}}$. This implies that the cycle ratio is solely determined by the even parameter $r_{12}$, independent of the network size.

Table 5: Comparison between theoretical and experimental values under $r_{12} = 30$.

| $N$(odd) | $r_{12}$ | Theory | Experiment | | $N$(even) | $r_{12}$ | Theory | Experiment |
|----------|----------|--------|------------|---|-----------|----------|--------|------------|
| 99 | 30 | 0.1445 | 0.1445 | | 100 | 30 | 0.2258 | 0.2258 |
| 199 | 30 | 0.1445 | 0.1440 | | 200 | 30 | 0.1967 | 0.1970 |
| 499 | 30 | 0.1445 | 0.1452 | | 500 | 30 | 0.1759 | 0.1762 |
| 999 | 30 | 0.1445 | 0.1441 | | 1000 | 30 | 0.1664 | 0.1668 |

The derivation is presented below, based on Corollary 2, under the condition that $N$ is odd and $r_{12}$ is even.

$$
C_{\text{oe}} = 2\binom{r_{12}}{r_{12}/2} \sum_{i=r_{12}/2}^{\frac{N-1}{2}} \binom{N-r_{12}}{i-r_{12}/2}
$$

$$
= 2\binom{r_{12}}{r_{12}/2} \sum_{i=0}^{\frac{N-1-r_{12}}{2}} \binom{N-r_{12}}{i}. \tag{78}
$$

From the binomial theorem $(a+b)^n = \sum_{k=0}^{n} \binom{n}{k} a^k b^{n-k}$, when $a = b = 1$:

$$
2^n = \sum_{k=0}^{n} \binom{n}{k}. \tag{79}
$$

Using the symmetry property of binomial coefficients $\binom{n}{k} = \binom{n}{n-k}$ and for odd $n$:

$$
2^n = 2\sum_{k=0}^{\frac{n-1}{2}} \binom{n}{k} \Rightarrow \sum_{k=0}^{\frac{n-1}{2}} \binom{n}{k} = 2^{n-1}. \tag{80}
$$

Since $N$ is odd and $r_{12}$ is even ($N - r_{12}$ is odd):

$$
C_{\text{oe}} = 2\binom{r_{12}}{r_{12}/2} \sum_{i=0}^{\frac{N-r_{12}-1}{2}} \binom{N-r_{12}}{i}
$$

$$
= 2\binom{r_{12}}{r_{12}/2} \sum_{k=0}^{\frac{N-r_{12}-1}{2}} \binom{N-r_{12}}{k}
$$

$$
= 2\binom{r_{12}}{r_{12}/2} 2^{N-r_{12}-1}. \tag{81}
$$

Thus the cycling ratio becomes:

$$
\frac{2\binom{r_{12}}{r_{12}/2} 2^{N-r_{12}-1}}{2^N} = \frac{\binom{r_{12}}{r_{12}/2}}{2^{r_{12}}}. \tag{82}
$$

Similarly, we can derive another corollary: when $r_{12}$ is any even number and we add any odd number to it to form $N$ (set $N$ odd), the resulting cycling ratio remains identical to the original case with

even $r_{12}$. The derivation proceeds as follows:

$$C_{\text{oo}} = 2 \begin{pmatrix} N - r_{12} \\ (N - r_{12})/2 \end{pmatrix} \sum_{i=\frac{N-r_{12}}{2}}^{\frac{N-1}{2}} \begin{pmatrix} r_{12} \\ i - \frac{N-r_{12}}{2} \end{pmatrix}$$

$$= 2 \begin{pmatrix} N - r_{12} \\ (N - r_{12})/2 \end{pmatrix} \sum_{i=0}^{\frac{r_{12}-1}{2}} \begin{pmatrix} r_{12} \\ i \end{pmatrix}$$

$$= 2 \begin{pmatrix} N - r_{12} \\ (N - r_{12})/2 \end{pmatrix} 2^{r_{12}-1}$$

$$= \begin{pmatrix} N - r_{12} \\ (N - r_{12})/2 \end{pmatrix} 2^{r_{12}}. \tag{83}$$

Thus, the cycling ratio becomes:

$$\frac{\begin{pmatrix} N-r_{12} \\ (N-r_{12})/2 \end{pmatrix} 2^{r_{12}}}{2^N} = \frac{\begin{pmatrix} N-r_{12} \\ (N-r_{12})/2 \end{pmatrix}}{2^{N-r_{12}}}. \tag{84}$$

By comparing Equations 82 and 84, we observe that: for an odd $r_{12}^{(\text{odd})}$, an even $r_{12}^{(\text{even})}$, and odd $N$, if they satisfy

$$N - r_{12}^{(\text{odd})} = r_{12}^{(\text{even})}, \tag{85}$$

then the cycle ratios under both conditions will be identical.

### B.3 STATISTICAL ANALYSES

We verify our experimental results using statistical error testing methods. Below are some statistics of essential calculation formulas:

Formula 1:

$$s = \sqrt{\frac{\sum_{i=1}^n (x_i - \bar{x})^2}{n - 1}}. \tag{86}$$

This formula calculates the sample standard deviation of experimental values, where:

- $x_i$ represents the $i$-th experimental value.
- $n$ represents the sample size.
- $\bar{x}$ represents the mean of $n$ experimental values.

Formula 2:

$$\text{SE} = \frac{s}{\sqrt{n}}. \tag{87}$$

This formula calculates the standard error of experimental values, where:

- $s$ represents the sample standard deviation of experimental values.
- $n$ represents the sample size.

Formula 3:

$$Z = \frac{\bar{x} - x_{\text{t}}}{\text{SE}}. \tag{88}$$

This formula calculates the standardized error (Z-score) of experimental values, where:

- $\bar{x}$ represents the mean of n experimental values.

- $x_t$ represents the theoretical value for the experiment.
- SE represents the standard error of experimental values.

Formula 4:

$$\text{d} = \frac{|\bar{x} - x_t|}{\sigma}. \tag{89}$$

This formula calculates Cohen's d effect size for experimental values, where:

- $\bar{x}$ represents the mean of $n$ experimental values.
- $x_t$ represents the theoretical value for the experiment.
- $\sigma$ represents the population standard deviation of experimental values.
- Effect size interpretation:
  - $\text{d} < 0.2$: negligible difference.
  - $0.2 \leq \text{d} < 0.5$: small difference.
  - $0.5 \leq \text{d} < 0.8$: medium difference.
  - $\text{d} \geq 0.8$: large difference.

The original definition of Cohen's d uses population standard deviation ($\sigma$). However, since sample standard deviation $s$ is an unbiased estimator of $\sigma$ (especially when n is large), and in practice $\sigma$ is usually unknown, we typically substitute it with sample standard deviation $s$. In our experiments, since we cannot exhaust all possible noise patterns, we also use sample standard deviation $s$ instead of population standard deviation ($\sigma$). Therefore, the practical version of Formula 4 becomes:

$$\text{d} = \frac{|\bar{x} - x_t|}{s}$$

Taking $N = 9$, $r_{12} = 3$ in Table 2as an example for analysis:

- Null hypothesis $H_0$: No significant difference between experimental observations and theoretical values, i.e., observed deviations are caused by random fluctuations.
- Alternative hypothesis $H_1$: Significant difference exists between experimental and theoretical values (possibly systematic bias).

Calculated results:

- Sample standard deviation: 0.0038.
- Standard error: 0.0005.
- Standardized error Z: 1.0802.

Since $|\text{Z}| \approx 1.0802 < 1.96 = \text{Z}_{\alpha=0.05}$, the difference is not statistically significant. We therefore fail to reject the null hypothesis ($H_0$): the observed discrepancy between experimental and theoretical values can be attributed to random fluctuations, with no evidence of a meaningful deviation.

To assess the practical relevance of this difference, we further compute Cohen's d effect size. The obtained Cohen's d is approximately $0.1817 < 0.2$, indicates a negligible effect, confirming that the deviation is not only statistically insignificant but also practically minor.

Table 6 reports the standardized errors and Cohen's $d$ values derived from the experimental results in Table 1, based on the procedure outlined above.

Table 7 below presents the standardized errors and Cohen's d effect sizes derived from the experimental data for odd values of $N$ in Table 2 , calculated using the aforementioned methodology.

Table 8 below presents the standardized errors and Cohen's d effect sizes derived from the experimental data for odd values of $N$ in Table 3, calculated using the aforementioned methodology.

Table 9 below presents the standardized errors and Cohen's d effect sizes derived from the experimental data for even values of $N$ in Table 3, calculated using the aforementioned methodology.

Table 6: Comparison between theoretical and experimental values under even $N$ when memorizing single memory message.

| $N$(EVEN) | THEORY | EXPERIMENT | Z | Cohen's d |
|---|---|---|---|---|
| 100 | 0.0796 | 0.0795 | -0.1703 | 0.0239 |
| 200 | 0.0563 | 0.0568 | 1.1872 | 0.1662 |
| 500 | 0.0357 | 0.0358 | 0.6359 | 0.0890 |
| 1000 | 0.0252 | 0.0250 | -1.1071 | 0.1550 |

Table 7: Comparison between theoretical and experimental values when memorizing two memory messages with their Hamming distance $r_{12}$ is odd 3.

| $N$(ODD) | $r_{12}$ | THEORY | EXPERIMENT | Z | Cohen's d |
|---|---|---|---|---|---|
| 99 | 3 | 0.0812 | 0.0820 | 0.3405 | 0.0477 |
| 199 | 3 | 0.0569 | 0.0557 | 0.8112 | 0.1137 |
| 499 | 3 | 0.0358 | 0.0359 | -1.3206 | 0.1849 |
| 999 | 3 | 0.0253 | 0.0252 | -0.3945 | 0.0552 |

Table 8: Comparison between theoretical and experimental values when memorizing two memory messages with their Hamming distance $r_{12}$ is even 4.

| $N$(ODD) | $r_{12}$ | THEORY | EXPERIMENT | Z | Cohen's d |
|---|---|---|---|---|---|
| 99 | 4 | 0.375 | 0.3751 | 0.1619 | 0.0223 |
| 199 | 4 | 0.375 | 0.3753 | 0.3576 | 0.0501 |
| 499 | 4 | 0.375 | 0.3750 | -0.0332 | 0.0046 |
| 999 | 4 | 0.375 | 0.3752 | 0.4294 | 0.0601 |

Table 9: Comparison between theoretical and experimental values when memorizing two memory messages with their Hamming distance $r_{12}$ is even 4.

| $N$(EVEN) | $r_{12}$ | THEORY | EXPERIMENT | Z | Cohen's d |
|---|---|---|---|---|---|
| 100 | 4 | 0.4258 | 0.4247 | -1.5549 | 0.2177 |
| 200 | 4 | 0.4106 | 0.4095 | -1.8358 | 0.2571 |
| 500 | 4 | 0.3974 | 0.3971 | -0.3403 | 0.0476 |
| 1000 | 4 | 0.3908 | 0.3919 | 0.6429 | 0.0900 |

Table 10: Comparison between theoretical and experimental values with different odd $N$ under the same $r_{12}$ ratio.

| $N$(ODD) | $r_{12}$ | THEORY | EXPERIMENT | Z | Cohen's d |
|---|---|---|---|---|---|
| 99 | 33 | 0.0978 | 0.0977 | -0.4832 | 0.0677 |
| 199 | 66 | 0.0978 | 0.0976 | -0.6299 | 0.0882 |
| 499 | 166 | 0.0618 | 0.0620 | 0.3571 | 0.0500 |
| 999 | 333 | 0.0309 | 0.0310 | 0.5085 | 0.0712 |

Table 10 below presents the standardized errors and Cohen's d effect sizes derived from the experimental data for odd values of $N$ in Table 4, calculated using the aforementioned methodology.

Table 11 below presents the standardized errors and Cohen's d effect sizes derived from the experimental data for even values of $N$ in Table 4, calculated using the aforementioned methodology.

Table 12 below presents the standardized errors and Cohen's d effect sizes derived from the experimental data for odd values of $N$ in Table 5, calculated using the aforementioned methodology.

Table 11: Comparison between theoretical and experimental values with different even $N$ under the same $r_{12}$ ratio.

| $N$(EVEN) | $r_{12}$ | THEORY | EXPERIMENT | Z | Cohen's d |
|---|---|---|---|---|---|
| 100 | 33 | 0 | 0 | None | None |
| 200 | 66 | 0.1600 | 0.1597 | -0.4243 | 0.0594 |
| 500 | 166 | 0.1028 | 0.1023 | -1.1737 | 0.1643 |
| 1000 | 333 | 0 | 0 | None | None |

Table 12: Comparison between theoretical and experimental values with different odd $N$ under even $r_{12} = 30$.

| $N$(ODD) | $r_{12}$ | THEORY | EXPERIMENT | Z | Cohen's d |
|---|---|---|---|---|---|
| 99 | 30 | 0.1445 | 0.1445 | 0.0179 | 0.0025 |
| 199 | 30 | 0.1445 | 0.1440 | -0.8602 | 0.1205 |
| 499 | 30 | 0.1445 | 0.1452 | 1.3036 | 0.1825 |
| 999 | 30 | 0.1445 | 0.1441 | -0.8564 | 0.1199 |

Table 13 below presents the standardized errors and Cohen's d effect sizes derived from the experimental data for even values of $N$ in Table 5, calculated using the aforementioned methodology.

Table 13: Comparison between theoretical and experimental values with different even $N$ under even $r_{12} = 30$.

| $N$(EVEN) | $r_{12}$ | THEORY | EXPERIMENT | Z | Cohen's d |
|---|---|---|---|---|---|
| 100 | 30 | 0.2258 | 0.2258 | 0.1191 | 0.0167 |
| 200 | 30 | 0.1967 | 0.1970 | 0.5556 | 0.0779 |
| 500 | 30 | 0.1759 | 0.1762 | 0.4179 | 0.0585 |
| 1000 | 30 | 0.1664 | 0.1668 | 0.7757 | 0.1086 |

From the data presented in Table 6 to Table 13, all standardized errors Z) fall below the critical value of $1.96$, indicating no statistically significant differences between experimental and theoretical results. Moreover, the majority of Cohen's d values are below $0.2$, suggesting that the observed differences are practically negligible. A few instances with $d > 0.2$ are expected and not unusual.

These slight deviations can be attributed to sampling variability and measurement noise. Under the null hypothesis ($H_0$), the presence of standard error due to finite sample sizes leads to natural dispersion in the sampling distribution of the mean difference. Even in the absence of a true effect, Cohen's d may occasionally exceed conventional thresholds (e.g., $d \approx 0.2$) due to random fluctuationespecially in low-power settings with limited samples or high variance. This phenomenon is consistent with asymptotic statistical theory: across repeated trials, such deviations will occur with probability $\alpha$ (Type I error rate), even without underlying effects.

In addition, we have converted the tabular data in the main text into corresponding line charts for clearer visualization. The left panel shows results for small-scale networks ($N \leq 20$), where the theoretical and experimental values coincide perfectly since exhaustive enumeration was applied. The right panel displays results for larger networks ($N > 20$), where Monte Carlo sampling was employed. As illustrated, the theoretical and experimental values remain closely aligned, thereby validating the correctness of our theoretical analysis.

The line chart of the data on the right side of Table 1 is shown in Figure 3.

The line chart of the data on the left side of Table 2 is shown in Figure 4.

The line chart of the data on the left side of Table 3 is shown in Figure 5.

The line chart of the data on the right side of Table 3 is shown in Figure 6.

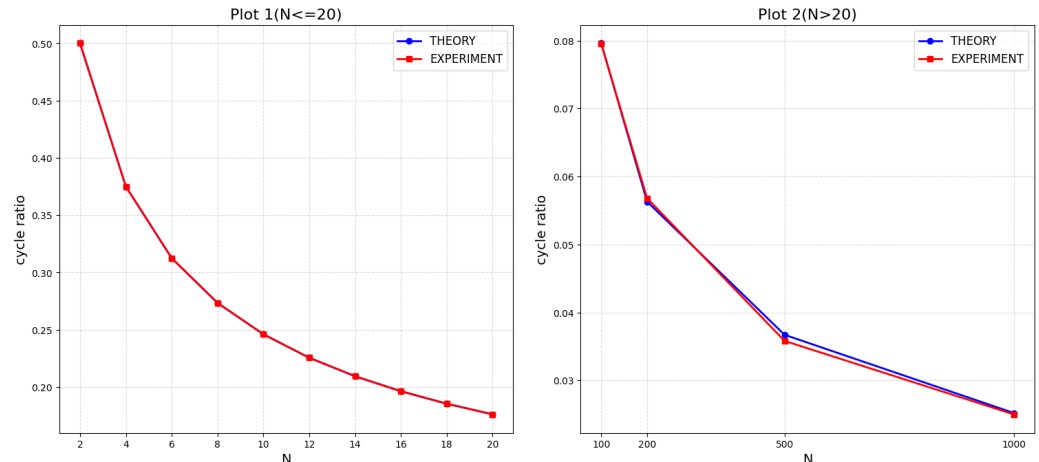

Figure 3: Line chart presentation of the data on the left side of Table 1.The panel on the left displays the cases where $N \leq 20$, while that on the right shows the scenarios when $N > 20$.

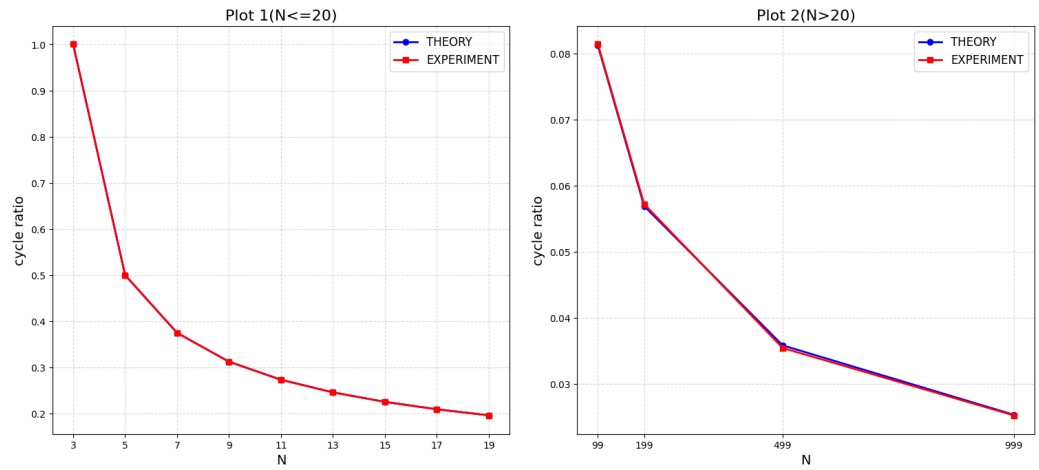

Figure 4: Line chart presentation of the data on the right side of Table 2.The panel on the left displays the cases where $N \leq 20$, while that on the right shows the scenarios when $N > 20$.

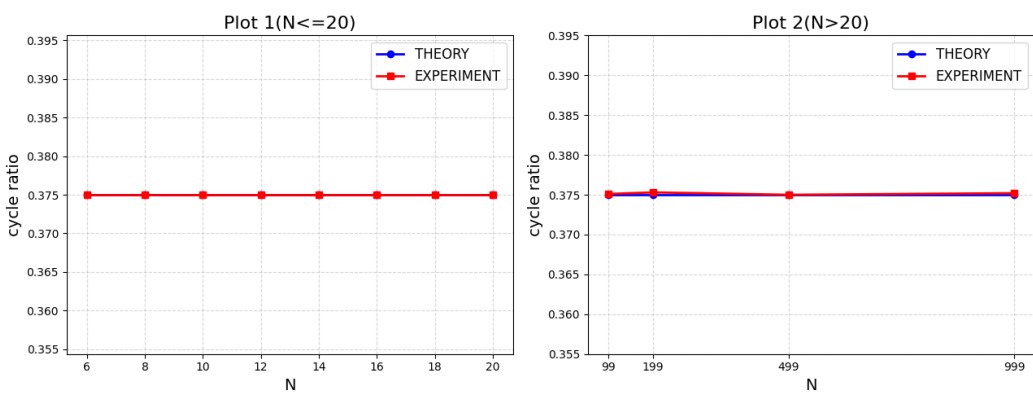

41

Figure 5: Line chart presentation of the data on the left side of Table 3.The panel on the left displays the cases where $N \leq 20$, while that on the right shows the scenarios when $N > 20$.

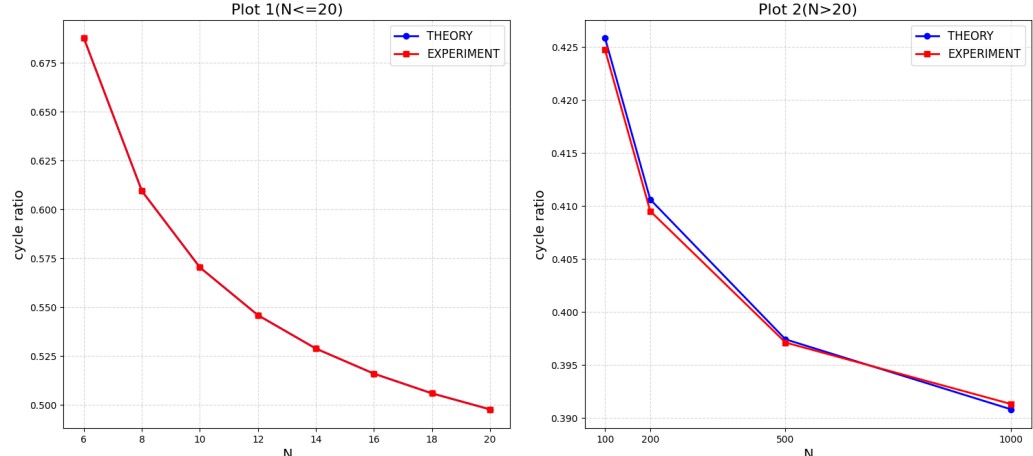

Figure 6: Line chart presentation of the data on the right side of Table 3.The panel on the left displays the cases where $N \leq 20$, while that on the right shows the scenarios when $N > 20$.

### B.4 VISUAL PRESENTATION

To facilitate understanding of the content and conclusions in this paper, the dynamic behavior of the Hopfield Neural Network will be demonstrated through specific examples and visual illustrations.

### B.4.1 MEMORIZING ONE MESSAGE

When memorizing one message, the convergence of the Hopfield network is only related to the parity of $N$. If $N$ is odd, there is no symmetric-cycle, and there are no cyan nodes in the middle part of Figure 1. If $N$ is even, the symmetric-cycle occurs. The following two examples demonstrate the relationship between the dynamic behavior of the network and the Hamming distance when $N$ is even and odd.

**Case 1: $N$ even.**

For example, when the number of neurons, $N = 4$, and one message $\begin{bmatrix} -1 & 1 & 1 & -1 \end{bmatrix}^{\mathrm{T}}$ is memorized (converted to decimal as the number 6), its symmetric message is 9. According to Case1 in Theorem 1, , we can conclude that the noise patterns with a Hamming distance less than 2 (i.e., a Hamming distance of 1 from 6) will converge to 6, while the noise patterns with a Hamming distance less than 2 from 9 will converge to 9. Noise patterns with a Hamming distance equal to 2 from both 6 and 9 will fall into a symmetric-cycle.

The visual simulation of the experiment is shown in Figure 7. As seen, the central circle in the left side of the diagram represents the memorized message, and the central circle on the right represents the symmetric message. Starting from the center, each expanding ring indicates an increase in the Hamming distance of the noise pattern from the message in central circle. The red arrows start from noise patterns and point to the stable fixed points (the memorized message), while the green arrows start from noise patterns and point to the stable fixed points (the symmetric message). The blue arrows represent noise patterns that fall into symmetric-cycle.

It is clear that all noise patterns in the first ring (Hamming distance of 1) around the memorized message 6 converge to 6, and similarly, the noise patterns in the first ring around the symmetric message 9 converge to 9. Noise patterns with a Hamming distance of 2 from both 6 and 9 fall into symmetric-cycle. The experimental simulation is fully consistent with the theoretical derivation.

Additionally, the new symmetries in the noise pattern landscape are reflected in the following aspects:

- Two noise patterns trapped in symmetric-cycle are mutually symmetric(for instance, 3 and 12, 5 and 10, etc.).

- Noise patterns converging to message 6 are $(2, 4, 7, 14)$, while those converging to message 9 are $(13, 11, 8, 1)$. The attraction basins of the memory message and its symmetric counterpart are identical in size (both of size 5, including the messages themselves), and the corresponding patterns are mutually symmetric (for instance, 2 and 13, 4 and 11, etc.).

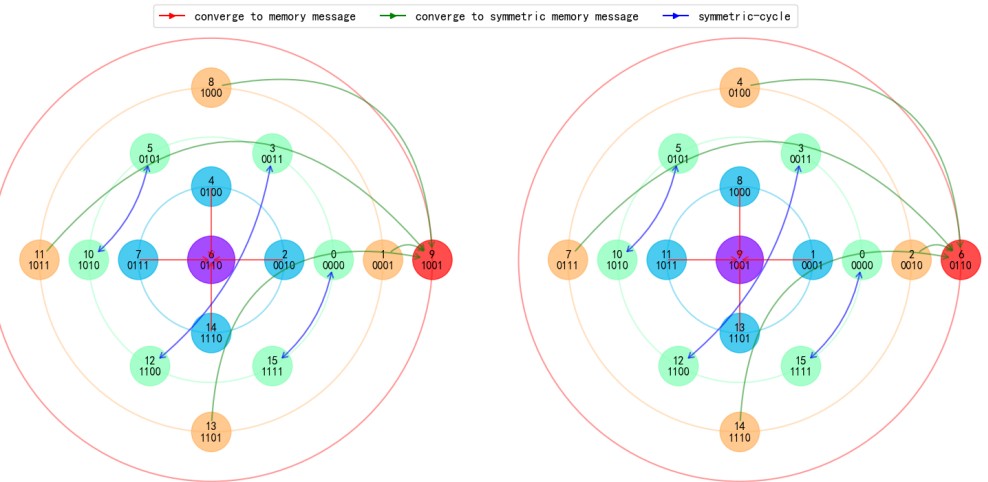

Figure 7: Evolution results of a HNN with $N = 4$, memorizing one message $\begin{bmatrix} -1 & 1 & 1 & -1 \end{bmatrix}^{\mathrm{T}}$ (The decimal representation is 6). Clear symmetry emerges along the evolution.

(Note: For ease of binary conversion in the diagram, the messages is composed of $\{0, 1\}$; however, in the actual experiments and theoretical derivations, all memories and noise patterns are binary vectors composed of $\{1, -1\}$. This will not be reiterated.)

**Case 2: $N$ odd.**

For example, when the number of neurons, $N = 5$, and one memory $\begin{bmatrix} -1 & -1 & 1 & 1 & -1 \end{bmatrix}^{T}$ is stored (converted to decimal as number 6), its symmetric memory is 25. According to Case 2 in Theorem 1, noise patterns with a Hamming distance less than 2.5 (i.e., Hamming distances 1 and 2) from 6 will converge to 6, and those with Hamming distances 1 and 2 from 25 will converge to 25.

The Figure 8 below shows the visualization of the simulation. The central circle in the left circular area represents the stored memory message, and the central circle in the right represents its symmetric memory message. Moving outward from the center, each ring indicates an increase of 1 in the Hamming distance. Red arrows start from a noise pattern and end at the final stable fixed point. Clearly, all noise nodes on the first and second rings (Hamming distances 1 and 2) around memory 6 converge to 6, and similarly, all noise nodes on the first and second rings around symmetric memory 25 converge to 25. The simulation results are entirely consistent with the theoretical derivation.

Additionally, the new symmetries in the noise pattern landscape are reflected in the following aspects:

- The sizes of the attraction basins for memory message 6 and its symmetric counterpart message 25 are identical (both are 16, including the messages themselves). Moreover, all noise patterns with a Hamming distance of 1 or 2 from message 6 converge to message 6, while all noise patterns with a Hamming distance of 1 or 2 from message 25 converge to message 25.

- Among the noise patterns that converge to message 6, those with a Hamming distance of 1 include $(2, 4, 7, 14, 22)$, while those with a Hamming distance of 1 that converge to

symmetric message 25 include $(29, 27, 24, 17, 9)$. The corresponding patterns are symmetric to each other (for instance, 2 and 29, 4 and 27, etc.). Similarly, among the noise patterns that converge to message 6, those with a Hamming distance of 2 include $(0, 3, 5, 10, 12, 15, 18, 20, 23, 30)$, while those with a Hamming distance of 2 that converge to message 25 include $(31, 28, 26, 21, 19, 16, 13, 11, 8, 1)$. The corresponding patterns are also symmetric to each other (for instance, 0 and 31, 3 and 28, etc.).

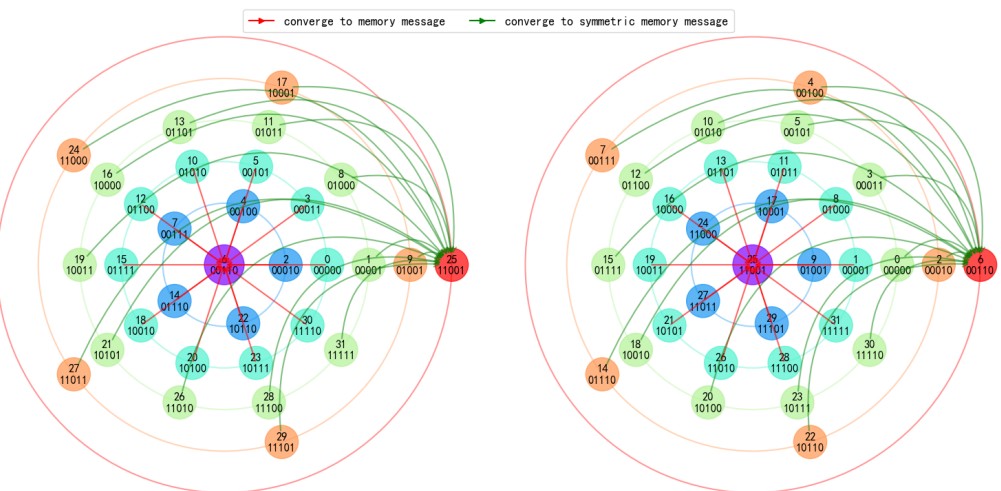

Figure 8: Evolution results of a HNN with $N = 5$, memorizing one message $\begin{bmatrix} -1 & -1 & 1 & 1 & -1 \end{bmatrix}^T$ (The decimal representation is 6). Clear symmetry emerges along the evolution.

### B.4.2 MEMORIZING TWO MESSAGES

**Case 1:** $N$ **even,** $r_{12}$ **even.**

For example, when the number of neurons $N$ is 6, two messages $\begin{bmatrix} -1 & -1 & -1 & 1 & 1 & -1 \end{bmatrix}^T$ (decimal 6) and $\begin{bmatrix} -1 & -1 & 1 & -1 & 1 & -1 \end{bmatrix}^T$ (decimal 10) are stored. The Hamming distance between them is 2. Their symmetric messages are 57 and 53, respectively. According to Corollary 3 and Corollary 4, there is no complete convergence domain, and $r_{max} = 1$. This means only noise patterns with a Hamming distance less than or equal to 1 from a memory message or its symmetric memory message can converge. Furthermore, according to the first case in Corollary 2, self-cycle, hetero-cycle, and symmetric-cycle occur.

The Figure 9 below shows the visualization of the simulation. To avoid clutter, the left panel only shows convergence trajectories, and the right panel only shows cycle trajectories. In the left panel, noise patterns converging to message 6, 10 and their symmetric message 57, 53 all have a Hamming distance of 1 from the respective memory. Not all six noise patterns on the first ring around message 6 converge to 6, indicating the absence of complete convergence domain, consistent with the theoretical derivation. In the right panel, orange lines represent self-cycles (e.g., between 9 and 58, 40 and 27, 49 and 61). Purple lines represent hetero-cycles (e.g., 29, 45, 60, 63 all update to 49, thus falling into the cycle between 49 and 61). Blue lines represent symmetric-cycles (e.g., between 31 and 32, 28 and 35, which are mutual symmetric noise patterns).

The specific cycle type a noise pattern falls into can be determined using Theorem 2. For example, consider noise pattern 45, represented by $V = \begin{bmatrix} 1 & -1 & 1 & 1 & -1 & 1 \end{bmatrix}^T$, and the memories $\xi^1 = \begin{bmatrix} -1 & -1 & -1 & 1 & 1 & -1 \end{bmatrix}^T, \xi^2 = \begin{bmatrix} -1 & -1 & 1 & -1 & 1 & -1 \end{bmatrix}^T$. The Hamming distance between $V$ and $\xi^1$ is $r_1 = 4$, and with $\xi^2$ is $r_2 = 4$. Since $r_1 + r_2 = 8 \neq 6 = N$, the **Condition I** in Theorem 2 is met. Furthermore, since $r_1 + r_2 = 8 \geq 7 = N + 1$, we need to check if $V$ contains

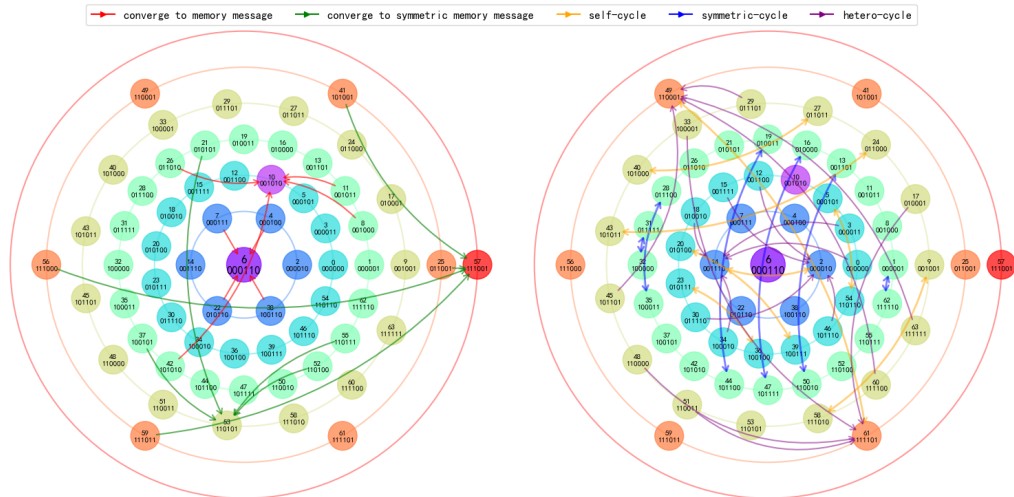

Figure 9: Evolution results of a HNN with $N = 6$, memorizing two messages $\xi^1 = \begin{bmatrix} -1 & -1 & -1 & 1 & 1 & -1 \end{bmatrix}^T$ (decimal 6) and $\xi^2 = \begin{bmatrix} -1 & -1 & 1 & -1 & 1 & -1 \end{bmatrix}^T$ (decimal 10). Clear symmetry emerges along the evolution. To avoid clutter, the left panel only shows convergence trajectories, and the right panel only shows cycle trajectories. In the left panel, noise patterns converging to message 6, 10 and their symmetric message 57, 53 all have a Hamming distance of 1 from the respective memory. Not all six noise patterns on the first ring around message 6 converge to 6, indicating the absence of complete convergence domain, consistent with the theoretical derivation. In the right panel, orange lines represent self-cycles (e.g., between 9 and 58, 40 and 27, 49 and 61). Purple lines represent hetero-cycles (e.g., 29, 45, 60, 63 all update to 49, thus falling into the cycle between 49 and 61). Blue lines represent symmetric-cycles (e.g., between 31 and 32, 28 and 35, which are mutual symmetric noise patterns).

a $V_{11}$ part. As explained in Section 4.1 of the main text, the $V_{11}$ part consists of components where $V = \xi^1 = \xi^2$. Observing the second component, all three values are $-1$, so the $V_{11}$ part exists, and thus the pattern falls into a hetero-cycle. Another example: noise pattern 35 has Hamming distances $r_1 = 3$ from $\xi_1$ and $r_2 = 3$ from $\xi_2$. Here, $r_1 + r_2 = 6 = N$ and $r_1 = r_2$, satisfying the **Condition III** in Theorem 2, resulting in a symmetric-cycle.

Additionally, the new symmetries in the noise pattern landscape are reflected in the following aspects:

- Memory messages 6, 10 and their symmetric counterparts 57, 53 possess attraction basins of identical size (each of size 5, including the pattern itself). Moreover, the noise patterns within these basins all have a Hamming distance of 1 from the memory pattern to which they converge.

- Two noise patterns trapped in symmetric-cycle are mutually symmetric(for instance, 1 and 62, 13 and 50, etc.).

- Noise patterns that enter self-cycle maintain equal Hamming distances to the original memory message, and consequently also to its symmetric counterpart. For example, patterns 2 and 14 both have a Hamming distance of 1 from memory message 6, while patterns 49 and 61 both have a Hamming distance of 5 from memory message 6. Similarly, patterns 5 and 54 both have a Hamming distance of 4 from memory message 10, as do patterns 20 and 39.

- Patterns entering hetero-cycle also exhibit equal Hamming distances between corresponding members. For instance, patterns 29, 45, 60, and 63 all first evolve to pattern 49 before entering the self-cycle between 49 and 61, while patterns 17, 33, 48, and 51 all first evolve to pattern 61 before entering the same cycle. Between these two groups of noise patterns, corresponding pairs have identical Hamming distances (for instance, the Hamming distance between 29 and 17 is 2, as is that between 45 and 33, 60 and 48, and 63 and 51).

**Case 2:** $N$ **even,** $r_{12}$ **odd.**

For example, when $N = 6$, two messages $\xi^1 = \begin{bmatrix} -1 & -1 & -1 & 1 & 1 & -1 \end{bmatrix}^T$ (decimal 6) and $\xi^2 = \begin{bmatrix} -1 & -1 & 1 & -1 & 1 & 1 \end{bmatrix}^T$ (decimal 11) are stored. The Hamming distance between them is 3. Their symmetric memories are 57 and 52, respectively. According to Corollary 3 and Corollary 4, $r_{ccd} = 1$ and $r_{max} = 2$. This means all noise patterns with a Hamming distance of 1 from the message or its symmetric message will converge to that message; only noise patterns with Hamming distance less than or equal to 2 might converge. Furthermore, according to the second case in Corollary 2, no cycle exist; all noise patterns will converge.

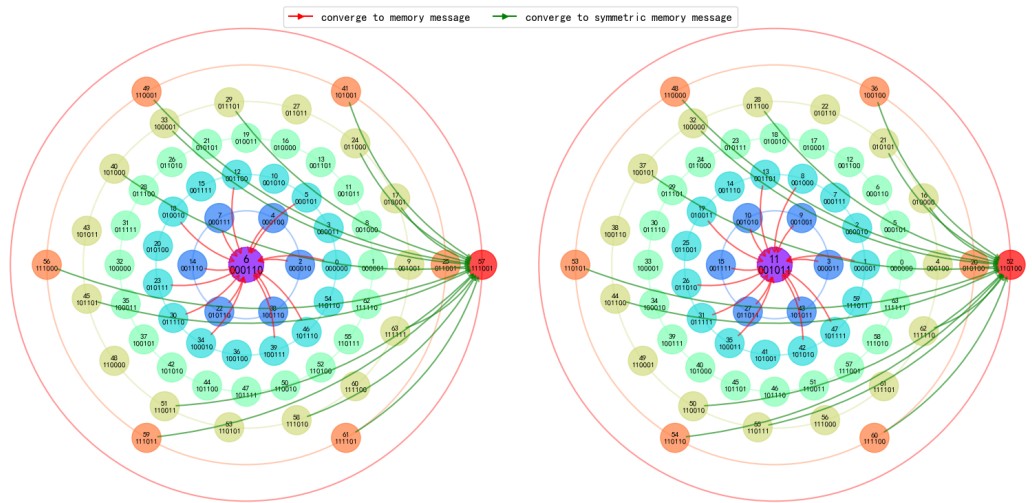

Figure 10: Evolution results of a HNN with $N = 6$, memorizing two messages $\xi^1 = \begin{bmatrix} -1 & -1 & -1 & 1 & 1 & -1 \end{bmatrix}^T$ (decimal 6) and $\xi^2 = \begin{bmatrix} -1 & -1 & 1 & -1 & 1 & 1 \end{bmatrix}^T$ (decimal 11). Clear symmetry emerges along the evolution. To avoid clutter, the left panel shows convergence for message 6 and its symmetric message 57, and the right for message 11 and its symmetric message 52. All noise patterns converge; no cycle occur. All noise patterns on the first ring (Hamming distance 1) around message 6 converge to 6, and similarly for message 11 in the right panel. (To save space, visualizations for symmetric message 57 and 52 are not shown, but the observation is consistent: all noise patterns on the first ring around a symmetric message converge to that symmetric message). Furthermore, in the visualization centered on message 6, only noise patterns on the first two rings (Hamming distances 1 or 2) can converge to 6, consistent with the $r_{max} = 2$.

The figure above shows the visualization of the simulation. To avoid clutter, the left panel shows convergence for message 6 and its symmetric message 57, and the right for message 11 and its symmetric message 52. All noise patterns converge; no cycle occur. All noise patterns on the first ring (Hamming distance 1) around message 6 converge to 6, and similarly for message 11 in the right panel. (To save space, visualizations for symmetric message 57 and 52 are not shown, but the observation is consistent: all noise patterns on the first ring around a symmetric message converge to that symmetric message). Furthermore, in the visualization centered on message 6, only noise patterns on the first two rings (Hamming distances 1 or 2) can converge to 6, consistent with the $r_{max} = 2$.

Why do some noise patterns within the convergence domain of a message not converge to it? For example, pattern 15 has a Hamming distance of 2 from message 6 but ultimately converges to message 11. There are two explanations:

- Noise pattern 15 has a Hamming distance of 1 from message 11, placing it within the **complete convergence domain** of message 11, hence it converges to 11.

- Consider noise pattern $V = \begin{bmatrix} -1 & -1 & 1 & 1 & 1 & 1 \end{bmatrix}^T$ and the message $\xi^1 = \begin{bmatrix} -1 & -1 & -1 & 1 & 1 & -1 \end{bmatrix}^T, \xi^2 = \begin{bmatrix} -1 & -1 & 1 & -1 & 1 & 1 \end{bmatrix}^T$. Noise pattern $V$ con-

tains a $V_{11}$ part (components 1, 2, 5), a $V_{21}$ part (component 4), and a $V_{22}$ part (components 3, 6). The Hamming distances are $r_1 = 2$, $r_2 = 1$. Since $r_2 + r_1 = 3 \leq 5 = N - 1$, $r_2 - r_1 = -1 \leq -1$, and it only contains $V_{11}$, $V_{21}$, and $V_{22}$ parts, this matches case 2.2.3 in the Proof of Theorem 2 in Appendix A.3. According to the conclusion, it will eventually converge to $\xi_2$, i.e., 11.

Additionally, the new symmetries in the noise pattern landscape are reflected in the following aspects:

- The attraction basins of memory messages 6, 11 and their symmetric counterparts 57, 52 are identical in size (each containing 16 patterns, including the memory message itself). Moreover, all noise patterns in these basins have a Hamming distance of either 1 or 2 from the memory message to which they converge.

- Among the noise patterns converging to memory message 6, those with a Hamming distance of 1 are $(2, 4, 7, 14, 22, 38)$. The corresponding noise patterns with a Hamming distance of 1 that converge to its symmetric counterpart 57 are $(61, 59, 56, 49, 41, 25)$. Each corresponding pair of patterns is symmetric to the other (for instance, 2 and 61, 4 and 59, etc.). The same rule applies to memory message 11 and its symmetric counterpart 52.

Cases where $N$ is odd, and $r_{12}$ is either odd or even, can be analyzed similarly, corresponding to the third and fourth cases in Corollary 2, respectively. In these cases, there are no symmetric-cycle, only self-cycle and hetero-cycle.

## C    RELATED WORKS

### C.1    HISTORY OF HNNS

In the 1920s, the Ising model was first introduced as a magnetic model, primarily focused on the thermal equilibrium state, which remained static over time. However, in 1972, Amari (1972) proposed a modification to the Ising model by adjusting the weights based on the Hebbian learning rule (Hebb, 1949), thereby presenting it as a model for associative memory. This concept was echoed by Little (1974) in 1974, whose work was later cited in Hopfield's 1982 paper. In 1982, Hopfield (1982) established the HNN, which is a spin-glass system designed to simulate neural networks, providing a framework for content-addressable memory systems with both binary threshold nodes and continuous variables. Additionally, the HNN was instrumental in enhancing our understanding of human memory. By 1984, Hopfield (1984) had further developed the network with continuous dynamics.

In recent years, there has been substantial progress in the development of continuous dynamics models with large memory storage capacity, which are now referred to as dense associative memory networks. A significant breakthrough came when the statistical and thermodynamic equivalence between the hybrid Boltzmann machine (HBM) and the HNN was rigorously established Barra et al. (2012). Building on this equivalence, a new approach was introduced for simulating the HNN using HBM, leading to an analysis of the advantages of HBM in memory storage and updates. Moreover, the relationship between HBM and the glass state transition of the HNN was examined, providing valuable insights for model optimization Agliari et al. (2013). More recently, in 2023, Ota analyzed the attentional Boltzmann machine (AttnBM), a novel variant of the Boltzmann machine derived from modern HNNs. This model features a tractable likelihood function and gradient for certain cases, is easy to train, and reveals connections to other single-layer models such as Gaussian-Bernoulli restricted Boltzmann machines and denoising autoencoders Ota & Karakida (2023).

In the past decade, significant research has focused on enhancing the storage and message processing capabilities of the HNN. A major avenue of this research has been the optimization of the energy function and update rules to improve network performance. In 2016, Krotov & Hopfield (2016) made substantial progress by modifying the network dynamics and energy functions, which led to an increase in the memory storage capacity of the HNN. This work was further extended in 2022 when Millidge introduced a unified framework for HNNs, building upon the contributions of Krotov and Hopfield Krotov & Hopfield (2021). Millidge's framework defined a general energy function based on local computational neural network dynamics and derived the corresponding neurodynamic equations, providing a more comprehensive understanding of HNN operations Millidge et al. (2022).

### C.2    MEMORY CAPACITY

The study of memory capacity in HNNs has evolved significantly over time, with numerous contributions building on previous work to enhance the understanding of HNNs' storage abilities. In 1982, Hopfield (1982) observed that experimental results showed negligible errors when the number of memory messages stored was less than approximately $0.14N$, where $N$ represents the number of neurons in the network. Three years later, Abu-Mostafa & St-Jacques (1985) formalized the concept of memory capacity for general memory systems and estimated the storage capacity of HNNs, revealing that the asymptotic capacity of an HNN with $N$ neurons is $N^3$ bits, with the number of stable state vectors bounded above by $N$. In 1987, McEliece et al. (1987) established that the theoretical capacity of an HNN is $N/(2\ln N)$, while the same year Kanter & Sompolinsky (1987) demonstrated that the memory capacity of the network could reach $N$, providing a more precise understanding of how HNNs store memories. Their findings formed a critical theoretical foundation for exploring the memory capabilities of the HNN.

A breakthrough in 1991 by Chiueh & Goodman (1991) expanded upon these findings, showing that under specific conditions and model settings, the memory capacity of HNNs could exhibit exponential growth. Later, in 1997, Storkey (1997) proposed that the memory capacity of HNNs could be increased to $N/(\sqrt{2\ln N})$, further advancing the understanding of how memory capacity can be optimized. The 2016 proposal of the Dense Associative Memory (DAM) model by Krotov & Hopfield (2016) represented another significant development, followed by Bao's theoretical proof of polynomial memory capacity (Bao et al., 2022). More recently, Bao extended this work to associative memory networks, deriving memory capacity bounds along with convergence radii for

various HNN models, including binary DAM networks, binary modern HNNs, and binary spherical HNNs (Bao & Zhao, 2025).

## C.3 CONVERGENCE, AND CYCLIC DYNAMICS

In 1984, Peretto (1984) unified Hopfield and Little networks under a statistical-mechanics framework, establishing when symmetric weights guarantee a well-defined energy function and convergence. In 1987, McEliece et al. (1987) made notable observations in his experiments, identifying three distinct types of convergence behaviors in HNNs under synchronous update: one-step convergence, two-step convergence, and multi-step convergence. In the same year, Baldi & Venkatesh (1987) explored the number of stabilization points in spin-glass systems, contributing to the understanding of convergence phenomena in HNNs. Later in 1988, Komlós & Paturi (1988) provided rigorous mathematical proofs concerning the convergence properties and error-correcting capabilities of Hopfield's associative memory model. They established the conditions under which the model could correct a linear number of arbitrary errors and demonstrated the existence of a large domain of attraction. Their work also proved the presence of an exponential number of stable extraneous memories, offering significant insights into the robustness and functionality of the HNN model. Then in 1997, Loukianova (1997) derived a lower bound for the memory capacity of HNNs in the presence of errors, contributing further to the study of error-tolerant memory systems. Two years later, in 1999, Bovier (1999) provided a strict upper bound for perfect convergence in HNNs, advancing the theoretical understanding of convergence behavior in HNNs. Recently, in 2018, Folli et al. (2018) systematically mapped how dilution and asymmetric couplings reshape the attractor landscape, revealing phase-transitionlike transitions between stable, cyclic, and chaotic regimes in binary recurrent networks. Later, Hwang et al. (2019) derived analytic expressions for the exponential growth rates of limit cycles in asymmetric networks and identified parameter thresholds at which synchronous dynamics transition into long periodic or chaotic behaviors.

These classical results collectively demonstrate that convergence and cyclicity in Hopfield-type systems are highly sensitive to symmetry, dilution, and the degree of asymmetry in the weight matrix. Our work differs fundamentally in that we isolate the role of the update rule itselfparticularly the zero-summation ambiguityand show that resolving this ambiguity restores symmetry, yields fully characterizable dynamic regimes, and enables exact enumeration of convergence versus cyclic states.

## C.4 OTHER FRONTIER RESEARCHES

In 2012, Barra et al. (2012) proposed that the HNN exhibits a certain equivalence to the Boltzmann machine, furthering the understanding of the connections between these two models in the context of associative memory and statistical mechanics. In 2021, Smart expanded on this idea, suggesting that the HNN could be considered equivalent to a confined Boltzmann machine. He made a breakthrough by employing QR decomposition to establish the exact mapping between the HNN with correlated patterns and the restricted Boltzmann machine (RBM). This innovation broke through the limitations of previous research, which primarily focused on uncorrelated modes, and opened up a promising new direction for further exploration (Smart & Zilman, 2021). In 2021, Tyulmankov et al. (2021) conducted studies on biological learning processes in key-value memory networks, exploring the intersection of biological systems and computational models. That same year, Liang et al. (2021) investigated whether fruit flies could learn word embedding methods, offering insights into the potential for non-human biological systems to engage with machine learning concepts.

In 2023, Schäfl et al. (2023) demonstrated the applicability of modern Hopfield networks to tabular data, showcasing their effectiveness on structured datasets. In 2024, Xu et al. (2024) leveraged generalized sparse Hopfield layers in BiSHop to tackle non-rotational invariance and feature sparsity in tabular data. Additionally, Krotov (2023) published a thorough survey of contemporary Hopfield research, charting the fields emerging directions. In 2024, Lucibello et al. advanced our understanding of dense associative memory models by establishing conditions for exponential capacity scaling, thus illuminating paths toward highly scalable memory architectures (Lucibello & Mézard, 2024). Concurrently, Agliari et al. (2024) employed Hopfield-like networks to mitigate overfitting and improve generalization, bolstering neural robustness in real-world tasks. Rolls (2024) offered a detailed comparative analysis of the hippocampal memory system versus generative AI models, extracting design insights for future neuro-inspired AI systems and guiding interdisciplinary research

at the nexus of neuroscience and machine learning. Most recently, Cabannes et al. (2024) dissected the training dynamics of associative memory networks under gradient descent, revealing how optimization trajectories shape memory retrieval performance.

In recent years, the modern Hopfield network landscape has rapidly evolved, driven by both theoretical innovations and practical applications. Specifically, Hu et al. (2023) introduced sparsemaxbased attention to enhance retrieval error bounds and memory capacity in Hopfield models, and in 2024 they extended this work with a nonparametric framework (Hu et al., 2024b) and kernelized variants (Hu et al., 2024e)each offering sub-quadratic complexity and rigorous capacity guaranteesand proposed the U-Hop+ algorithm for sub-linear retrieval time, alongside fine-grained complexity analyses that reveal efficiency phase transitions based on pattern norms (Hu et al., 2024c). Concurrently, Li et al. (2024) applied circuit complexity theory to delineate fundamental expressive limits of modern and kernelized Hopfield networks, while Niu et al. (2024) demonstrated in-material Oja's rule adaptation for energy-efficient associative memory. Santos et al. (2024) bridged FenchelYoung losses with sparse and structured Hopfield energies, extending the framework to SparseMAP transformations (Dos Santos et al., 2024). Wu et al. (2024) further boosted capacity through a twostage U-Hop retrieval with learnable feature mappings , and Hu et al. (2024) addressed outlier inefficiencies in large Transformer-based Hopfield models with OutEffHop, maintaining exponential capacity and quantization robustness (Hu et al., 2024a). Finally, Hu et al. (2025) provided a comprehensive statistical analysis of conditional diffusion transformers and clarified prompt-tuning efficiency limits in foundation models (Hu et al., 2024f;d).

The 2024 Nobel Prize in Physics was awarded to John J. Hopfield and Geoffrey E. Hinton for their foundational contributions to machine learning through artificial neural networks, which reignited interest and enthusiasm for HNNs. This accolade highlights the continued relevance and impact of Hopfield's pioneering work, bringing renewed attention to the rich potential of HNNs in both theoretical and practical domains.

