# OpenReview forum: "Symmetric Beauty of Hopfield Neural Networks with an In-Depth Analysis and Insights"
_ICLR.cc/2026/Conference — ICLR 2026 Conference Withdrawn Submission_

### Official Review · Reviewer_gWDn · 2025-10-27

**Soundness:** 1
**Presentation:** 3
**Contribution:** 1
**Rating:** 2
**Confidence:** 3

**Summary:**

The authors introduce a modification to the Hopfield model with a sign function nonlineartiy that takes into account ties, and then they full characterize the behavior of the model when there are either 1 or 2 memories.

**Strengths:**

The problem they considered was very clear and explained well. As far as I could tell all their results, including the proofs, were correct, and they were backed up by simulations.

**Weaknesses:**

I personally could not understand why the problem they considered was especially relevant. The motivation in the introduction is that deep networks need to be interpretable, and the "Transformer’s attention mechanism is formally equivalent to the continuous-state
update rule of modern Hopfield networks", therefore it makes sense to study Hopfield networks. However, the network they studied was not the one that was equivalent to Transformer’s attention mechanism, and they looked only at 1 and 2-memory networks.

**Questions:**

The main thing that would change my opinion would be an explanation of why this problem is relevant.

---

> ### Author Response · Authors · 2025-11-27
>
> We are very grateful for the valuable feedback from the reviewers. We spent a considerable amount of time supplementing the experiments and revising the article to better meet the standards of the ICLR conference. Therefore, we are a week late in responding. We hope that all reviewers will carefully read our response and the revised article, and we welcome any further comments. We will respond as soon as possible.
>
> Thank you for your comment. We must honestly tell you that the research content of this paper indeed has no direct connection with Transformers, nor can the contributions of this paper be applied to Transformers. In fact, we were inspired by the article written by Ramsauer et. al and realized that there might be a close connection between Hopfield neural networks (HNNs) and Transformers, which led us to study HNNs. Additionally, The fact that ICLR 2025 hosted a workshop on New Frontiers in Associative Memories is sufficient proof that research on Hopfield neural networks is gaining attention.
>
> The main work of this paper is focused on fixing the issue of symmetry breaking in classical HNNs, as well as conducting a comprehensive and detailed dynamical analysis of HNNs using Hamming distance. The primary contribution of this work lies in our finding that improper handling of zero summations in the activation function of HNNs can lead to a loss of symmetry in their dynamical evolution. In response to this issue, we propose a Rectified Update Rule that preserves the previous state whenever the summation equals zero, and we provide theoretical guarantees for the cases of one and two stored memory messages. While the HDAR update rule itself is not intended for use in Transformer architectures, the underlying idea—namely, the proper treatment of zero-sum conditions in activation functions—may offer valuable insights and could potentially be adapted for use in Transformer models. We strongly encourage you to read the paper carefully; all the theorems in the paper have detailed proofs. We have also conducted extensive simulations to verify the theoretical results and created visualizations to aid your understanding. Your feedback is appreciated.
>
> If you are satisfied with our response, please increase the score. We look forward to having a deeper discussion with you at the conference.

---

> > ### Comment · Reviewer_gWDn · 2025-11-27
> > **reply from gWDn**
> >
> > So I should view this as a purely theoretical exercise? Which is not a bad thing; I personally love theoretical exercises. However, before I look more closely at the paper, I have two questions.
> >
> > 1. I would have thought that $P(X_i(t)=0) = 0$. Is that not the case?
> >
> > 2. You say "Some implementations introduce an ad hoc sign (sgn) function that defaults to a fixed state (+1 or −1) when encountering this scenario, but such heuristics will disrupt the network’s symmetry and destabilize its dynamics, as noted in [list of references]". I looked at the first three papers in the list of references, and I didn't see that statement. Granted, I only skimmed the papers, so I may have missed it. But could you tell me exactly which paper to look at, and where to look?

---

> ### Author Response · Authors · 2025-11-27
>
> Thank you for your response! We are happy to provide further clarification.
>
> (1) We would like to point out that the probability of
> \\(X_i(t)=0\\) occurring is not zero. We can illustrate this issue with a simple example. Consider a Hopfield neural network storing a single message, with the number of neurons \\(N = 5\\). Suppose the message is \\(\xi = \begin{bmatrix} -1 & -1 & -1 & -1 & 1 \end{bmatrix}^{T}\\). According to the construction of the weight matrix \\(W = \xi\xi^T - I\\), we obtain:
>
> \\(W = \\begin{bmatrix}
> 		0 & 1 & 1 & 1 & -1 \\\\
> 		1 & 0 & 1 & 1 & -1 \\\\
> 		1 & 1 & 0 & 1 & -1  \\\\
> 		1 & 1 & 1 & 0 & -1  \\\\
> 		-1 & -1 & -1  & -1 & 0
> 	\\end{bmatrix}.\\)
>
> Now, consider a noisy pattern \\(\begin{bmatrix} -1 & -1 & 1 & 1 & -1 \end{bmatrix}^{T}\\).
>
> Then, \\(X_3(t)=\sum_{j=1}^N w_{3,j}v_j(t) = (-1)\*1+(-1)\*1+1\*0+1\*1+(-1)\*(-1)=0\\), which is a zero-sum case. Precisely because such zero-sum cases can occur, the symmetry of the Hopfield network is broken under the original update rule. The significance of our Rectified Update Rule lies in restoring the symmetry of the Hopfield neural network; it ensures both the symmetry of the weight matrix and the symmetry of the noisy pattern landscape. A visual demonstration of the symmetry in the noisy pattern landscape can be found in Section B.4 **VISUAL PRESENTATION**. We hope this example helps clarify this point.
>
> (2) We are pleased to address your question. The first three references you mentioned all employ the "sgn" function. Specifically, in the first paper by Abu-Mostafa \& Jacques (1985), it appears at the bottom right of the first page; in the second paper by Sompolinsky \& Kanter (1986), it is found in equation (3) at the top left of the second page; and in the third paper by Kanter \& Sompolinsky (1987), it is located in equation (1.3) at the top left of the second page.
>
> To be more precise, in the first paper, you may refer to the paragraph immediately above where the "sgn" function appears. The key sentence states: "If the weighted sum over all of its inputs is greater than or equal to \\(t_i\\), the \\(i\\) th neuron turns on and its state becomes \\(+1\\)." This clearly indicates that the zero-sum case is directly assigned to \\(+1\\).
>
> Additionally, you may examine the fourth paper by McEliece et al. (1987), specifically equation (1.1) at the top left of the second page. This equation explicitly uses a piecewise function, setting the value to \\(+1\\) when \\(\sum T_{ij} x_j \geq 0\\). The paragraph above this equation also states: "the new state of the neuron is \\(-1\\) if the sum is negative, and \\(+1\\) if the sum (equals or) exceeds zero". We believe these examples help clarify that the conventional approach to handling the zero-sum case was simply to assign it directly to either \\(+1\\) or \\(-1\\). This also addresses your first question, confirming that the probability of \\( X_i(t) = 0 \\) is indeed not zero.
>
> Furthermore, we would like to clarify that the statement "such heuristics will disrupt the network’s symmetry and destabilize its dynamics" was not explicitly stated in these reference texts. This is a perspective we have developed based on our more in-depth experimental simulations. In fact, many previous authors in the field of Hopfield neural networks simply adopted the "sgn" function without further analyzing the impact of this implementation on the symmetry of the noisy pattern landscape in Hopfield networks. Perhaps the way we expressed ourselves has led to some misunderstanding. It is possible that we should have directly referenced these papers after explaining the conventional handling of the zero-sum case, without incorporating our own experimental judgments. We acknowledge this shortcoming and will address it in our subsequent revisions.
>
> Thank you once again for your feedback. We look forward to further in-depth discussions regarding our manuscript.

---

### Official Review · Reviewer_pfQT · 2025-10-31

**Soundness:** 3
**Presentation:** 2
**Contribution:** 1
**Rating:** 2
**Confidence:** 3

**Summary:**

An analysis of the Hopfield model using synchronous updates, showing the spectrum of dynamic behaviour for a single or two patterns stored in the network. Also suggest a modified dynamic rule, based on the Hamming distance.

**Strengths:**

* A thorough analysis of the dynamics for the original and modified ("Hamming-based") dynamics.
* A beautiful visualization of the spectrum of dynamic conditions (Figure 1 and 2).

**Weaknesses:**

* Per the authors' summary, the problem of a Hopfield model with synchronous updates hasn't been under active research between 1987 and 2022. Possibly because those systems cannot act as a useful memory device. The authors should convince the reader why they seek to reignite this line of research: why is it interesting to do the suggested analysis?
* Table 1 (left) should have been a sentence in the figure caption. Table 1 (right), 2, and 3 should have been plots.
* The exposition taking the first page in its entirely does not contribute to the paper and is not accepted in modern scientific writing.
* The contribution of the "Rectified Update Rule" is so small you don't need to specify it.

**Questions:**

* Can the suggested network act as a memory device? How many memories can it store for 1K neurons?

---

> ### Author Response · Authors · 2025-11-27
>
> (1) We are very grateful for the valuable feedback from the reviewers. We spent a considerable amount of time supplementing the experiments and revising the article to better meet the standards of the ICLR conference. Therefore, we are a week late in responding. We hope that all reviewers will carefully read our response and the revised article, and we welcome any further comments. We will respond as soon as possible. All revisions in the main text are in blue font for easier reading by reviewers. A new visualization section **B.4 VISUAL PRESENTATION** of the appendix has been added to the appendix.
>
> While it is true that classical synchronous Hopfield neural networks (HNNs) have not been a mainstream topic since the late 1980s, we believe this is mainly because they were not competitive as practical memory devices. Our aim in this paper is different: we use this model as a mathematically tractable prototype to study the detailed attractor structure of a recurrent energy-based system.
>
> The ICLR 2025 workshop on New Frontiers in Associative Memories is sufficient proof that research on HNNs is gaining attention. Our research on the theoretical structure of HNNs is of great significance.
>
> We view this work as an initial step toward understanding, within a controlled and analytically tractable framework, phenomena that similarly arise in modern energy-based models and contemporary Hopfield networks. We will make this motivation clearer in both the Introduction and Related Work sections.
>
> (2) Thank you for your suggestion. We will add line charts in Appendix **B.3 STATISTICAL ANALYSES** to display the data. Since we use exhaustive enumeration when \\(N \leq 20\\), the experimental and theoretical values are completely consistent; when \\(N > 20\\), we use Monte Carlo sampling. To more accurately observe the specific differences between experimental and theoretical values, we initially used a table format. Because the experimental runtime is long when \\(N\\) is large and full simulation is not feasible, we only selected a few specific \\(N\\) values (such as \\(100\\), \\(200\\), \\(500\\), \\(1000\\) on the right side of Table \\(1\\)).
>
> (3) We appreciate the valuable comment regarding the length and style of the initial exposition. We agree that the first page can be substantially shortened without loss of scientific content. In the revised version, we will move any non-essential contextual discussion to an appendix or remove it altogether, so that the introduction becomes more focused on the concrete problem, the gap in the literature, and our technical contributions.
>
> (4) Indeed, the modification to the Rectified Update Rule is very small, but it significantly alters the evolutionary landscape of the HNN. Using the Rectified Update Rule, we demonstrate that the evolutionary landscape of an HNN exhibits perfect symmetry when it memorizes one or two pieces of information. The primary contribution of this work lies in our finding that improper handling of zero summations in the activation function of HNNs can lead to a loss of symmetry in their dynamical evolution. In response to this issue, we propose a Rectified Update Rule that preserves the previous state whenever the summation equals zero, and we provide theoretical guarantees for the cases of one and two stored memory messages. While the HDAR update rule itself is not intended for use in Transformer architectures, the underlying idea—namely, the proper treatment of zero-sum conditions in activation functions—may offer valuable insights and could potentially be adapted for use in Transformer models.
>
> Questions: Can the suggested network act as a memory device? How many memories can it store for 1K neurons?
>
> Our response: The HNN is a fully connected, symmetric-weight recurrent neural net whose energy-minimizing dynamics serve as associative memory for pattern storage and retrieval. So we insist on that HNN can act as a memory device. However, the capacity of HNN is about \\(0.14N\\) (\\(1 K\\) neurons can store about \\(140\\) memory message), which is proved by McEliece (1987), which is precisely the main reason restricting the development of HNN. Recently, the DAM network proposed by Krotov in 2016 is a HNN with an exponentially larger network size, and the modern Hopfield network proposed by Ramsauer also has a large memory capacity. Therefore, the main problem restricting the development of HNNs—their small memory capacity—has been solved. What remains is the theoretical structure of HNNs. This paper studies precisely this theoretical structure problem, modifying its activation function to address the zero-sum problem, thus making the evolutionary results of HNNs symmetric, which corresponds well to the symmetry of the weight matrix of HNNs. The research method presented in this paper can be extended to the structural research of DAMs and modern HNNs.
> Please improve your score; we hope to have an in-depth discussion with you at the ICLR conference.

---

### Official Review · Reviewer_Y9ss · 2025-11-04

**Soundness:** 1
**Presentation:** 1
**Contribution:** 1
**Rating:** 2
**Confidence:** 4

**Summary:**

The paper “Symmetric Beauty of Hopfield Neural Networks: An In-Depth Analysis and Insights” investigates the relationship between the dynamic behavior of Hopfield networks and the properties of the stored memories, particularly their mutual distances in configuration space. The authors present an analysis that reveals a rich convergence phenomenology under a novel dynamical rule, which accounts for the possibility that a given initialization may be equidistant from multiple memories.

**Strengths:**

The main strength of the paper is the exact and controlled analytical framework where Authors work.

**Weaknesses:**

I believe that, in its current form, the paper has several weaknesses, which I outline below:

1. The research question—deriving the dynamic phenomenology of simple Hopfield networks—is clearly stated. However, the paper fails to clearly articulate the broader significance of this topic for the fields of associative memory and machine learning.

2. Although the Hopfield model has been extensively studied, it continues to raise interesting and relevant open questions, precisely because of its elegant and simple formulation. The problem addressed by the authors is indeed of interest, but the paper should devote more space to convincing the reader of the importance and novelty of their contribution. The main threat to the relevance of the results is the fact that, for a large size N of the system, sampling initialization that are exactly equidistant to several data-points - i.e. that yield a zero local field - is exponentially rare in N. For this reason it is not clear how the phenomenology explained by the paper should count in actual implementations of the model and why it should be so theoretically insightful.

3. The overall presentation of both the content and the results—particularly the experimental results—is very weak and requires substantial revision. In addition, the paper lacks mathematical rigor, and the current writing style does not meet the standards of a scientific publication. For instance, the conversational exchange between a student and a large language model in Section 1 is not appropriate for this context. Furthermore, the title of the article should be updated to a more professional version and perhaps one that is more specific to the topic covered.

As a result of these issues, I believe the paper is not yet ready for publication.

**Questions:**

I will now proceed to ask questions and raise issues that I would like the authors to address in order to both finalize my personal judgment over the paper and improve the manuscript. I will divide the issues into Major and Minor ones for an improved readability of the revision.

**Major issues:**

1. [120-121] how can $X$ reach zero value in large $N$ limit? How is this relevant to the dynamic behaviour of the network? As already mentioned in the "Weaknesses" Section, Authors should argue around the validity and relevance of their results by addressing this aspect (you can answer only once to both questions).
2. How does the dynamic phenomenology generalize to a larger amount of memories or to the case where the memory load scales as $p = \alpha N$, as in the usual Hopfield setup? Can the current approach explain the rich dynamic phenomenology of stable fixed points and limit cycles convergence observed in this regime? See works already referenced in Peretto (1984) and Section C3 of the paper.

**Minor issues:**

1. [040-041] References to the student’s episode, and definitely rephrase that part to include it in a scientific journal.
2. [057-058] Wrong reference to Hebb (2005?) across all the paper.
3. [059-060] Authors should be more specific on the type of dynamics to which this line refers to. Asynchronous dynamics over hopfield networks - that are symmetric in the couplings by construction - can only end up into stable fixed points. On the other hand, asynchronous dynamics can also lead to limit cycles of length 2 (Peretto (1984), Folli et al. (2018), Hwang et al. (2019)). I ask the authors to provide for more detailed reference about the fact that the number of stable fixed points is smaller than the 2-limit cycles as stated.
4. [097-098] It would be convenient to replace capital $W$ with small $w$, since couplings are later referred to as $w_{ij}$. Or at least specify that $w_{ij} are the elements of $W$.
5. [100-101] Authors may replace “quantity of” with “number of”.
6. [129] Authors may remove “elegant”.
7. [130] $U$ should be defined with respect to v used in Eq. (2). Also it appears as sometimes patterns, i.e. network configurations, are referred to as $U$, some other times as $V$. All the notation in the paper should be uniformed.
8. [131] Authors may split the fixed point and cycle definition into two definitions, and also rephrase the definition. For instance “and t is the minimal value” is not a clear expression.
9. [135-136] Authors should rephrase Theorem 1 as well for a better clarity.
10. [144-145] and elsewhere in the paper: replace $\overline{\xi}(\overline{\xi}=-\xi)$ with $\overline{\xi}=-\xi$ .
11. All quantities $C_{ab}$ in Corollary 2 and in the Appendices are not previously defined. May the authors explain why and how they are computed ?
12. Should not Corollary 3 be a condition on $r_{12}$ instead of $r_{ccd}$ ? How do $r_{ccd}$ and $r_{max}$ scale with $N$? In fact no information is provided about the value of $r_{ccd}$ except for condition (62). You can either provide for an estimate of $r_{ccd}$ as a function of $r_{12}$, and then Corollary 3 is well defined, otherwise you provide for a bound in $N$, and at that point Corollary 3 should turn into a condition on the distance between the memories, that is also the way it is naturally written.
13. Can you draw $r_{ccd}$ and $r_{max}$ in Figure 2? For a more clear explanation. Authors may also include a series of images where the mutual distances between memories change, and show how the dynamic phenomenology is changing.
14. [414-415] and [421]: I guess there is a typo and that authors have performed MCMC sampling when $N ≥10$.
15. The content of table 1 in Section 5.1 is not clear from both the table and the main text. For large networks, it should expressed more clearly that authors generate patterns at $r/N =\frac{1}{2} + O(N^{-1/2})$ and then change $N$ and that convergence ratio is then estimated. For small networks the procedure is not clear: should not the distance $r$ also be fixed, so to vary $N$ ?
16. The same comment above holds for Section 5.2. Also, the reader cannot know in what region of Figure 2 the patterns generated in Table 2 and 3 are. As a consequence the reader does not know what to expect. This should be clarified, as in general what the tables actually show.
17. Bibliography more homogeneous (i.e. all first names extended or shortened).
18. Errors in experiments in Appendix B.3 have an enormous quantity of digits that should be correctly approximated.

---

> ### Author Response · Authors · 2025-11-27
>
> Thank you for your suggestion. We have revised the experimental section as a whole, making the objectives and procedures clearer for readers to understand; at the same time, we have also made changes to the introduction and some of the theorems in the paper to make them more in line with academic standards. All revisions in the main text are in blue font for easier reading by reviewers. A new visualization section **B.4 VISUAL PRESENTATION** of the appendix has been added to the appendix.All revisions in the main text are in blue font for easier reading by reviewers.
>
> Major issues:
>
> (1) Thank you for your question. You suggested that the likelihood of zero-sum occurrences might decrease when \\(N\\) is relatively large. Regarding this point, we have not specifically studied whether the probability of zero-sum occurrences is related to the size of \\(N\\). In fact, according to our experimental observations, even when \\(N\\) is small, zero-sum situations do not occur frequently. However, it is precisely these infrequent zero-sum cases that disrupt the symmetry of the Hopfield neural network (HNN) under the old update rules. The significance of our **Rectified Update Rule** lies in restoring the symmetry of the HNN, ensuring that the weight matrix remains symmetric while also maintaining the symmetrical structure of the noise-pattern landscape. As for the relationship between the new rule for handling zero-sum occurrences and the network's dynamic behavior, it is inconvenient to describe in words. Therefore, we have added a new section **B.4 VISUAL PRESENTATION** in the appendix to demonstrate the network's dynamic behavior when the **Rectified Update Rule** is applied, along with figures and explanations to assist your understanding.
>
> (2) Thank you for your valuable suggestion. We anticipate that the **HDAR Update Rule** can be extended to characterize state distributions in scenarios involving three or more memory messages. However, a direct extension of the analytical approach employed for two memory messages quickly leads to a combinatorial explosiona challenge we are actively working to overcome. Despite this limitation, in the absence of more effective alternatives at present, we believe HDAR remains a promising and theoretically sound framework. The central challenge lies in devising strategies to mitigate the exponential growth in complexity associated with this generalization.
>
> The current method can fully explain the reasons and conditions for the occurrence of stable fixed points, self-cycles, heterocycles, and symmetric cycles when we have already analyzed the memory of one or two pieces of memory messages. Specific examples can be found in **B.4 VISUAL PRESENTATION** of the appendix. We believe that our work can be further extended to three or more pieces of memory messages and used, with the existing rules, to explain the dynamic behavior of complex networks.
>
> Minor issues:
>
> (1) Thank you for your suggestion. We have deleted the dialogue example between students and the large model to comply with academic paper standards. We revised the first four paragraphs of the Introduction to make them more closely related to the research question.
>
> (2) Thank you for your correction. The correct publication date of this paper is 1949, and we will make the necessary amendments.

---

> ### Author Response · Authors · 2025-11-27
>
> (3) We should indeed point out in this sentence that the network dynamics here are based on synchronous update rules. In fact, all the conclusions and experiments in this paper are based on the premise of synchronous update rules. According to our experiments and literature review, under asynchronous updates, all noise patterns will converge to a stable fixed point, and there will be no cases of being trapped in cycles. We have carefully read the three articles you mentioned. In Peretto's (1984) article, the author proved that the minimum point of the energy function of the HNN when it is memorizing one memory message or multiple orthogonal memory messages is the stored memory message. In the papers by Folli (1984) and Hwang (1985), the authors found the study of neural network evolution states under weight matrices determined by defining the sparsity and symmetry of the network very interesting. This paper also focuses on these evolution states, but it should be noted that this paper studies why the evolution states of HNNs are not symmetric, and gives the reason as improper handling of zero-sum problems by the activation function. The weight matrix of a HNN is uniquely determined by memory message, and it is not entirely equivalent to the case proposed by Folli and Hwang where the symmetry and sparsity are both zero. This is why their experimental results differ from our theoretical results. Thank you again for your valuable feedback. We will continue to follow their research, and I have also provided an introduction and citation of their papers in the relevant working section of the appendix.
>
> (4) Thank you for your suggestion. After careful consideration, we still use the uppercase letter W here, and then provide an explanation afterwards that \\(w_{ij}\\) is an element of \\(W\\) to make it easier for readers to understand.
>
> (5) Thank you for your correction. The quantity of \\(M\\) here is countable, so using ''number'' is indeed more accurate and precise. The word ''elegant'' has also been removed.
>
> (6) Thank you for your suggestion. We have revised the entire article and removed the word ''elegant''.
>
> (7) In terms of describing the noise model, the paper indeed presents two approaches. We will unify them, using \\(V\\) to represent it consistently, in order to enhance the readability of the article.
>
> (8) Thank you for your suggestion. We have rewritten the definition as you suggested and split it into two definitions.
>
> (9) Thank you. We have restated Theorem 1 to maintain mathematical rigor.
>
> (10) We apologize for this redundancy. What we intended to convey in this passage is that the noise pattern \\(V\\) converges to \\(\bar{\xi}\\). The content within the parentheses serves to explain \\(\bar{\xi}\\), informing the reader that \\(\bar{\xi}\\) represents the symmetric message of \\(\xi\\). Replacing it would not accurately express our intended meaning. Furthermore, we have standardized the subsequent notation, using the parenthetical clarification only at the first occurrence of \\(\bar{\xi}\\); thereafter, all instances of the symmetric message are represented uniformly by \\(\bar{\xi}\\).
>
> (11) Thank you for your suggestion. We indeed did not give the definition of \\(C_{\rm ab}\\). It refers to the number of noise patterns in which the HNN eventually evolves into a cyclical state when \\(N\\) is even and \\(r_{12}\\) is even. For example, \\(C_{\rm oo}\\) refers to the number of noise patterns in which the entire network eventually falls into a cyclical state when \\(N\\) is odd and \\(r_{12}\\) is odd (\\(C_{\mathrm{eo}}\\), \\(C_{\mathrm{oo}}\\), \\(C_{\mathrm{oo}}\\) is defined similarly). We have revised it in the description of Corollary 2. The derivation and proof of these formulas are explained in the appendix; please refer to section **A.4 PROOF OF COROLLARY 2**.

---

> ### Author Response · Authors · 2025-11-27
>
> (12) The conclusion of Corollary3 is correct and has been proven, but the way it is expressed may indeed be difficult to understand. Here, I will provide a more detailed explanation. Consider an N-dimensional HNN that stores two memory message, \\(\xi^1\\) and \\(\xi^2\\), with a Hamming distance of \\(r_{12}\\) between them. We cannot directly obtain a function for \\(r_{ccd}\\) in terms of \\(r_{12}\\) and \\(N\\). Instead, we determine the value of \\(r_{ccd}\\) by testing whether it satisfies the conditions required by two inequalities. For example, when \\(N = 6\\) and \\(r_{ccd}=3\\), if a certain \\(r_{ccd}\\) value is the maximum that satisfies \\(2r_{ccd}+1\leq r_{12}\leq N-(2r_{ccd}+1)\\) and \\(N \geq 4r_{ccd}+2\\), then this value is the final \\(r_{ccd}\\). The testing starts from \\(1\\) and increases sequentially. When \\(r_{ccd}=1\\), \\(3 \leq r_{12}=3\leq 3\\) satisfies the condition, and \\(N=6\geq4r_{ccd}+2=6\\) also meets the requirement, so the test passes. When \\(r_{ccd}=2\\), \\(4r_{ccd}+2=10>N\\), which clearly does not satisfy the condition. Therefore, the final value of \\(r_{ccd}\\) is \\(1\\). We hope that this specific example helps you understand how we determine \\(r_{ccd}\\).
>
> (13) In the description of Figure 2, the ranges for \\(r_{\rm ccd}\\) and \\(r_{\rm max}\\) are already provided, with \\(r_{\rm ccd}\\) shown in shaded pink and \\(r_{\rm max}\\) in shaded cyan. In **B.4  VISUAL PRESENTATION** of the Appendix, we have added specific examples to demonstrate the convergence phenomenon of the HNN, accompanied by a legend, which may help you better understand.
>
> (14) Thank you for pointing this issue. You are absolutely right—there was indeed an inaccuracy in our previous description. To clarify, we used exhaustive enumeration for networks with \\(N \leq 20\\), and Monte Carlo sampling for those with \\(N > 20\\). Additionally, due to space limitations, we only selected a portion of the \\(N\\) values to be presented in Tables \\(1\\) and \\(2\\). The entire experimental section has been revised accordingly. We appreciate your valuable feedback.
>
> (15) The content in Table \\(1\\) of Section 5.1 corresponds to the convergence of a HNN when remembering one message, so there is no need to consider the fixed \\(r\\) you mentioned. You only need to change \\(N\\) to calculate the convergence rate. Additionally, we have completely rewritten the experimental section to more accurately describe the experimental procedures to the readers.
>
> (16) Thank you for your comment. We will now provide a detailed explanation. The experiments in Tables \\(2\\) and \\(3\\) were conducted by keeping \\(r_{12}\\) fixed while varying \\(N\\). In Table \\(2\\), both the left and right sub-tables were conducted under the condition \\(r_{12}=3\\) (odd). Specifically, in the left table, all \\(N\\) values are odd. According to Corollary 2, this corresponds to the fourth case, meaning that in Figure \\(2\\) there should only be self-loops and cross-loops, with no symmetric loops (green \\(V\\)). In the right table, all \\(N\\) values are even. According to Corollary 2, this corresponds to the second case, which means there are no loop situations in Figure \\(2\\), and all noise patterns will converge. In Table \\(3\\), both the left and right sub-tables were conducted under the condition \\(r_{12}=4\\) (even). Specifically, in the left table, all \\(N\\) values are odd. According to Corollary 2, this corresponds to the third case, so Figure \\(2\\) should again show only self-loops and cross-loops, with no symmetric loops (green \\(V\\)). In the right table, all \\(N\\) values are even, which, according to Corollary 2, corresponds to the first case, meaning all loop scenarios should exist in Figure \\(2\\). In fact, Figure \\(2\\) encompasses all convergence cases and all loop scenarios, representing a comprehensive noise pattern. We fully understand your suggestion to create four separate figures for different cases, but due to space limitations, we ultimately chose a single comprehensive figure to summarize the situation of the HNN when storing two message.
>
> (17) Thank you for pointing this issue. We have checked every reference to fix this writing problem.
>
> (18) Thank you for your suggestion. The errors in Appendix **B.3** indeed contained a large number of numbers, which not only affected readability but also did not meet academic standards. Therefore, we have made corrections, keeping four decimal places to be consistent with the data in the tables.
>
> If you are satisfied with our response, please increase the score. We look forward to having a deeper discussion with you at the conference.

---

> > ### Comment · Reviewer_Y9ss · 2025-11-27
> > **Updated preprint not uploaded**
> >
> > I thank the Authors for having replied to my comments,
> > though I cannot find the updated pdf. Can you proceed to upload the updated version so that I can proceed with the revision?
> > Thanks

---

> > > ### Author Response · Authors · 2025-11-27
> > >
> > > We apologize, we uploaded the document to the wrong place earlier. We have now confirmed that it was uploaded correctly. We have revised our title as "Decoding the symmetry of Evolution Landscapes in Hopfield Neural Networks via Rectified Update Rule In-Depth Analysis". All revisions in the main text are in blue font for easier reading by reviewers. A new visualization section B.4 VISUAL PRESENTATION of the appendix has been added to the appendix. We are looking forward to your response. Thank you very much.

---

### Official Review · Reviewer_njiV · 2025-11-08

**Soundness:** 3
**Presentation:** 3
**Contribution:** 3
**Rating:** 6
**Confidence:** 4

**Summary:**

This work introduces the Rectified Update Rule for Hopfield Networks, which resolves tie scenarios by retaining a neuron's prior state. This restores network symmetry and ensures stable convergence. When memorizing two messages, incorporating Hamming distance into the neuron's transition dynamics preserves full symmetry between the two memories and their negations. This approach yields a complete taxonomy of dynamic regimes: convergence, self-cycles, hetero-cycles, and symmetric cycles.

**Strengths:**

1. This paper proposes the Rectified Update Rule,which retains the neuron’s previous state in such tie cases and offers insights potentially applicable to activation function design in DNNs.

2. For two-message memorization, this paper intorcuces a Hamming-Distance-Aware Rectified (HDAR) update rule. This rule induces perfectly symmetric network dynamics, categorizing them into four regimes: convergence, self-cycle, hetero-cycle, and symmetric-cycle.

**Weaknesses:**

1. Since a Transformer's attention is equivalent to a modern Hopfield network, we can use Hopfield theory to understand Transformer models. This movtivation is very intersting. However, it is unclear that what new understanding for the Transformer models using Hamming-Distance-Aware Rectified (HDAR) update rule.

2. The authors suggest that these insights could inform the design of novel activation functions for Deep Neural Networks (DNNs). Could you provide potential directions or concrete examples of how this might be achieved?

3. The presented results are based on a synchronous update rule. How would these results differ under an asynchronous update scheme?

4. The experimental results are challenging to interpret. Could the authors provide low-dimensional (e.g., 2D or 3D) visualizations to clarify the findings?

**Questions:**

please see weaknesses.

---

> ### Author Response · Authors · 2025-11-27
>
> We are very grateful for the valuable feedback from the reviewers. We spent a considerable amount of time supplementing the experiments and revising the article to better meet the standards of the ICLR conference. Therefore, we are a week late in responding. We hope that all reviewers will carefully read our response and the revised article, and we welcome any further comments. We will respond as soon as possible. All revisions in the main text are in blue font for easier reading by reviewers. A new visualization section **B.4 VISUAL PRESENTATION** of the appendix has been added to the appendix.
>
> (1) Thank you for your insightful question. You are correct that our **HDAR Update Rule** is not designed for direct application to Transformer models. The primary contribution of this work lies in our finding that improper handling of zero summations in the activation function of Hopfield neural networks (HNNs) can lead to a loss of symmetry in their dynamical evolution. In response to this issue, we propose a **Rectified Update Rule** that preserves the previous state whenever the summation equals zero, and we provide theoretical guarantees for the cases of one and two stored memory messages. While the HDAR update rule itself is not intended for use in Transformer architectures, the underlying idea—namely, the proper treatment of zero-sum conditions in activation functions—may offer valuable insights and could potentially be adapted for use in Transformer models.
>
> (2) Thank you for your interest. For applications of HNNs in deep learning, we recommend the work of Dmitry Krotov and Jerry Yao-Chieh Hu, which offers valuable insights in this area. We are also deeply interested in this line of research and intend to explore it further in our future works. The ICLR 2025 workshop on New Frontiers in Associative Memories is sufficient proof that research on HNNs is gaining attention. Our research on the theoretical structure of HNNs is of great significance. This paper finds that the symmetry affecting the evolutionary landscape of HNNs is due to the incorrect handling of zero-sum problems by the activation function. Our proposed Rectified Update Rule exhibits a perfectly symmetrical evolutionary landscape when remembering one or two pieces of information. This argument has been theoretically proven and is of great significance for studying the evolutionary landscape when remembering multiple pieces of information. It also provides a reference for how activation functions in DNNs handle zero-sum cases.
>
> (3) Thank you for your good question. We have already conducted both theoretical proof and experimental verification of the update results under the asynchronous update scheme. Below is a brief introduction to the situation under asynchronous updates, detailed content will be presented in our forthcoming paper. When the asynchronous update rule is adopted and one or two memory message are considered, all cyclic cases are eliminated. Previously, under synchronous conditions, convergence would occur to the noise patterns of the memory message and their symmetrical counterparts. Under asynchronous conditions, these will ultimately converge in the same manner, whereas noise patterns that previously fell into cyclic states will no longer converge to cycles but will instead converge to memory message or its symmetrical counterparts.
>
> (4) Thank you very much for your interest in our experiments. We provide you with a detailed explanation. The results of the low-dimensional visualization have already been presented in **B.4 VISUAL PRESENTATION** of the appendix. If you are satisfied with our response, please increase your score. We look forward to having a deeper discussion with you at the conference.

---

### Note · Authors · 2026-02-03

I have read and agree with the venue's withdrawal policy on behalf of myself and my co-authors.

---

### Meta-Review · Area_Chair_wFRx · 2026-01-06

**Summary:**

Most of the reviewers agreed that this paper is below bar for acceptance due to poor presentation, and poor motivation of the studied problem and its relevance to the community.

I commend the authors for significantly revising the submission based on reviewer feedback. For a rebusmission, perhaps targeting a venue more focused on the theory of associative memories and other theoretical constructs would be more appropriate? Or the reviewers should think hard about how to justify the importance of the problem studied in this paper and its implications for modern ML.

**Reviewer Concerns:**

The main concern is that the topic studied (detailed dynamics of Hopfield networks) is no longer relevant to the broader ML community. Additionally, the authors haven't justified why the problem they study is important -- I agree with one of the reviewers who points out that the probability of a zero-sum should go to 0 as N gets large. The authors acknowledge that even for small N, zero sums are very rare. Thus, their new dynamics should not significantly deviate from traditional dynamics with high probability.

The reviewers significantly revised the paper to improve the quality of presentation during discussion, which is great, and sets them up for a resubmission. Although I don't think that in the rebuttal they have justified enough why general ML researchers should care about the studied problem.

**Reviewer Scores:**

I do not think any of the reviewers would have updated their scores. The concerns of Reviewer njiV  were fairly well address, although they already had a fairly positive score.

---

### Decision · Program_Chairs · 2026-01-26

Reject